# Dynamic SAS-6 phosphorylation aids centrosome duplication and elimination in *C. elegans* oogenesis

Feifei Qi[1], Shanshan Yin[1], Xiangrui Yang[1], Ning Ju[1], Bohan Liu[1], Xing Zhang[1], Zixuan Zhu[2], Li Ji[1], Fuxin Zhang[1], Li Zhao[1], Ruoxi Wang[1], Min Liu [ID][1], Liangran Zhang[1], Huijie Zhao[1], Jun Zhou [ID][1,2][✉] & Jinmin Gao [ID][1,2][✉]

## Abstract

In most metazoans, centrosome elimination during oogenesis ensures accurate centriole inheritance in the zygote, yet the molecular mechanisms remain poorly understood. Here, we reveal a critical role for controlled SAS-6 phosphorylation in centrosome dynamics during oogenesis. Centrioles disassemble during late meiotic prophase, while the cartwheel protein SAS-6 exhibits dynamic behavior in early meiotic prophase. Purified SAS-6 undergoes phase separation in vitro, and overexpressed SAS-6 forms droplets in cells. Mass spectrometry and kinase assays reveal that SAS-6 is phosphorylated at its C-terminus in cells and in vivo, with CDK-1 identified as a direct kinase. This phosphorylation inhibits SAS-6 phase separation and weakens interactions between centriolar proteins. SAS-6 degradation confirms its role in centrosome stability, and CDK-1 activity is required for timely centriole disassembly. Phospho-mimetic and phospho-deficient mutants demonstrate that dynamic SAS-6 phosphorylation is essential for centrosome assembly and elimination. We propose that the disordered C-terminus of SAS-6 facilitates cartwheel stacking via multivalent weak interactions, promoting centriole stability. Phosphorylation disrupts these interactions, impairing centrosome duplication and promoting elimination during oogenesis.

**Keywords** Centrosome Elimination; SAS-6; Phase Separation; CDK-1; Oogenesis

**Subject Categories** Cell Adhesion, Polarity & Cytoskeleton; Development; Post-translational Modifications & Proteolysis

## Introduction

The centrosome is an important organelle in animal cells essential for mitotic spindle assembly, asymmetric cell division, cell polarity establishment, and cilium formation (Badano et al, 2005; Bornens, 2012; Chavali et al, 2014; Qi and Zhou, 2021). It consists of two centrioles with a nine-fold radial arrangement of microtubules and pericentriolar material (PCM). The ultrastructure of centrioles is highly conserved across eukaryotic organisms. Centrioles undergo cell-cycle coupled duplication, during which the assembly of a nine-fold radially symmetric ring-shaped cartwheel structure on the mother centriole serves as the starting building block (Woglar et al, 2022).

The SAS-6 homo-oligomers have been identified as the principal component of the cartwheel (Cottee et al, 2015; Guichard et al, 2017; Pelletier et al, 2006; van Breugel et al, 2011; Woglar et al, 2022). Despite relatively low sequence conservation, the SAS-6 molecular architecture remains remarkably conserved across species (Hodges et al, 2010). It comprises a globular head domain at the N-terminus, a middle coiled-coil domain, and an unstructured C-terminal region (Cottee et al, 2015; Kitagawa et al, 2011; van Breugel et al, 2011). Structural studies employing crystallography, electron microscopy, and atomic-force microscopy have revealed that SAS-6 proteins form dimers via a strong interaction between their coiled-coil domains, which then multimerize through weaker interactions between two head domains into nine-fold symmetric assemblies (Kitagawa et al, 2011; Nievergelt et al, 2018). Thus, the oligomerization of head domains forms the cartwheel ring, and the coiled-coil domain and the unstructured C-terminal region account for the spoke appearance. Cryo-electron microscopy analysis suggests that the cartwheel is built by multi-layers of SAS-6 rings that stack on top of one another (Nazarov et al, 2020; Vakonakis, 2021). Notably, these rings are connected in the more peripheral regions through their spokes (Klena et al, 2020; Nazarov et al, 2020). Electron tomography of centrioles from multiple species reveal that spokes from stacked SAS-6 rings merge first into binary and then bundles via the coiled-coil domains (Guichard et al, 2017; Klena et al, 2020; Nazarov et al, 2020). The C-terminal disordered region, potentially in conjunction with its interacting proteins, constitutes densities vertically connecting spokes. However, although the fine organization of the cartwheel structure has been elucidated, the molecular mechanisms governing SAS-6 ring stacking remain elusive.

While centrioles undergo cell cycle-coupled duplication and the accurate number of centrioles is inherited in daughter cells in mitosis, they are eliminated in the oocyte during oogenesis in many metazoan species, including the starfish, *Drosophila melanogaster*, *Caenorhabditis elegans*, mouse and human (Hertig and Adams, 1967; Sathananthan et al, 1996; Szollosi et al, 1986). Such an elimination process is believed to be a vital physiological process to prevent excessive centrioles in the

[1]Center for Cell Structure and Function, College of Life Sciences, Shandong Provincial Key Laboratory of Animal Resistance Biology, Collaborative Innovation Center of Cell Biology in Universities of Shandong, Shandong Normal University, Jinan, China. [2]State Key Laboratory of Medicinal Chemical Biology, Haihe Laboratory of Cell Ecosystem, Tianjin Key Laboratory of Protein Science, College of Life Sciences, Nankai University, Tianjin, China. ✉E-mail: junzhou@sdnu.edu.cn; jinmingao@sdnu.edu.cn

fertilized eggs. Compared with centrosome assembly, the molecular mechanisms underlying centrosome elimination remain less studied. A few recent studies have shed light on centrosome elimination strategies used by different organisms. In starfish oocytes, three centrioles, including two mother centrioles and one daughter centriole, are extruded into polar bodies during the two rounds of meiotic divisions. The remaining centriole is subsequently destroyed by unknown mechanisms after PCM shedding at anaphase of meiosis II (Borrego-Pinto et al, 2016; Schoborg and Rusan, 2016). In *D. melanogaster*, oocytes inherit all centrioles from their neighboring nurse cells, forming an aggregate termed "centriole complex" (Mahowald and Strassheim, 1970). The centriole complex has been suggested to be eliminated in two steps: PCM components are lost first due to the down-regulation of polo kinase activity, followed by the disappearance of centriole components during late oogenesis (Pimenta-Marques et al, 2016; Schoborg and Rusan, 2016). The centrosome elimination process has also been examined during oogenesis in *C. elegans*, and it takes place during late meiotic prophase (Kim and Roy, 2006; Mikeladze-Dvali et al, 2012; Pierron et al, 2023). A recent novel study demonstrated that centriolar elimination begins with the loss of the central tube, followed by microtubule disassembly (Pierron et al, 2023). In addition, several regulators, such as CGH-1 helicase (Mikeladze-Dvali et al, 2012), have been proposed to contribute to the elimination process. However, the precise molecular mechanisms remain unclear.

Here, we combined live-cell imaging, cytological analysis, and biochemical approaches in the *C. elegans* germline to investigate centrosome elimination and its underlying mechanisms. We uncovered a dynamic property of the centriole protein SAS-6, which forms the highly ordered cartwheel structure, suggesting that weak interactions drive its assembly in vivo. Using mammalian cells and in vitro assays, we demonstrated that SAS-6 undergoes phase separation, regulated by CDK-1-mediated phosphorylation at its C-terminus. Phosphorylation attenuates both intramolecular inter-actions of SAS-6 and its binding to SAS-4. Furthermore, human SASS6 also shows phosphorylation-regulated phase separation at its C-terminus, suggesting evolutionary conservation. Genetic and cytological analyses further revealed that SAS-6 phosphorylation regulates centrosome assembly and elimination during oogenesis. We propose a model in which CDK-1-mediated phosphorylation disrupts the weakly interacting cartwheel structure, destabilizing the centriole and promoting centrosome elimination.

# Results

## Centrioles are disassembled during late meiotic prophase

Six conserved proteins essential for the initiation of centriole duplication have been identified in *C. elegans*, including SAS-7, SPD-2, ZYG-1, SAS-6, SAS-5, and SAS-4 (Delattre et al, 2004; Kemp et al, 2004; Leidel et al, 2005; Leidel and Gonczy, 2003; O'Connell et al, 2001; Sugioka et al, 2017). The availability of fluorescent protein-tagged lines for these proteins allowed us to explore centriole elimination kinetics along the germline during oogenesis (Figs. 1A and EV1A). Consistent with previous studies (Kim and Roy, 2006; Mikeladze-Dvali et al, 2012; Pierron et al, 2023), the fluorescence intensity of GFP-tagged centriolar proteins, SAS-5, SAS-6, SAS-4, and SAS-7, begins to decay in diplotene

(Figs. 1B and EV1B). Additionally, quantification of fluorescence intensity revealed that centriolar proteins persist during early meiotic prophase (Fig. 1C,D), suggesting that centrosomes remain intact during these stages and are eliminated only after pachytene exit. The progressive loss of foci enriched with centriolar and PCM components during late meiotic prophase aligns with a gradual process of centriole elimination (Pierron et al, 2023). Therefore, we use changes in the intensity of centrosome protein foci as indicators of the centrosome elimination process, although the actual disassembly of centrioles and centrosomes may occur only after reaching a specific threshold.

## Cartwheel proteins have dynamic properties in vivo

The centrosome is a complex organelle composed of numerous components, and dynamic weak interactions have been proposed to contribute to the maturation of the PCM during the cell cycle (Blanco-Ameijeiras et al, 2022; Mittasch et al, 2020; Woodruff, 2021). To examine whether weak interactions also contribute to centriole formation, we treated worm gonads with 1,6-hexanediol, a compound known to disrupt weak hydrophobic interactions (Kroschwald et al, 2017). This reagent has been shown to effectively disassemble cellular condensates, such as nuclear pore complex (Patel et al, 2007) and stress granules (Molliex et al, 2015).

We assessed the effects of 1,6-hexanediol on centriolar protein localization in vivo and found that a 5-min exposure to 10% 1,6-hexanediol effectively disrupted GFP::SAS-5 foci in the germline (Fig. 1E,F). Notably, this disruption was reversible, as the fluorescence intensity of GFP::SAS-5 foci was significantly restored within 10 min after 1,6-hexanediol washout (Fig. 1E,F). Similarly, SAS-6::GFP foci dissolved following a 5-min treatment with 10% 1,6-hexanediol and reassembled within 10 min after the compound was removed (Fig. 1E,F). Moreover, endogenous SAS-6 tagged with Flag exhibited comparable sensitivity to 1,6-hexanediol treatment and also reformed after the compound was washed out (Fig. EV1C,D). In addition, the fluorescence intensity of GFP::SAS-7, a PCM component, decreased upon 1,6-hexanediol treatment and was notably restored within 10 min after washout (Fig. EV1E,F). These observations suggest that multiple centrosome components may have dynamic properties.

The ability of cartwheel proteins to reassemble implies that certain substructures of the centriole may remain stable and intact during 1,6-hexanediol exposure, providing the structural framework for the reassembly of other dynamic components after hexanediol washout. Consistent with this, we observed that the microtubule core structure of the centriole remained unchanged following 1,6-hexanediol treatment, as visualized by ultrastructure expansion and stimulated emission depletion (U-Ex-STED) microscopy (Fig. 1G,H).

To further examine the mobility of centriolar proteins, we performed fluorescence recovery after photobleaching (FRAP) analysis in the germline of *gfp::sas-5* and *sas-6::gfp* worms. The fluorescence intensity of both GFP::SAS-5 and SAS-6::GFP was recovered to about 50% of their initial levels within less than a minute (Fig. 1I–K). In line with previous reports (Le Guennec et al, 2020; Woglar et al, 2022; Yoshiba et al, 2019), our expansion microscopy analysis suggests that SAS-6 does not have a major pool within the PCM (Fig. EV1G). Thus, we propose that the observed dynamics of these proteins reflect the behavior of cartwheel proteins within the centriole. Taken together, these findings revealed that centriolar cartwheel components exhibit dynamic

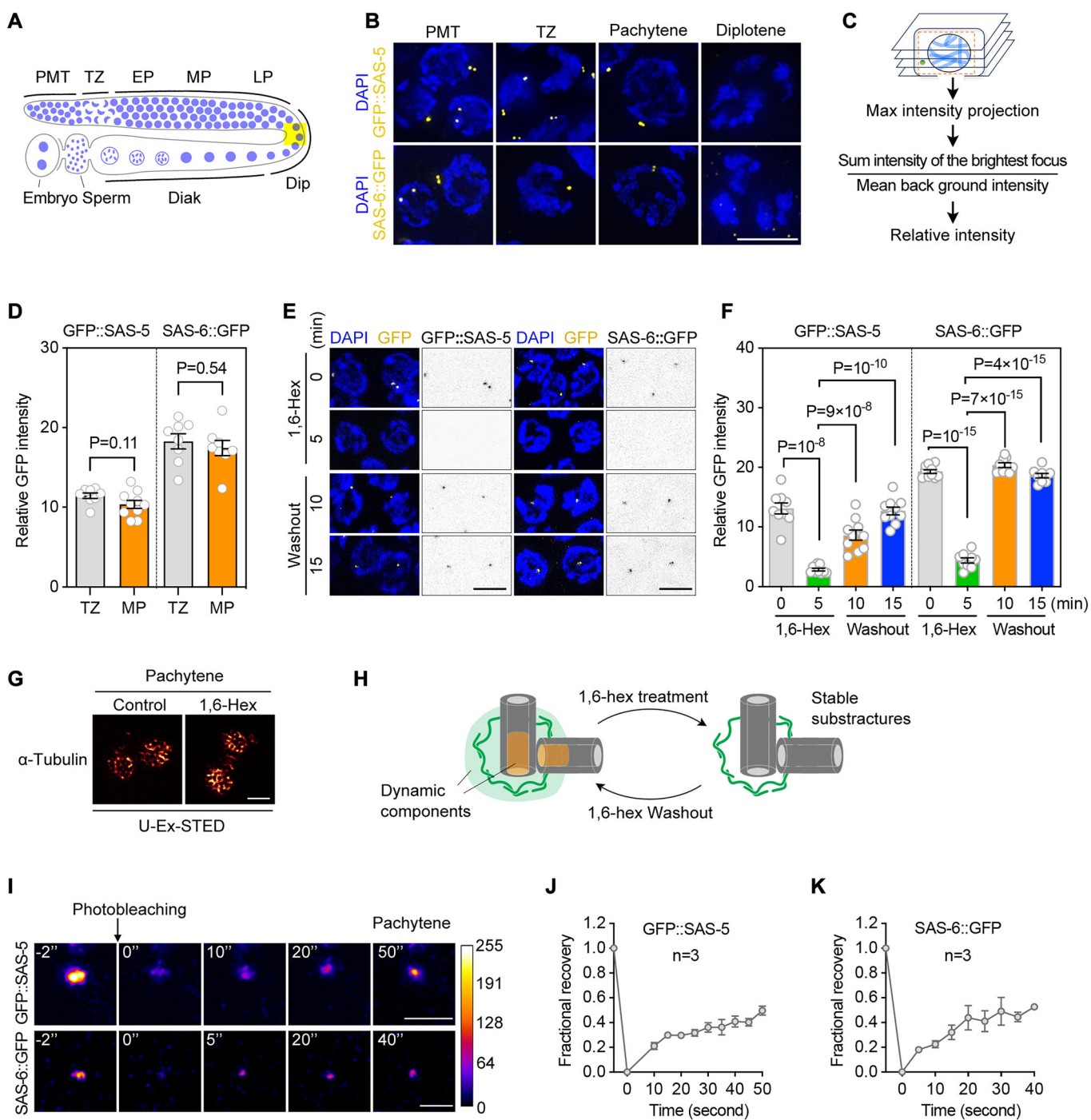

properties in vivo, despite their highly ordered structure, and that weak interactions likely play a role in their assembly.

## SAS-6 undergoes phase separation in vitro

To investigate the dynamic properties of SAS-6, we conducted a series of in vitro experiments. We first expressed and purified recombinant SAS-6 protein tagged with mCherry or GFP from *Escherichia coli* (*E. coli*). Fluorescence microscopy analysis revealed that SAS-6-mCherry formed bright droplets in phase separation buffer, whereas mCherry alone remained evenly diffused (Figs. 2A,B and EV2A). These SAS-6 droplets increased in size over time (Fig. 2D). Similar results were observed with GFP-tagged SAS-6 (Figs. 2A,C,D and EV2B). Real-time imaging further demonstrated fusion events between SAS-6 droplets, indicative of liquid-like properties (Fig. 2E,F). Consistent with the behavior of SAS-6 foci in germ cells, SAS-6 droplets formed in vitro were also sensitive to 1,6-hexanediol treatment, as shown by real-time imaging (Figs. 2G and EV2C).

To assess the mobility of SAS-6 molecules within droplets, we performed FRAP analysis. In SAS-6-mCherry droplets, fluorescence

Figure 1. Centrosome elimination and dynamic properties of centriole proteins during meiotic prophase in *C. elegans*.

(A) Schematic representation of meiotic prophase substages in hermaphrodite germline. PMT, premeiotic tip; TZ, transition zone; EP, early pachytene; MP, middle pachytene; LP, late pachytene; Dip, diplotene; Diak, diakinesis. The region highlighted in yellow indicates where centrosome elimination occurs in wild-type worms. (B) Fluorescence images of germ cells expressing GFP::SAS-5 or SAS-6::GFP at the indicated meiotic stages. Chromatin was stained with DAPI (blue). Scale bar, 5 μm. (C) Quantitative schematic of relative fluorescence intensity of centriolar proteins. (D) Quantification of GFP signal intensity in germ cells at the transition zone (TZ) or mid-pachytene (MP) stages. Each dot represents the mean relative fluorescence intensity of centriolar proteins at the indicated stages within a gonad. Data are shown as mean ± SEM. At least eight gonads were measured. *P* values were calculated using two-tailed unpaired t-tests. (E, F) Representative images (E) and quantification (F) of GFP::SAS-5 or SAS-6::GFP fluorescence intensity before and after treatment of *gfp::sas-5* or *sas-6:: gfp* worms with 10% 1,6-hexanediol (1,6-Hex) for 5 min, followed by washout for 10 min or 15 min. Chromatin was stained with DAPI (blue). Scale bar, 5 μm. Quantification data are shown as mean ± SEM. *P* values were calculated using two-tailed unpaired t-tests. Each dot represents data from a single gonad; at least nine gonads were measured per condition. (G) U-Ex-STED analysis of pachytene centrioles in *sas-6::flag* worms, treated with or without 1,6-hexanediol and stained for α-tubulin. Centrioles from six cells, each likely originating from a different gonad, were reconstructed per condition, with consistent phenotypes observed. Scale bar, 500 nm. (H) A hypothetical model depicting how 1,6-Hex treatment disrupts the dynamic components of the centrosome. (I–K) FRAP analysis of GFP::SAS-5 or SAS-6::GFP foci in *gfp::sas-5* or *sas-6::gfp* gonad (I). Quantitative FRAP data are shown as mean ± SEM (n = 3) in (J, K). Scale bar, 1 μm. The color bar represents fluorescence intensity. Source data are available online for this figure.

intensity fully recovered within 20 s after bleaching (Fig. 2H,I), while SAS-6-GFP droplets showed complete fluorescence recovery within 60 s (Fig. 2H,J). These observations demonstrate that SAS-6 undergoes phase separation in vitro and that the formed droplets exhibit liquid-like properties. A slightly slower recovery of GFP-tagged SAS-6 may be due to the weak dimerization property of GFP, which increases interaction valency and has been previously shown to modestly affect droplet properties (Wang et al, 2024).

Next, we aimed to identify the essential regions driving SAS-6 phase separation. Based on the predicted secondary structure of SAS-6, we generated four constructs: full-length SAS-6, the N-terminal head domain (1–170 aa), the central coiled-coil domain (171–410 aa), and the C-terminal unstructured region (401–492 aa) (Fig. 2K). The corresponding recombinant proteins were purified from *E. coli* (Fig. 2L). In vitro phase separation assays revealed that the coiled-coil domain and the C-terminal region were capable of phase separation, while the N-terminal domain did not exhibit this behavior (Figs. 2M and EV2D,E).

## SAS-6 forms droplets in cells

To further verify the properties of the cartwheel protein SAS-6, we examined its behavior when overexpressed in mammalian HEK293T cells. Both GFP- and mCherry-tagged *C. elegans* SAS-6 formed spherical droplets in the cytoplasm (Fig. 3A,B). However, SAS-6 droplets did not colocalize with the endogenous centrosome (Fig. EV3A). Fusion events between SAS-6 droplets were observed (Fig. 3C), and FRAP analysis within the droplets (Fig. 3D–G) further indicated that the droplets exhibit liquid-like properties. Additionally, both SAS-6-mCherry and SAS-6-GFP droplets rapidly dissolved upon exposure to 1,6-hexanediol and reassembled within 90 s after its removal (Fig. 3H), further supporting the phase separation properties of SAS-6 in cells.

To explore the molecular organization within the droplets, we analyzed HEK293T cells expressing SAS-6-GFP using transmission electron microscopy (TEM). Cartwheel ring-like structures were only occasionally observed within the spherical assemblies (Fig. EV3B), suggesting that the highly ordered cartwheel structure found in centrioles is not maintained in SAS-6 droplets. This is likely due to the absence of the outer interacting interface present in the centriole. Nevertheless, the internal dynamics of the droplets imply that multivalent weak interactions drive the coacervation of SAS-6 proteins, which, based on previous findings (Guichard et al, 2017; Hilbert et al, 2013; Klena et al, 2020; Nazarov et al, 2020), may exist as dimers or multimers within the droplets.

Interestingly, live-cell imaging revealed that SAS-6 droplets exhibit dynamic changes during the cell cycle: they disappeared at prometaphase and reformed following cytokinesis (Fig. 3I). This cell cycle-dependent behavior was further confirmed in fixed cells (Appendix Fig. S1A,B). These observations led us to hypothesize that SAS-6 may be subject to cell cycle-dependent regulation, which modulates its molecular properties or functions both in vivo and in cultured cells.

## SAS-6 is phosphorylated in cells and in *C. elegans*

Phosphorylation mediated by cell cycle kinases may play a role in regulating SAS-6 droplet formation. To identify potential phosphorylation sites, we immunoprecipitated SAS-6-GFP from HEK293T cells arrested at the prometaphase and performed mass spectrometry analysis. This revealed five phosphorylated serine/threonine residues (Ser408, Thr426, Thr437, Thr464, and Thr487) located within the C-terminal region of SAS-6 (Fig. 4A). Notably, these residues correspond to S/T-P motifs, which are primary recognition sites for cyclin-dependent kinase 1 (CDK-1), suggesting that SAS-6 is a potential substrate of CDK-1.

To determine whether SAS-6 is phosphorylated in *C. elegans*, we immunoprecipitated SAS-6::FLAG from lysates of synchronized young adult worms generated by endogenous tagging (Fig. EV4A). Mass spectrometry analysis identified a phosphorylation site at Ser408 (Figs. 4A and EV4B), which overlaps with the phosphorylation sites identified in HEK293T cells. Due to limited peptide coverage in the mass spectrometry analysis, the other phosphorylation sites identified in cells could not be confirmed in vivo.

To further validate these findings, we generated a phosphorylation-specific antibody targeting SAS-6 phosphorylated at Ser408 (pSer408) by immunizing rabbits. This antibody detected phosphorylation of SAS-6::FLAG immunoprecipitated from worm lysates but did not recognize SAS-6(5A)::FLAG immunoprecipitated from lysates of mutant worms carrying phospho-dead mutations, including Ser408-Ala (Figs. 4B and EV4A). These results confirmed that SAS-6 is phosphorylated in vivo at Ser408.

## CDK-1 directly phosphorylates SAS-6 in vitro

Consistent with the hypothesis that SAS-6 is a substrate of CDK-1, we detected an interaction between SAS-6 and CDK-1 using two approaches: GST pulldown assays with bacterially purified GST-SAS-6-mCherry and lysates from cells expressing Myc-CDK-1 (Fig. 4C), and co-immunoprecipitation of Myc-CDK-1 and SAS-6-GFP co-expressed in HEK293T cells (Fig. 4D).

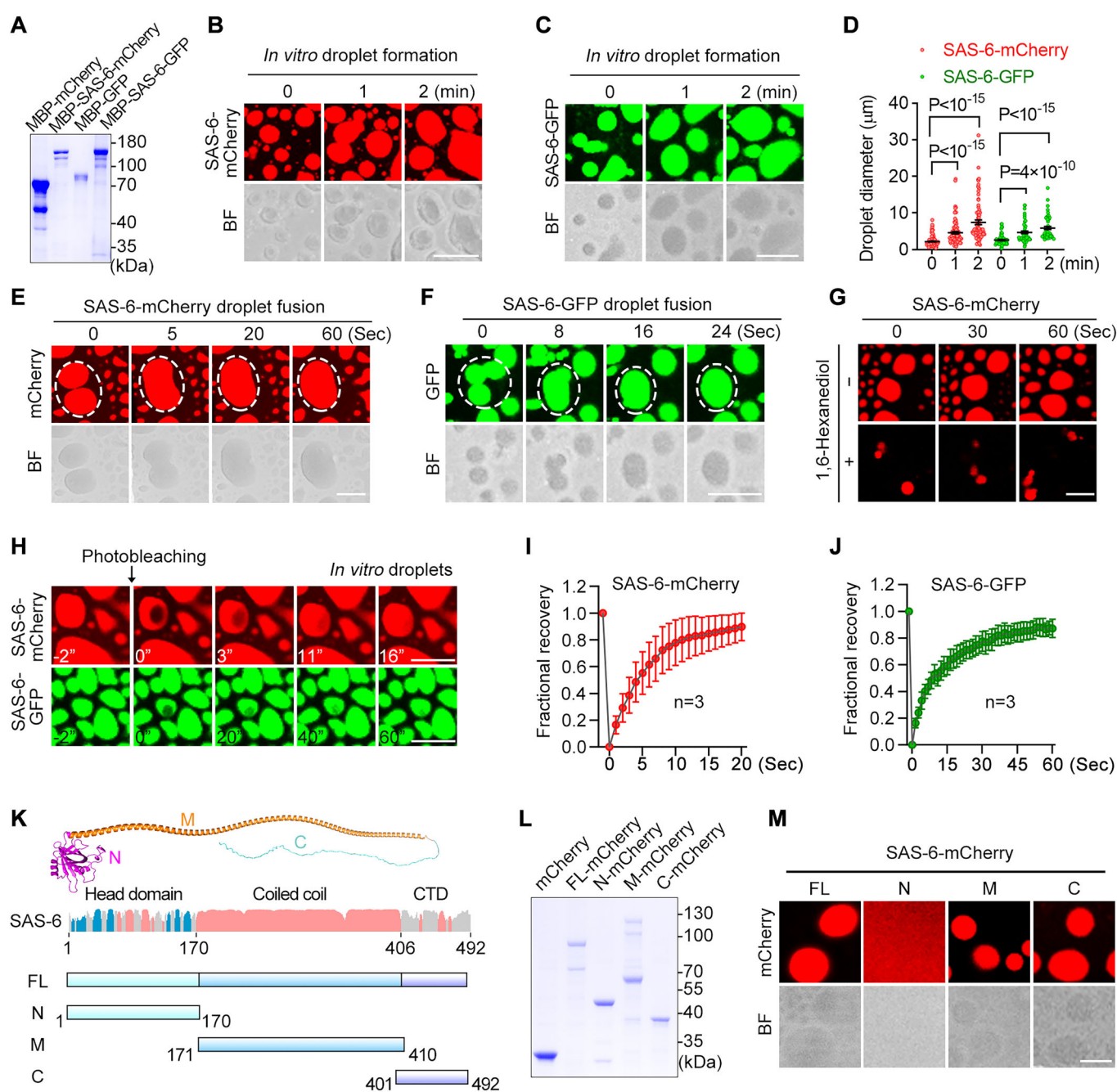

**Figure 2. Purified SAS-6 protein forms liquid droplets in vitro.**

(A) SDS-PAGE and Coomassie brilliant blue staining of purified MBP-mCherry, MBP-SAS-6-mCherry, MBP-GFP, and MBP-SAS-6-GFP proteins. (B, C) Bright-field (BF) and fluorescence images of droplets formed by purified MBP-SAS-6-mCherry (B) and MBP-SAS-6-GFP (C) proteins. Scale bar, 10 μm. (D) Quantification of MBP-SAS-6-mCherry and MBP-SAS-6-GFP droplet sizes at the indicated incubation time. Data are shown as mean ± SEM. P values were determined using two-tailed unpaired t-tests. 200 droplets were measured for each time point. (E, F) BF and fluorescence time-lapse imaging of MBP-SAS-6-mCherry (E) and MBP-SAS-6-GFP (F) droplet fusion. On average, MBP-SAS-6-mCherry droplets underwent ~30 fusion events and MBP-SAS-6-GFP droplets ~20 events during a 5-min video recording (n = 3 independent experiments). Scale bar, 10 μm. (G) Fluorescence images of SAS-6-mCherry droplets with or without 10% 1,6-hexanediol treatment. Scale bar, 5 μm. (H) FRAP analysis of MBP-SAS-6-mCherry (Top) and MBP-SAS-6-GFP (Bottom) droplets. Scale bar, 10 μm. (I, J) Quantification of FRAP analysis of the droplets depicted in (H). Data represent mean ± SEM (n = 3). (K) Top: Structure prediction of SAS-6 protein by AlphaFold3. The SAS-6 protein consists of an N-terminal head domain, a coiled-coil domain and a C-terminal disordered region. Middle: Secondary protein structure of SAS-6 protein. Bottom: Schematic diagrams showing the construction of SAS-6 truncated mutants. (L) SDS-PAGE and CBB staining of purified full-length (FL) or truncations of SAS-6 proteins. (M) BF and fluorescence images of droplets formed by purified FL, N-, M-, or C-domain of SAS-6-mCherry protein. Scale bar, 10 μm. Source data are available online for this figure.

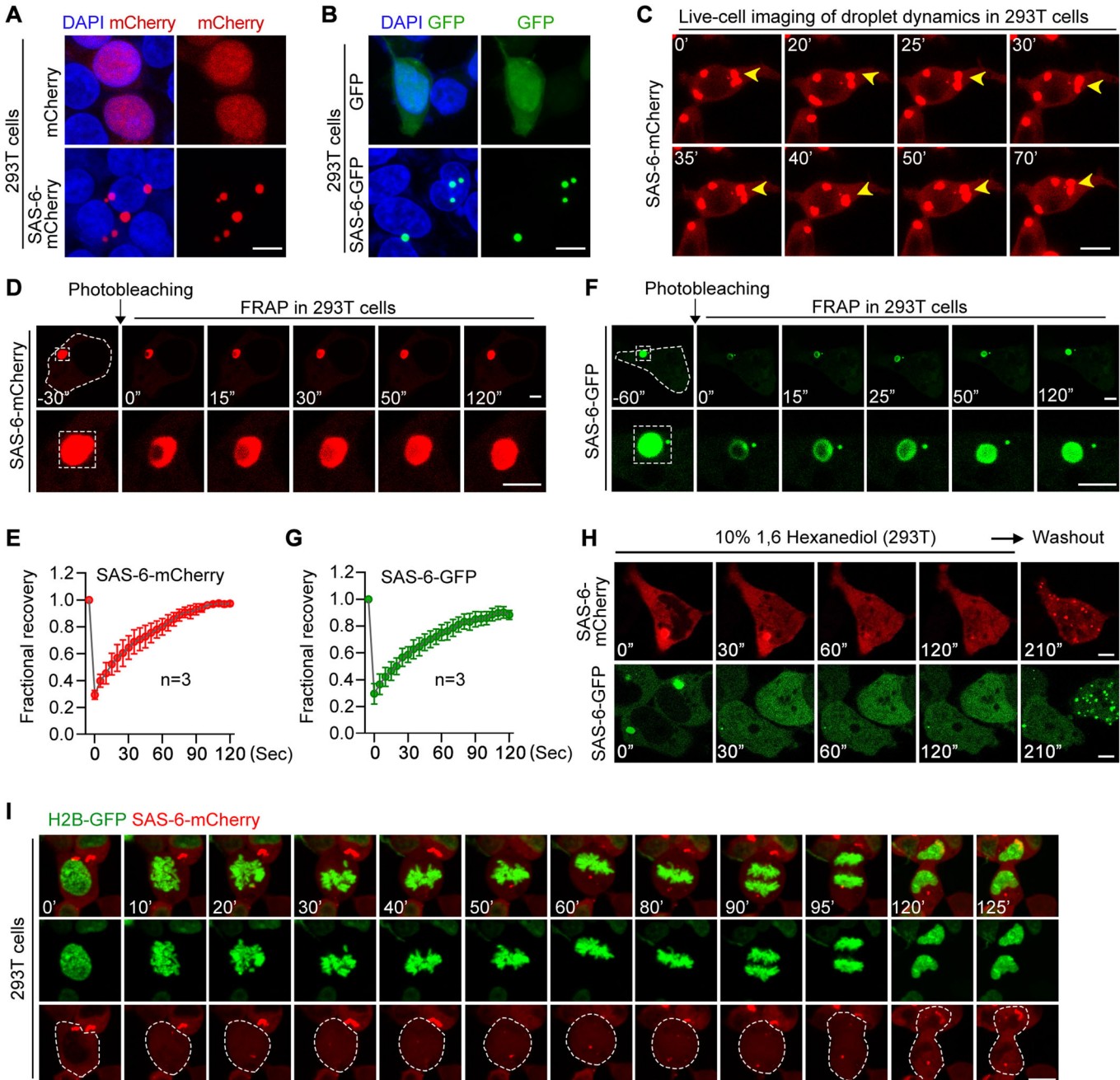

**Figure 3. SAS-6 forms droplets in a cell cycle-dependent manner in HEK293T cells.**

(A, B) Fluorescence images of mCherry-tagged SAS-6 (A) or GFP-tagged SAS-6 (B) overexpressed in HEK293T cells. Chromatin was stained with DAPI (blue). Scale bar, 10 μm. (C) Live-cell imaging of SAS-6-mCherry in HEK293T cells. Arrowheads indicate the fusion of SAS-6 condensates. Fusion was observed in 20 of 28 transfected cells across three independent experiments. Scale bar, 10 μm. (D, E) FRAP time-lapse imaging (D) and quantification of recovery (E) for SAS-6-mCherry condensates in HEK293T cells. Quantification data represent mean ± SEM of 3 biological replicates. Scale bar, 4 μm. (F, G) FRAP time-lapse imaging (F) and quantification of recovery (G) for SAS-6-GFP condensates in HEK293T cells. Quantification data represent mean ± SEM of 3 biological replicates. Scale bar, 4 μm. (H) Live-cell imaging of mCherry- or GFP-tagged SAS-6 condensates under 10% 1,6-hexanediol treatment and washout. Consistent phenotypes were observed in 20 SAS-6-mCherry and 15 SAS-6-GFP condensate-containing cells across three independent experiments. Scale bar, 10 μm. (I) Live-cell tracking of SAS-6-mCherry condensate formation during mitotic cell cycle in HEK293T cells co-expressing H2B-GFP. Phenotype consistent in 4/4 cells imaged. Scale bar, 10 μm. Source data are available online for this figure.

To determine whether CDK-1 directly phosphorylates SAS-6, we performed in vitro kinase assays using HA-tagged CDK-1 immunoprecipitated from HEK293T cells. Phosphorylation of recombinant SAS-6 by CDK-1-HA was detected using a thiophosphate-ester antibody (Fig. 4E). Additionally, phosphorylation of SAS-6 by CDK-1 was confirmed using mitotic (phospho-) protein monoclonal-2 (MPM-2), an antibody that specifically recognizes phosphorylated S/T-P motifs (Davis et al, 1983; Lau

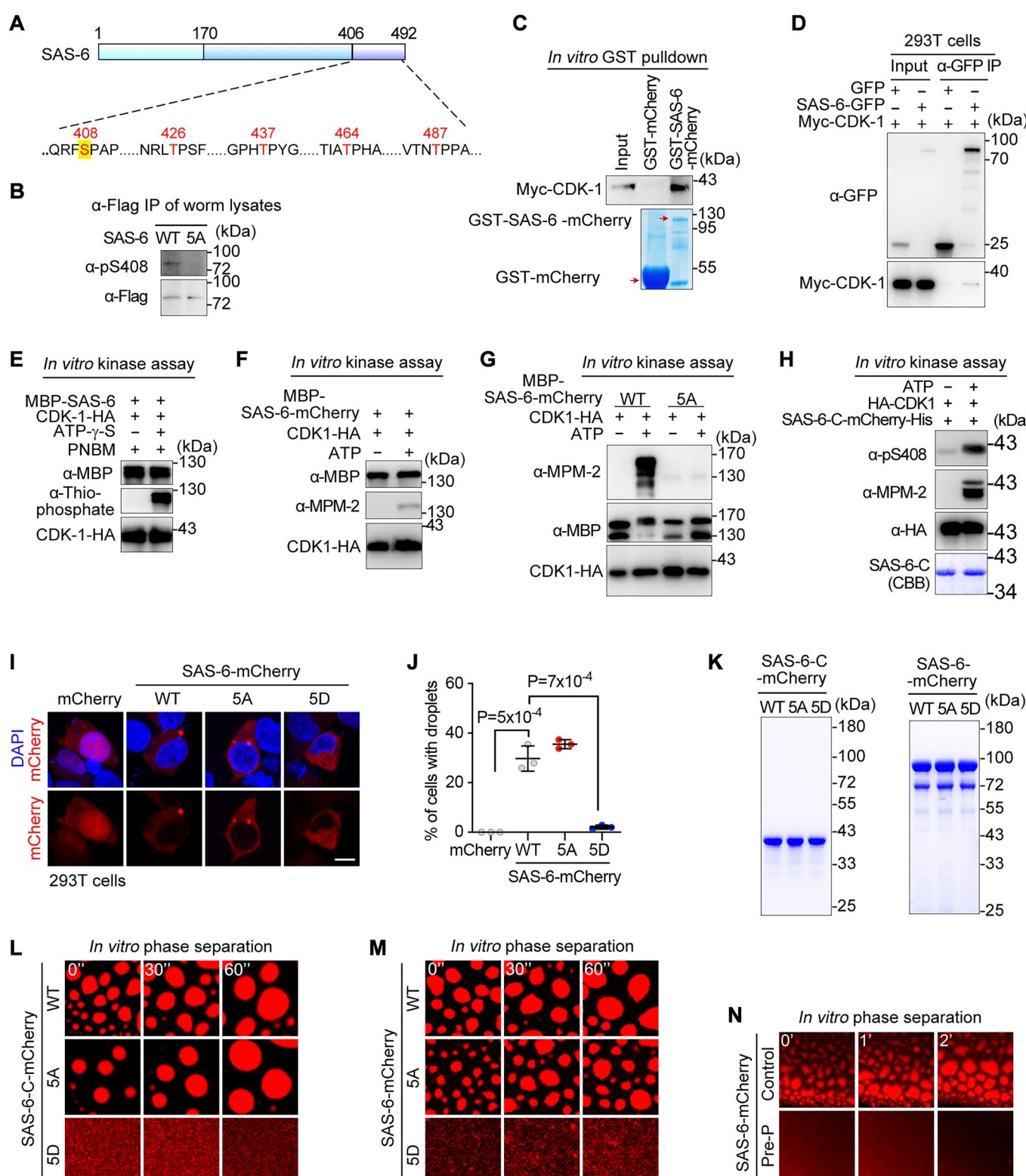

et al, 2021) (Fig. 4F). Importantly, phosphorylation of recombinant
SAS-6 by CDK-1 was abolished in the presence of RO3306, a CDK-1 inhibitor (Appendix Fig. S2A,B).

To verify whether the five phosphorylated serine/threonine residues identified in cells are the primary targets of CDK-1, we mutated these residues to alanines and purified the recombinant SAS-6(5 A) mutant. In vitro kinase assays demonstrated that wild-type SAS-6, but not the SAS-6(5A) mutant, was phosphorylated by CDK-1 (Fig. 4G; Appendix Fig. S2C), indicating that these sites are the main phosphorylation targets of CDK-1. Furthermore, phosphorylation at

**Figure 4. CDK-1-mediated phosphorylation inhibits SAS-6 phase separation in vitro and in HEK293T cells.**

(A) Schematic diagram depicting SAS-6 phosphorylation sites identified by mass spec analysis of immunoprecipitations of SAS-6 from mitotic cells (five sites) and worm lysates (Ser408). (B) WB examination of Ser408 phosphorylation of SAS-6 immunoprecipitated from lysates of *sas-6::Flag* and *sas-6(5A)::Flag* worms. (C) GST pulldown and WB analysis revealing the interaction between purified GST-SAS-6-mCherry and Myc-CDK-1 expressed in HEK293T cells. WB was performed with α-Myc antibody and Coomassie brilliant blue staining was performed to visualize GST proteins. The red arrows indicate the purified target proteins. (D) α-GFP IP and WB analysis showing the interaction between SAS-6-GFP and Myc-tagged worm CDK-1 in HEK293T cells. (E) In vitro kinase assay performed with worm CDK-1 immunoprecipitated from HEK293T cell lysates and purified MBP-SAS-6. Protein phosphorylation was detected by WB with α-thiophosphate ester antibody. PNBM, p-nitrobenzyl mesylate. (F) In vitro kinase assay using human CDK1 immunoprecipitated from HEK293T cells and purified MBP-SAS-6-mCherry as substrate. Protein phosphorylation was detected by WB with α-MPM-2 antibody. (G) In vitro kinase assay performed with human CDK1 and purified MBP-SAS-6-mCherry or MBP-SAS-6(5 A)-mCherry mutant as substrates. Protein phosphorylation was detected by WB with α-MPM-2. (H) In vitro kinase assay using human CDK1 and purified SAS-6-C-mCherry. Protein phosphorylation was detected by WB with α-MPM-2 and α-pS408 antibodies. (I, J) Fluorescence images (I) and quantification (J) of transfected cells with droplets formed by wild-type or mutants of SAS-6 proteins in HEK293T cells. Data are mean ± SEM from three biological replicates per condition; each dot represents one replicate. P values were determined using two-tailed unpaired t-tests. Scale bar, 10 μm. (K) SDS-PAGE analysis of purified SAS-6 C-terminus or full-length proteins containing wild-type sequence or 5 A/5D point mutations. (L) Real-time images showing the phase separation of wild-type or mutant SAS-6-C proteins in vitro. Consistent phenotypes were observed for wild-type and mutant SAS-6-C proteins across three independent experiments. Scale bar, 10 μm. (M) Real-time images showing the phase separation of wild-type or mutant SAS-6-mCherry proteins in vitro. Consistent phenotypes were observed for wild-type and mutant SAS-6 proteins across three independent experiments. Scale bar, 10 μm. (N) Phase separation analysis of mCherry tagged SAS-6 protein with (Pre-P) or without (Control) pre-phosphorylation by CDK1. Consistent phenotypes were observed in two independent replicates. Scale bar, 10 μm. Source data are available online for this figure.

Ser408 in the C-terminal region of SAS-6 was confirmed by Western blotting using the SAS-6 pSer408 antibody (Fig. 4H).

## Phosphorylation inhibits phase separation of SAS-6

To determine whether phosphorylation regulates SAS-6 phase separation, we generated constructs expressing SAS-6(5A)-mCherry and SAS-6(5D)-mCherry mutants, in which the five phosphorylatable residues were mutated to alanines (phospho-deficient) or aspartic acids (phospho-mimetic), respectively. When expressed in HEK293T cells, the phospho-mimetic SAS-6(5D) mutant exhibited significantly reduced droplet formation, while the phospho-deficient SAS-6(5A) mutant retained robust droplet-forming ability (Fig. 4I,J). Consistently, immunofluorescence microscopy revealed that the pSer408 antibody robustly detected overexpressed SAS-6 in mitotic cells but not in cells rest in interphase cells or in cells expressing SAS-6(5A) mutant (Appendix Fig. S1A,B). Furthermore, treatment of mitotic cells with RO3306 led to the formation of SAS-6 droplets and a near-complete loss of the pSer408 immunostaining signal (Appendix Fig. S1C). These results collectively suggest that phosphorylation inhibits SAS-6 phase separation in cells.

Next, we purified wild-type and 5D mutants of full-length SAS-6 and its C-terminal domain (401–492 aa) from bacteria (Fig. 4K). Phase separation assay and real-time imaging revealed that the phospho-mimetic mutants failed to form droplets and remained dispersed (Fig. 4L,M), suggesting that phosphorylation at the C-terminal region inhibits SAS-6 phase separation. This observation also suggests that the coiled-coil domain's phase separation capacity is likely overshadowed in full-length SAS-6, possibly due to steric hindrance. Together, these findings demonstrate that C-terminal phosphorylation suppresses SAS-6 phase separation.

Additionally, in vitro phosphorylation of full-length SAS-6 by CDK-1 abolished its phase separation ability (Fig. 4N), consistent with the hypothesis that CDK-1-mediated phosphorylation disrupts SAS-6 phase separation.

## SAS-6 phosphorylation attenuates interactions between centriolar proteins

The observation that phosphorylation disrupts the phase separation of SAS-6, promotes us to hypothesize that phosphorylation can disrupt the multivalent weak interactions mediated by the C-terminal domains, leading to cartwheel instable in the centriole.

To determine whether the C-terminus mediates SAS-6 intermolecular interactions, we purified GST-SAS-6 with domain deletions (ΔN: Δ1–170; ΔM: Δ171–410; ΔC: Δ401–492) and assessed their ability to interact with the SAS-6 C-terminus (Fig. 5A). Protein structure predictions indicated that these deletions do not disrupt the folding of the remaining domains (Appendix Fig. S3). GST pulldown and Western blot analysis revealed that both the central coiled-coil domain and the C-terminus are required for this interaction, while the N-terminal domain is dispensable (Fig. 5A). Bead recruitment assays confirmed the necessity of the central and C-terminal regions for mediating interactions with the C-terminus (Fig. 5B). Previous models suggest that the N-terminal domain of SAS-6 mediates ring structure formation, while the coiled-coil domain facilitates self-dimerization. The requirement of the central domain for detectable interactions with the C-terminus may reflect the need for multimerization of the C-terminal domain to enable measurable interactions.

We next investigated whether phosphorylation disrupts interactions between full-length SAS-6 molecules. GST pulldown and Western blot analysis revealed that CDK1-phosphorylated SAS-6 lost its ability to interact with full-length SAS-6 coupled to beads (Fig. 5C). Additionally, GST pulldown and bead recruitment assays demonstrated that CDK1-mediated phosphorylation of the SAS-6 C-terminus impaired its interaction with both full-length and SAS-6(ΔN) (Fig. 5D,E). These findings demonstrate that the C-terminus of SAS-6 mediates intermolecular interactions, which are disrupted by CDK-1-mediated phosphorylation (Fig. 5F).

Given that SAS-4 co-localizes with the C-terminal end of SAS-6 in the centriole (Woglar et al, 2022), we tested whether the SAS-6 C-terminus mediates interactions with SAS-4, potentially contributing to centriole stability. GST pulldown assays revealed that FL and SAS-6(ΔN), but not SAS-6(ΔM) or SAS-6(ΔC), interacted with SAS-4-GFP (Fig. 5G). Furthermore, GFP pulldown and Western blot analysis showed that CDK1-mediated phosphorylation of SAS-6 nearly abolished its interaction with SAS-4 (Fig. 5H). Bead recruitment assays confirmed that GFP-tagged SAS-4 expressed in HEK293T cells was recruited by beads coupled with FL or SAS-6(ΔN) but not by SAS-6(ΔM), SAS-6(ΔC), or the SAS-6 C-terminus alone. Moreover, GFP-SAS-4 recruitment was impaired when FL SAS-6 and SAS-6(ΔN) were pre-phosphorylated (Fig. 5I,J). A weak interaction between SAS-4 and the SAS-6 M domain

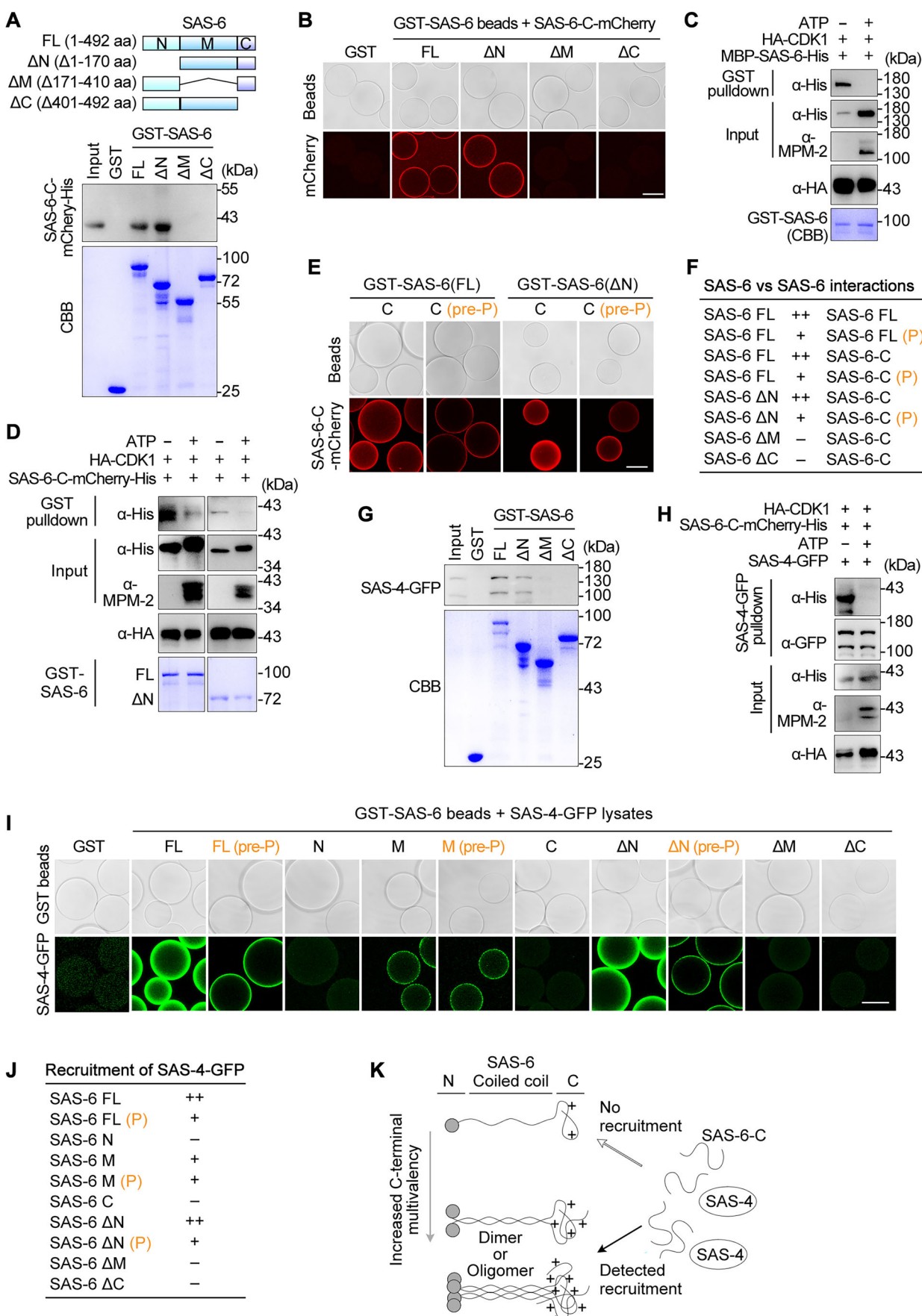

**Figure 5.   SAS-6 phosphorylation impairs the interaction of SAS-6 with its binding partners.**

(A) Top: Schematic representation of SAS-6 truncation constructs. Bottom: GST pulldown examination of interactions between mCherry-His tagged SAS-6 C-terminus and FL or truncated GST-SAS-6. Proteins were detected by immunoblotting with α-His antibody and CBB staining. (B) Bead recruitment assay examining interactions between mCherry-His tagged SAS-6 C-terminus and FL or truncated GST-SAS-6 that coupled to the agarose beads. Scale bar, 50 μm. (C) GST pulldown examination of interactions between GST-SAS-6 and MBP-SAS-6-His, with or without pre-phosphorylation by CDK1. Proteins were detected by immunoblotting with α-His, α-MPM-2, and α-HA antibodies and CBB staining. (D) GST pulldown examination of interactions between GST-SAS-6 full-length (FL) or ΔN mutant and mCherry-His tagged SAS-6 C-terminus, with or without pre-phosphorylation by CDK1. Proteins were detected by immunoblotting with α-His, α-MPM-2, and α-HA antibodies and CBB staining. (E) Bead recruitment assay examining interactions between GST-SAS-6 FL or ΔN mutant and mCherry-His tagged SAS-6 C-terminus with or without pre-phosphorylation by CDK1 (Pre-P). Scale bar, 50 μm. (F) A diagram summarizing the interactions between SAS-6 proteins with or without pre-phosphorylation. The number of plus sign (+) indicates the binding strength, while minus sign (−) denotes no interaction. (G) GST pulldown examination of interactions between HEK293T-expressed SAS-4-GFP and purified GST-SAS-6 FL or truncated forms. (H) GFP pulldown examination of interactions between immunoprecipitated SAS-4-GFP from HEK293T cells and purified SAS-6-C-mCherry-His with or without pre-phosphorylation by CDK1. (I) Bead recruitment assay of HEK293T cell-expressed SAS-4-GFP with bead-coupled FL or truncated SAS-6 (N, M, C, ΔN, ΔM, ΔC), with or without pre-phosphorylation. Consistent phenotypes were observed in ≥10 fields across three independent replicates. Scale bar, 50 μm. (J) A diagram summarizing the interactions between SAS-6 FL or truncated variants and SAS-4 proteins. The number of plus sign (+) indicates the binding strength, while minus sign (−) denotes no interaction. (K) A proposed model illustrating how the SAS-6 C-terminus binds to either itself or the SAS-4 protein. The '+' symbols represent interacting valences, which increase when SAS-6 multimers form and decrease when the C-terminus is phosphorylated (not illustrated). Source data are available online for this figure.

was detected, which remained unaffected by pre-phosphorylation (Fig. 5I). We speculate that this interaction might not occur in FL SAS-6 due to potential steric hindrance from the terminal domains.

Since both the coiled-coil domain and the C-terminus of SAS-6 are required for interactions with SAS-4 and the SAS-6 C-terminus, we propose that SAS-6 multimerization enhances the multivalency of its C-terminus, enabling stable interactions with other proteins (Fig. 5K). Furthermore, colocalization analysis in cells also suggested the impact of phosphorylation in regulating protein interactions. While SAS-4-GFP is co-recruited with wild-type SAS-6 to form droplets, it became diffused in the cytoplasm when co-expressed with the SAS-6(5D) mutant (Appendix Fig. S4A). Similarly, SAS-5 also formed bright droplets with wild-type SAS-6, but not in cells co-expressed with SAS-6(5D) mutant (Appendix Fig. S4B). In summary, these analyses suggest that SAS-6 phosphorylation attenuates C-terminal-mediated interactions, potentially regulating centriole formation and stability.

## Human SASS6 C-terminus exhibits phase separate properties regulated by phosphorylation

Despite limited sequence conservation, the secondary structure of SAS-6 is highly conserved, supporting its role in cartwheel formation across species (Leidel et al, 2005; Nakazawa et al, 2007). To explore the phase separation behavior of human SASS6, we overexpressed SASS6-GFP and its C-terminus-truncated variant in HEK293T cells. Full-length SASS6 formed droplets in cells, a process dependent on its C-terminus (Fig. 6A,B). We also purified mCherry-tagged SASS6 C-terminus from bacteria and confirmed its phase separation capability in vitro (Fig. 6C–F).

In vitro kinase assays revealed that the SASS6 C-terminus is phosphorylated by CDK1 (Fig. 6G), and this phosphorylation abolished its phase separation ability (Fig. 6H). Within the SASS6 C-terminus, two putative phosphorylation sites, Thr495 and Ser510, are present, both located within S/T-P motifs, resembling the phosphorylation sites found in C. elegans SAS-6. Additionally, treatment of HEK293T cells with 1,6-hexanediol disrupted the centrosomal localization of several centrosome proteins, including SASS6 (Fig. 6I). This finding suggests that weak interactions play a role in the formation of centrosomal substructures, such as the centriole cartwheel, aligning with observations in the C. elegans germline.

## SAS-6 promotes centrosome stability in C. elegans germline

To determine whether SAS-6 is essential for centrosome stability, we used the auxin-inducible degradation (AID) system (Zhang et al, 2015). We generated a sas-6$^{AID}$ strain by crossing genome-edited sas-6::degron::flag worms with worms expressing the plant-specific F-box protein TIR1 in the germline. Tagged SAS-6 localized similarly to SAS-6::GFP, forming foci next to the nucleus and supporting normal meiosis (Fig. 7B).

Germ cells take approximately 6 h to transition from leptotene/zygotene (transition zone, TZ) to pachytene, with the pachytene stage lasting ~36 h, during which germ cells advance about one row per hour (Fig. 7A) (Crittenden et al, 2006). A 12-h auxin induction robustly depleted SAS-6 (Fig. 7B), allowing us to assess its role in centrosome maintenance during and after late pachytene, when centrosome duplication is complete. SAS-6 depletion led to an accelerated decay of GFP::SAS-7 foci compared to controls, with their fluorescence intensity significantly reduced before the diplotene stage and becoming nearly undetectable by diplotene (Fig. 7C–F). In addition, GFP::SAS-5 foci became nearly undetectable throughout the germline upon SAS-6 depletion (Fig. 7G–J). The depletion of SAS-6 had a more pronounced effect on SAS-5, likely due to the direct interaction between SAS-5 and the coiled-coil domain of SAS-6 (Qiao et al, 2012), whereas SAS-7 is primarily localized to the PCM (Woglar et al, 2022) (Fig. EV1G). Taken together, these results suggest that SAS-6 degradation leads to premature dispersion of centriolar proteins, highlighting its role in centrosome stability during oogenesis.

## CDK-1 promotes timely centriole disassembly during oogenesis

To investigate whether CDK-1 activity is required for centrosome elimination, we analyzed the expression and activity dynamics of CDK-1. Using strains expressing tagged CDK-1, we observed CDK-1 accumulation in the nuclei of late prophase oocytes, a pattern coinciding with centrosome disappearance (Appendix Fig. S5A,B).

MPM-2 antibody, which recognizes CDK-1-phosphorylated substrates (Davis et al, 1983; Lau et al, 2021; Swygert et al, 2023), revealed two waves of signal during oogenesis: an increase as germ

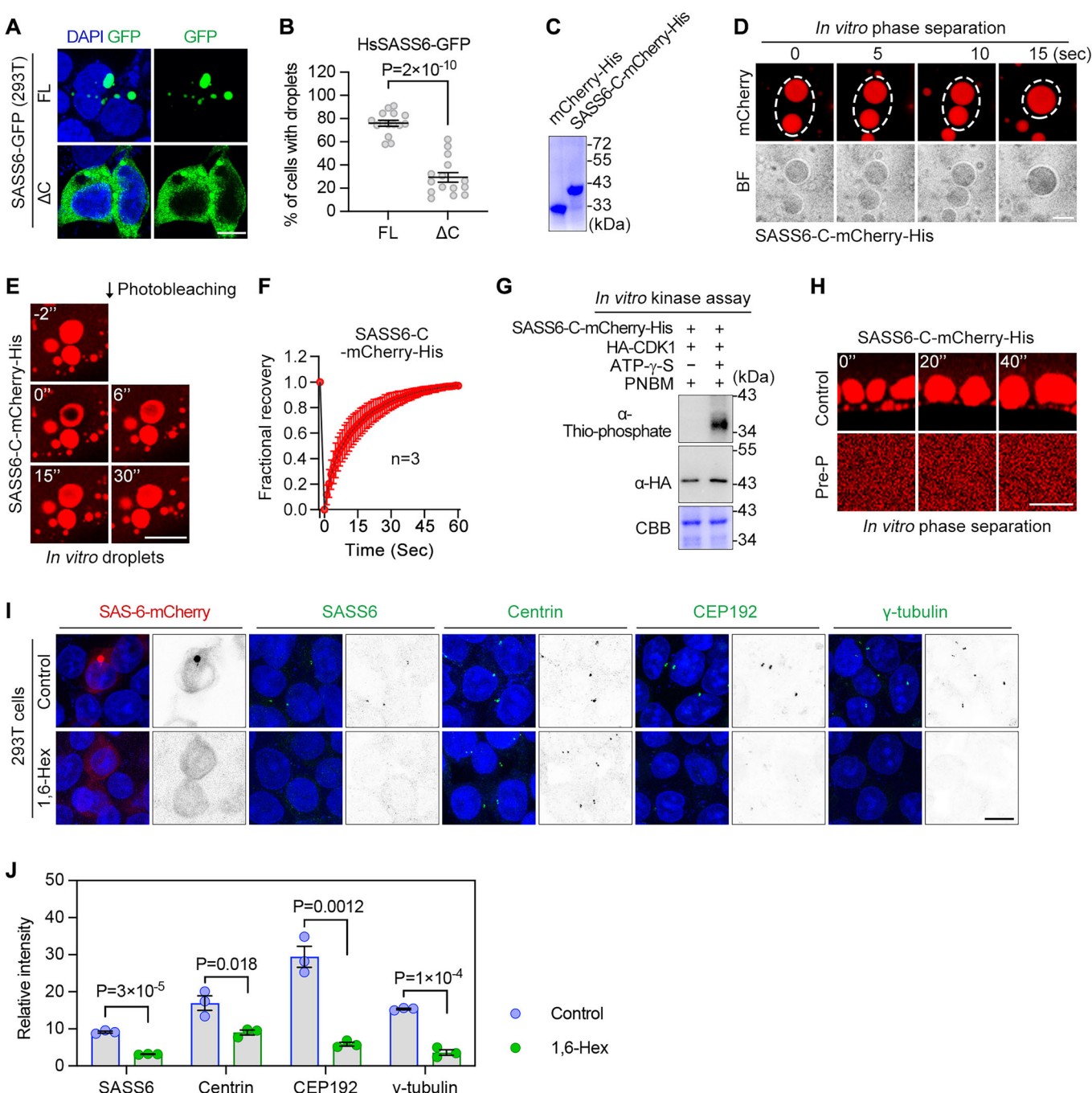

**Figure 6. Human SASS6 C terminus undergoes phase separation in vitro.**

(A, B) Fluorescence images (A) and quantification (B) of transfected cells with droplets formed by FL or C-terminally truncated SASS6 in HEK293T cells. 15 microscopic fields from three biological replicates were scored. Data are shown as mean ± SEM. *P* values were determined using two-tailed paired t-tests. Scale bar, 10 μm. (C) SDS-PAGE analysis of purified mCherry-His and SASS6-C-mCherry-His proteins. (D) Bright-field (BF) and fluorescence images of droplets formed by purified SASS6-C-mCherry-His. Droplet formation was consistently observed in all five videos from three independent replicates. Scale bar, 5 μm. (E, F) FRAP images (E) and quantification of recovery (F) of SASS6-C-mCherry-His droplets. Scale bar, 10 μm. (G) In vitro kinase assay using immunoprecipitated CDK1-HA from HEK293T cells and purified SASS6-C-mCherry-His. Protein phosphorylation was detected by α-thiophosphate ester antibody. (H) Phase separation analysis of mCherry-His tagged SASS6-C with (Pre-P) or without (Control) pre-phosphorylation by CDK1. Consistent phenotypes were observed in two biological replicates. Scale bar, 5 μm. (I, J) Representative images (I) and quantification (J) of endogenous centriolar proteins in HEK293T cells with or without 10% 1,6-hexanediol treatment for 5 min. Cells transfected with worm SAS-6-mCherry were used as control. Quantification data are shown as mean ± SEM. *P* values were determined using two-tailed paired t-tests. *n* = 3 biological replicates. Scale bar, 10 μm. Source data are available online for this figure.

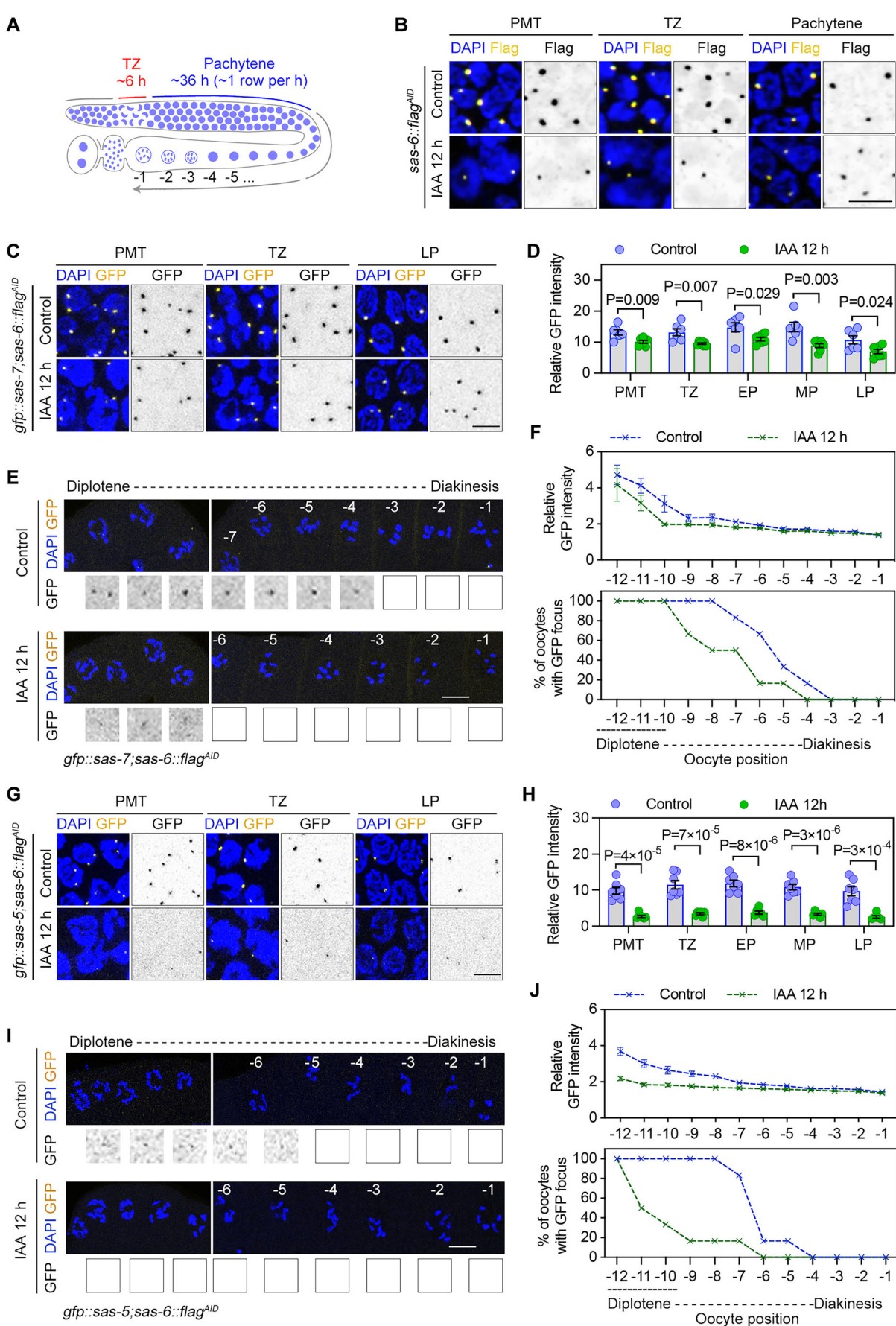

Figure 7. Auxin-induced degradation of SAS-6 promotes centrosome disassembly in *C. elegans* germline.

(A) Schematic illustration of the timing of germ cell development in *C. elegans* hermaphrodite germline. (B) Immunostaining with Flag antibody showing efficient degradation of SAS-6::degron::Flag in the germline using the auxin induced degradation (AID) system. Young adult worms were exposed to 1 mM IAA for 12 h. Nuclear DNA was stained with DAPI (blue). All seven gonads imaged across three biological replicates showed markedly reduced SAS-6::Flag fluorescence after IAA treatment. Scale bar, 10 μm. (C, D) Images (C) and quantification of fluorescence intensity (D) of GFP::SAS-7 foci (yellow) in oocytes of the indicated genotypes with or without SAS-6 depletion by AID. Quantification data are shown as mean ± SEM. *P* values were determined using two-tailed unpaired t-tests. Each dot represents data from a single gonad; at least six gonads were measured per condition. Scale bar, 5 μm. (E, F) Images (E) and quantification of fluorescence intensity (F) of GFP::SAS-7 foci (yellow) before and after SAS-6 degradation by AID, from diplotene to diakinesis. Quantification data are shown as mean ± SEM. Each dot represents data from a single gonad; at least six gonads were measured per condition. Scale bar, 10 μm. (G, H) Images (G) and quantification of fluorescence intensity (H) of GFP::SAS-5 foci (yellow) in oocytes of the indicated genotypes with or without SAS-6 depletion by AID. Quantification data are shown as mean ± SEM. *P* values were determined using two-tailed unpaired t-tests. Each dot represents data from a single gonad; at least six gonads were measured per condition. Scale bar, 5 μm. (I, J) Images (I) and quantification of fluorescence intensity (J) of GFP::SAS-5 foci (yellow) before and after SAS-6 degradation by AID, from diplotene to diakinesis. Quantification data are shown as mean ± SEM. At least six gonads were measured. Scale bar, 10 μm. Source data are available online for this figure.

cells entered late pachytene, a decrease during late diplotene/early diakinesis, and a subsequent increase at the end of meiotic prophase (Fig. 8A). In *cdk-1(ne2257)* mutants (with partially impaired kinase activity) and worms treated with the CDK-1 inhibitor purvalanol A (PA), MPM-2 signal was reduced but not abolished, indicating that CDK-1 may contribute to part of the MPM-2 signal (Fig. 8A,B; Appendix Fig. S5C). These findings suggest that CDK-1 activity is activated during oogenesis.

To assess the role of CDK-1 in centrosome disassembly, we treated *gfp::sas-5* and *gfp::sas-7* worms with CDK-1 inhibitors (RO3306 or PA) for 24 h from the L4 stage. Confocal imaging and quantification showed delayed decay of GFP::SAS-5 and GFP::SAS-7 foci compared to controls (Fig. 8C–F). Together, these results suggest that CDK-1 promotes timely centrosome disassembly during oogenesis.

## Dynamic regulation of SAS-6 phosphorylation is crucial for centrosome assembly and subsequent elimination during oogenesis

Although direct confirmation of the five putative phosphorylation sites at the SAS-6 C-terminus in the germline was not achieved, the in vivo expression pattern of CDK-1, along with its direct interaction with and phosphorylation of SAS-6 in vitro, suggests that these sites are likely phosphorylated in the germline, where they may together perform a regulatory function. To determine whether SAS-6 phosphorylation triggers centrosome instability, we generated phospho-mimetic (*sas-6(5D)*) and phospho-deficient (*sas-6(5A)*) mutants (Fig. EV4A). While *sas-6(5A)* mutants were fully viable, *sas-6(5D)* mutants produced no viable progeny (Fig. 9A). Cytological analysis revealed that SAS-6 foci in *sas-6(5A)* mutants exhibited stronger fluorescence intensity than wild-type, indicating excessive centriole assembly (Fig. 9B,C). In contrast, SAS-6(5D) foci displayed reduced intensity throughout the germline and were nearly undetectable by late pachytene (Figs. 9B,C and EV5A,B), suggesting that sustained phosphorylation may impair centriole assembly and duplication.

To analyze centrosome disassembly kinetics, we examined GFP::SAS-5 and GFP::SAS-7 foci in *sas-6* mutant backgrounds. In *sas-6(5A)* mutants, GFP::SAS-5 and GFP::SAS-7 foci persisted into diakinesis, indicating delayed centrosome elimination, likely due to centriole overduplication in pre-meiotic cells (Fig. 9D–G). However, centrosomes were still eliminated by the end of meiotic prophase, consistent with the lack of significant embryonic lethality in *sas-6(5A)* mutants. Furthermore, this delay in elimination was not due to

changes in the timing of oocyte maturation, as indicated by the unchanged number of oocytes positively stained for phosphorylated MAPK, a marker of matured oocytes (Lee et al, 2007) (Fig. EV5C,D).

In *sas-6(5D)* mutants, cytological analysis suggested mitotic defects in the PMT region, evidenced by enlarged nuclei (Fig. EV5A,B). Germ cells entering meiosis were reduced, leading to smaller gonads and a lack of mature oocytes (Fig. EV5A–D), likely causing infertility. Structured illumination microscopy (SIM) further revealed fewer SAS-6 foci in early meiotic prophase cells, suggesting impaired centriole duplication (Fig. 9H). Nevertheless, each germ cell retained at least one centrosome, allowing analysis of elimination. The intensity of GFP::SAS-5 and GFP::SAS-7 foci was significantly reduced in *sas-6(5D)* mutants, with GFP::SAS-5 foci nearly absent by late pachytene (Fig. 9I–L). Premature centrosome elimination may result from reduced centriolar material assembly in pre-meiotic cells. Since oocytes in *sas-6(5D)* mutants were not activated (Fig. EV5C,D), this premature centrosome elimination cannot be attributed to early oocyte activation. The more pronounced reduction of SAS-5 compared to SAS-7 likely reflects SAS-5's direct dependence on SAS-6 for localization (Qiao et al, 2012), while SAS-7 localizes to the PCM after centrosome duplication (Woglar et al, 2022). This is also consistent with a model in which centrosome elimination can be initiated through the disassembly of the centriole's inner tube (see Discussion).

Overall, our findings suggest that dynamic regulation of SAS-6 phosphorylation is crucial for centrosome assembly and functions in vivo. Phosphorylation may reduce cartwheel stability within the centriole, thereby promoting the centrosome elimination process during oogenesis. Furthermore, this regulatory mechanism may extend beyond oogenesis, as evidenced by the altered elimination timing observed in SAS-6 phospho-mutants in the intestine (Fig. EV5E,F).

## Discussion

Our findings reveal the properties of SAS-6 across in vitro, cellular, and *C. elegans* germline contexts (Fig. 10). Purified SAS-6 undergoes phase separation in vitro and forms droplets in cells when overexpressed. These droplets are large and disorganized, with the unstructured C-terminal region playing a key role in mediating weak interactions that drive phase separation. In contrast, within the centriole in vivo, SAS-6 forms a highly ordered structure, which we propose also relies on multivalent weak interactions mediated by the C-terminal domain. In both contexts,

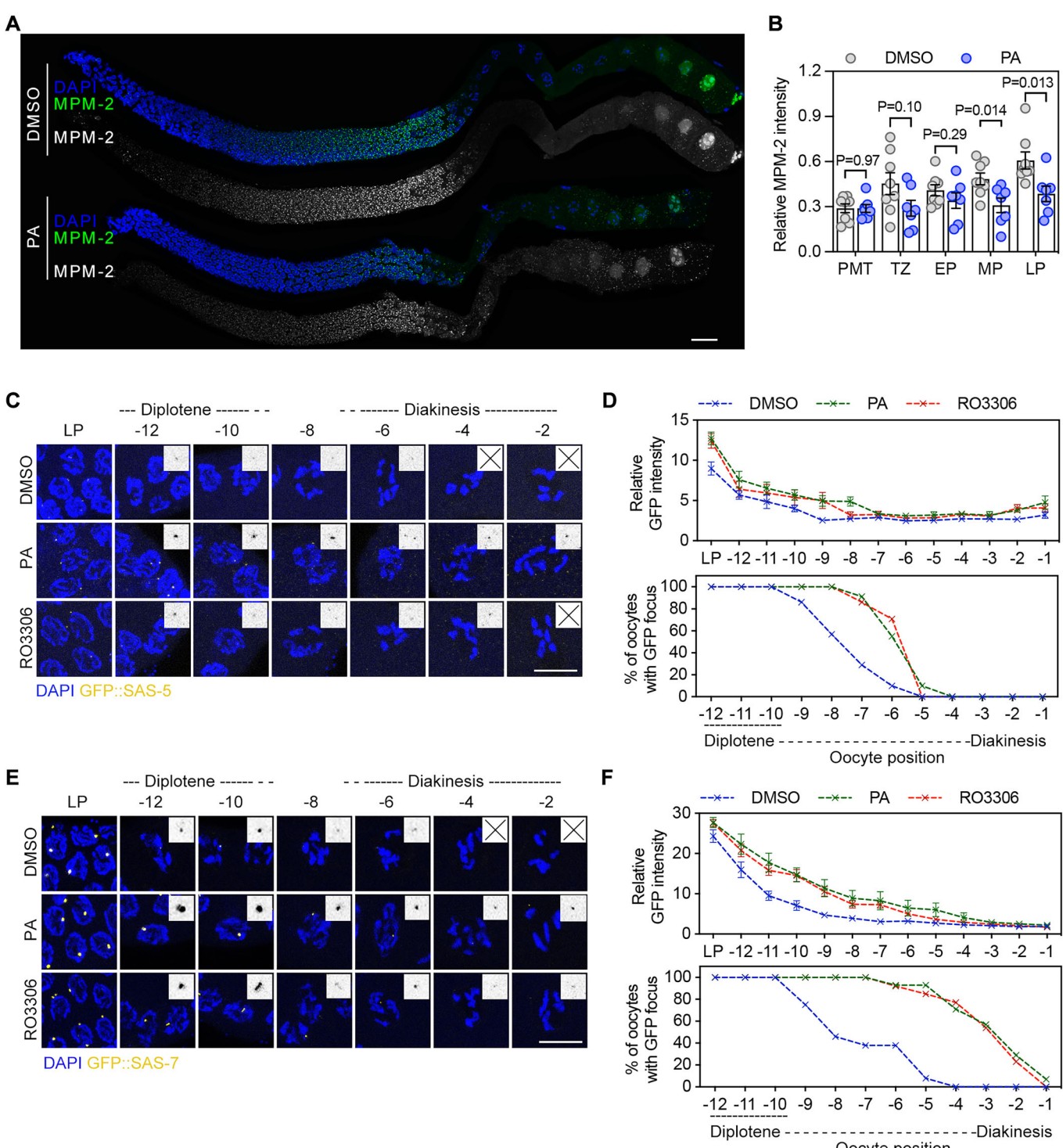

phosphorylation at the C-terminal domain alters the behavior of SAS-6, affecting its phase separation in vitro and cartwheel assembly in vivo. While these processes exhibit distinct forms of organization, they share similar amino-acid-based interaction mechanisms. The functional role of the C-terminal domain provides insights into cartwheel stacking, centrosome elimination during oogenesis, and potentially other biological processes involving centrosome elimination.

## Molecular mechanisms of cartwheel stacking

The cartwheel has an highly ordered structure and is crucial for centriolar symmetry (Vakonakis, 2021). The cartwheel is ~100 nm high, containing stacked ring oligomers of SAS-6 proteins. Recent advances in resolving the cartwheel structure proposed several mechanisms for cartwheel assembly. An electron tomography map of centrioles at higher resolution revealed that pairs of SAS-6 rings

**Figure 8.   Inhibition of CDK-1 delays centrosome disassembly in the *C. elegans* germline.**

(A) MPM-2 Immunostaining in the germline of worms treated with DMSO or PA. DNA was stained with DAPI (blue). Scale bar, 20 μm. (B) Quantification of relative MPM-2 intensity in the germline of worms treated with DMSO or PA. Each point represents the average relative MPM-2 intensity of cells at a specific stage within a gonad. Data are shown as mean ± SEM. *P* values were determined using two-tailed unpaired t-tests. At least seven gonads were measured. (C) GFP::SAS-5 focus maintenance in the germline of worms treated with DMSO or CDK-1 inhibitors. DNA was stained with DAPI (blue). Insects are 2-fold magnified images of the detected foci, and undetected foci are marked with cross symbols. The negative numbers indicate oocyte position as depicted in Fig. 7a. Scale bar, 10 μm. (D) Quantification of GFP::SAS-5 focus fluorescence intensity and maintenance in oocytes of worms treated as in (C). The numbers of gonad arms scored are: DMSO, n = 10; PA, n = 10; RO3306, n = 13. (E) GFP::SAS-7 focus maintenance in the germline of worms treated with DMSO or CDK-1 inhibitors. DNA was stained with DAPI (blue). Scale bar, 10 μm. (F) Quantification of GFP::SAS-7 focus fluorescence intensity and maintenance in oocytes of worms treated as in (E). The numbers of gonad arms scored are: DMSO, n = 15; PA, n = 13; RO3306, n = 14. Source data are available online for this figure.

were stacked ~5 nm apart, thereby creating an average vertical periodicity at the cartwheel hub of ~4.2 nm (Klena et al, 2020). The cryo-electron tomography data of *Trichonympha* and *Ternympha* centrioles suggested that spokes from two successive SAS-6 ring pairs merged first into binary and then tetrameric bundles. Therefore, four SAS-6 rings into a ~17 nm-height periodic unit, matching the vertical spacing of the pinhead structure that connects cartwheels and centriolar microtubules (Nazarov et al, 2020). However, the repeating structural unit of the cartwheel in both *Paramecium* and *Chlamydomonas* is composed of three SAS-6 ring pairs, thus forming a periodicity of 25 nm (Klena et al, 2020). Despite the diversity in the number of SAS-6 rings in each cartwheel stacking unit, the connections between units may be conserved. As revealed by ET analysis, spokes merged into a bundle that rely on the coiled-coil domain. Moreover, the extremities of the spokes are connected together vertically via the C-terminal disordered region and its binding proteins (Guichard et al, 2017). However, how the C-terminal disordered region connects spokes vertically is unknown.

Given the disordered nature of the C-terminal domain of SAS-6 protein across species, we propose that the C-terminus of SAS-6 protein can connect spokes vertically in vivo via multivalent weak interactions to stabilize the centriole during early meiotic prophase (Fig. 10). Moreover, the stacked SAS-6 rings have dynamics that they can exchange with peripheral cartwheel elements. While SAS-6 is phosphorylated by CDK-1 at the C-terminus, intra-molecular interactions are weakened or abolished, causing the staked SAS-6 rings to become unstable and further promoting centriole disassembly during late meiotic prophase. The ability of SAS-6 C-terminus to mediate interactions is also supported by a previous study showing that the C-terminus inhibits SAS-6 cartwheel structure formation in vitro (Guichard et al, 2017). Our findings also revealed an example where multivalent weak interactions support the assembly of a highly ordered structure with the dynamics of its building subunits. Such a mechanism resembles the assembly of the synaptonemal complex (SC) that forms between the paired homologous chromosomes during the meiotic prophase (Zhang et al, 2020). While the SC central region has an ordered molecular organization, its subunits are dynamic, and its assembly is suggested to be driven by multivalent weak interactions between the assembly units in *C. elegans* (Rog et al, 2017; Zhang et al, 2023).

## Regulatory pathways promoting centrosome elimination during oogenesis

Investigations of the centrosome elimination process in starfish and *Drosophila* have shown a shared regulatory pathway achieved by two steps: PCM components are lost first, followed by the

disappearance of centriole components (Borrego-Pinto et al, 2016; Pimenta-Marques et al, 2016; Schoborg and Rusan, 2016), indicating an elimination pathway from outside PCM to inside centriole. Here, we proposed a mechanism that acts on the cartwheel, an inner structure of centriole, to promote centrosome disassembly. We found that CDK-1-mediated phosphorylation of SAS-6 inhibits its phase separation in vitro. Consistently, phospho-mimetic mutation of SAS-6 resulted in the premature elimination of centrosome at the late pachytene stage, despite defects in centrosome duplication. Since centrosome duplication relies on multiple intermolecular interactions, our findings suggest that SAS-6 phosphorylation weakens protein associations, thereby disrupting centrosome integrity. Moreover, our findings are consistent with a recent novel study indicating that the intracentriolar central tube removal initiates centriole elimination (Pierron et al, 2023). SAS-6 has been shown to localize inside the central tube and is required for central tube formation and elongation. Based on this, we propose that SAS-6 phosphorylation, which weakens intermolecular interactions, along with SAS-1 removal, collectively facilitates efficient centrosome elimination. Existing knowledge suggests that the cartwheel structure is present only in the daughter centriole of human cells during mitosis. However, evidence from our SIM analysis and previous work suggests that the cartwheel structure is present in both the mother and daughter centrioles in *C. elegans* and *Drosophila* during meiotic prophase (Woglar et al, 2022), suggesting that SAS-6-mediated regulation of centriole elimination might operate in both centrioles during oogenesis.

However, both phospho-dead mutation of SAS-6 and CDK-1 depletion did not completely prevent centrosome elimination during late meiotic prophase, suggesting that there may be other kinases that can phosphorylate SAS-6 or other post-transcriptional modifications may be involved in this process. In addition, sequence alignment revealed the existence of CDK-1 kinase recognition motifs on SAS-6 proteins in various species, including *Mus musculus, Danio rerio, Patiria miniate*, and *Homo sapiens*. Indeed, SAS-6 has also been identified as a phosphorylation target of CDK1 in human cells, and such regulation is involved in centriole disengagement and licensing during mitosis (Huang et al, 2022). Thus, the CDK-1-mediated SAS-6 phosphorylation may also be a conserved mechanism to promote the process of centrosome elimination in oocytes. Given that CDK1 kinase is active in various cell types, there may be other mechanisms that resist the phosphorylation of SAS-6 by CDK-1, such as the involvement of phosphatases, thereby protecting centrosomes from elimination in other cell types. Collectively, centrosome elimination is a complicated process that may be regulated by multiple pathways, and our identified mechanism could collaborate with the PCM-to-centriole pathway to guarantee the effective elimination of centrosomes.

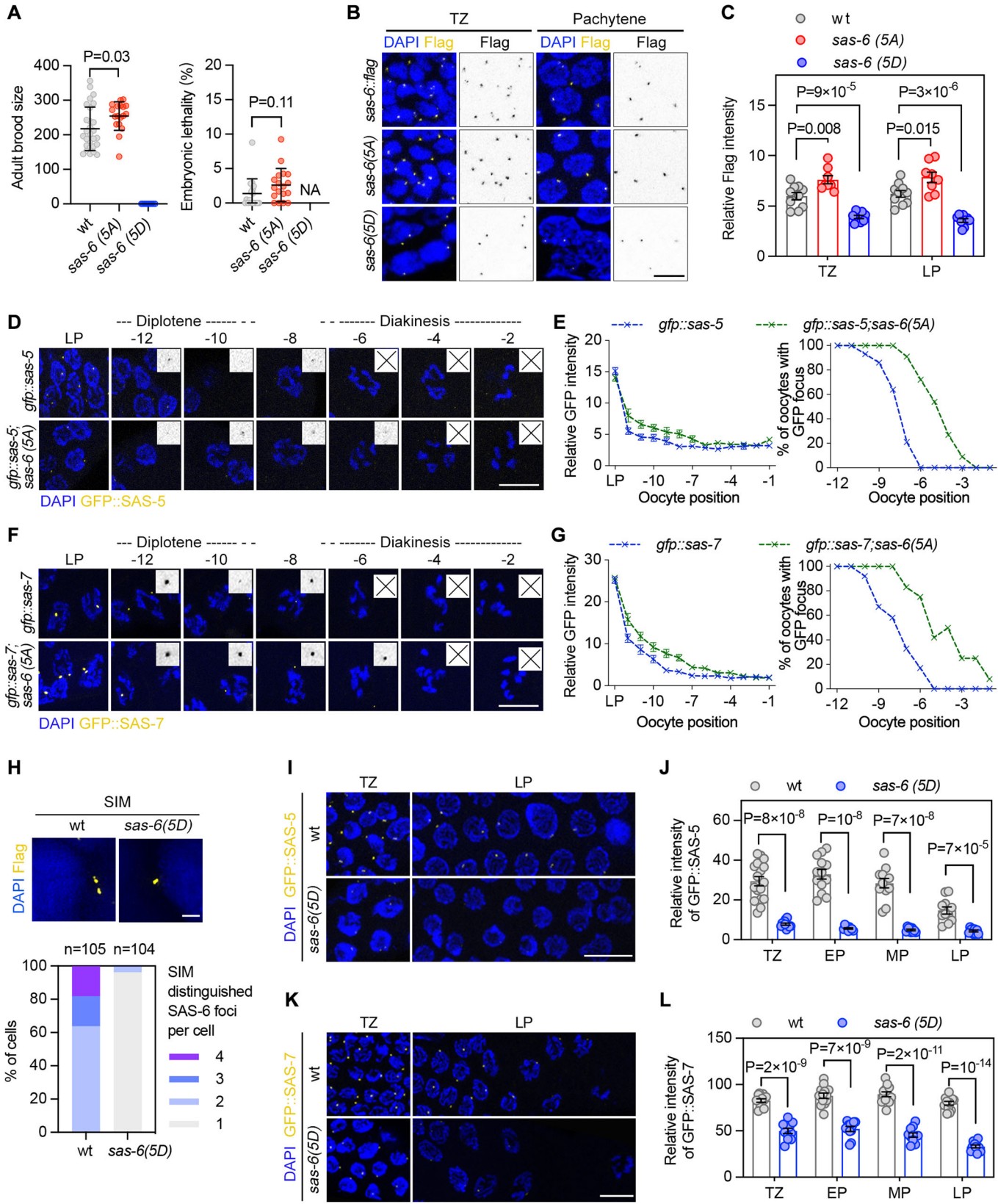

**Figure 9. SAS-6 phosphorylation promotes centrosome disassembly in vivo.**

(A) Plate phenotypes of *sas-6::degron::flag* (wt), *sas-6(5A)::degron::flag*, and *sas-6(5D)::degron::flag* worms. Data are shown as mean ± SEM. *P* values were determined using two-tailed unpaired t-tests. P0 worms analyzed: wt, $n = 28$; 5 A, $n = 18$; 5D, $n = 10$. NA indicates no embryo production. (B) Anti-Flag immunostaining of the endogenously tagged SAS-6 at the transition zone (TZ) and pachytene stage of the gonad from the indicated genotypes. DNA was stained with DAPI (blue). Scale bar, 5 µm. (C) Quantification of Flag focus intensity in gonads immunostained as in (B). Data are shown as mean ± SEM. *P* values were determined using two-tailed unpaired t-tests. Gonad arms scored: wt, $n = 10$; *sas-6(5 A)*, $n = 8$; *sas-6(5D)*, $n = 9$. (D) GFP::SAS-5 focus maintenance in the germline of the indicated genotypes. DNA was stained with DAPI (blue). Scale bar, 10 µm. (E) Quantification of GFP::SAS-5 focus fluorescence intensity and maintenance in oocytes of the indicated genotypes. Data are shown as mean ± SEM. Gonad arms scored: *gfp::sas-5*, $n = 17$; *gfp::sas-5;sas-6(5A)*, $n = 14$. (F) GFP::SAS-7 focus maintenance in the germline of the indicated genotypes. DNA was stained with DAPI (blue). Scale bar, 10 µm. (G) Quantification of GFP::SAS-7 focus fluorescence intensity and maintenance in oocytes of the indicated genotypes. Data are shown as mean ± SEM. The numbers of gonad arms scored are: *gfp::sas-7*, $n = 10$; *gfp::sas-7;sas-6(5 A)*, $n = 10$. (H) Upper panel: SIM images of SAS-6 foci in early pachytene cells of *sas-6::flag* (wt) and *sas-6(5D)::flag* mutant worms immunostained with anti-Flag antibody. Scale bar, 1 µm. Lower panel: Quantification of SAS-6 focus number in early pachytene cells. (I, J) Images (I) and quantification (J) of relative fluorescence intensity of GFP::SAS-5 foci (yellow) in oocytes of indicated genotypes. Quantification data are shown as mean ± SEM. *P* values were determined using two-tailed unpaired t-tests. At least nine gonads were measured. DNA was stained with DAPI (blue). Scale bar, 10 µm. (K, L) Images (K) and quantification (L) of relative fluorescence intensity of GFP::SAS-7 foci (yellow) in oocytes of indicated genotypes. Quantification data are shown as mean ± SEM. *P* values were determined using two-tailed unpaired t-tests. At least eleven gonads were measured. DNA was stained with DAPI (blue). Scale bar, 10 µm. Source data are available online for this figure.

# Methods

### Reagents and tools table

| Reagent/Resource | Reference or Source | Identifier or Catalog Number |
|---|---|---|
| **Experimental models** | | |
| HEK293T cells | ATCC | Cat#CRL-3216 |
| *C. elegans*: N2 wild type (Bristol) | Caenorhabditis Genetics Center | N2 |
| *C. elegans: ieSi64[gld-1p::tir1::mRuby::gld-1 3'UTR + Cbr-unc-119(+)] II; unc-119(ed3) III* | Caenorhabditis Genetics Center | CA1352 |
| *C. elegans: sas-7(or1940[gfp::sas-7]) III; ltIs37[pie-1p::mCherry::his-58 + unc-119(+)] IV* | Caenorhabditis Genetics Center | EU3000 |
| *C. elegans: Psas-6::sas-6::gfp; Ppie-1::mCherry::sas-4* | Li et al, 2017 | GOU899 |
| *C. elegans: unc-119(ed3) III; ddIs36[pie-1p::gfp::sas-5 + unc-119(+)]; sas-6(syb1706[sas-6::degron::flag::ha::flag::ha]) IV; ieSi64[gld-1p::tir1::mRuby::gld-1 3'UTR + Cbr-unc-119(+)] II; unc-119(ed3) III* | This Paper | MGC595 |
| *C. elegans: sas-7(or1940[gfp::sas-7]) III; ltIs37[pie-1p::mCherry::his-58 + unc-119(+)]IV; sas-6(syb1706[sas-6::degron::flag::ha::flag::ha]) IV; ieSi64[gld-1p::tir1::mRuby::gld-1 3'UTR + Cbr-unc-119(+)] II; unc-119(ed3) III* | This Paper | MGC618 |
| *C. elegans: unc-119(ed3) III; ddIs36[pie-1p::gfp::sas-5 + unc-119(+)]; sas-6(syb3988[sas-6(5 A)::degron::flag::ha::flag::ha])IV;ieSi64[gld1p::tir1::mRuby::gld-1 3'UTR + Cbr-unc-119(+)] II; unc-119(ed3) III* | This Paper | MGC619 |
| *C. elegans: sas-7(or1940[gfp::sas-7]) III; ltIs37[pie-1p::mCherry::his-58 + unc-119(+)]IV; sas-6(syb3988[sas-6(5 A)::degron::flag::ha::flag::ha]) IV; ieSi64[gld-1p::tir1::mRuby::gld-1 3'UTR + Cbr-unc-119(+)] II; unc-119(ed3) III* | This Paper | MGC620 |
| *C. elegans: unc-119(ed3) III; ddIs36[pie-1p::gfp::sas-5 + unc-119(+)]; sas-6(syb5687[sas-6(5D)::degron::4xflag])IV/nT1[myo-2p::GFP+pes-10p::GFP + F22B7.9p::GFP qIs51] (IV;V); ieSi64[gld-1p::tir1::mRuby::gld-1 3'UTR + Cbr-unc-119(+)] II; unc-119(ed3) III* | This Paper | MGC621 |
| *C. elegans: sas-7(or1940[gfp::sas-7]) III; ltIs37[pie-1p::mCherry::his-58 + unc-119(+)]IV; sas-6(syb5687[sas-6(5D)::degron::4xflag]) IV / nT1[myo-2p::GFP+pes-10p::GFP + F22B7.9p::GFP qIs51] (IV;V); ieSi64[gld-1p::tir1::mRuby::gld-1 3'UTR + Cbr-unc-119(+)] II; unc-119(ed3) III* | This Paper | MGC622 |
| *C. elegans: sas-6(syb1706[sas-6::degron::flag::ha::flag::ha]) IV* | This Paper | PHX1706 |
| *C. elegans: sas-6(syb3988[sas-6(5 A)::degron::flag::ha::flag::ha]) IV* | This Paper | PHX3988 |
| *C. elegans: sas-6(syb5687[sas-6(5D)::degron::4xflag]) IV /nT1[myo-2p::GFP+pes-10p::GFP + F22B7.9p::GFP qIs51] (IV;V)* | This Paper | PHX5687 |
| *C. elegans: unc-119(ed3) III; ddEx10[pie-1p::gfp::sas-4 + unc-119(+)]* | Caenorhabditis Genetics Center | TH26 |

| Reagent/Resource | Reference or Source | Identifier or Catalog Number |
|---|---|---|
| *C. elegans: unc-119(ed3) III; ddIs36[pie-1p::gfp::sas-5 + unc-119(+)]* | Caenorhabditis Genetics Center | TH61 |
| *C. elegans: cdk-1(q923[3xMyc::cdk-1])* | Caenorhabditis Genetics Center | JK5751 |
| *C. elegans: neSi12 [cdk-1::GFP + Cbr-unc-119(+)] II; unc-119(ed3) III* | Caenorhabditis Genetics Center | WM242 |
| *C. elegans: cdk-1(ne2257)* | Caenorhabditis Genetics Center | WM99 |
| **Recombinant DNA** | | |
| pET28a(+) | Addegen | Cat#69864-3 |
| pGEX-6P-1 | Miaoling | Cat#P0005 |
| pCDNA3.1-3xHA-N | Miaoling | Cat#P0160 |
| pCMV-Myc-N | Miaoling | Cat#P0816 |
| Plasmid: MBP-SAS-6-mCherry | This paper | N/A |
| Plasmid: MBP-SAS-6-GFP | This paper | N/A |
| Plasmid: Myc-CDK-1 | This paper | N/A |
| Plasmids: SAS-6-mCherry-His and its mutants | This paper | N/A |
| Plasmids: SAS-6-mCherry-HA and its mutants | This paper | N/A |
| Plasmid: SAS-6-GFP | This paper | N/A |
| Plasmids: GST-SAS-6 and its mutants | This paper | N/A |
| Plasmid: HA-CDK1 | This paper | N/A |
| Plasmid: SAS-4-GFP | This paper | N/A |
| Plasmid: SAS-5-GFP | This paper | N/A |
| Plasmids: SASS-6-GFP and its mutants | This paper | N/A |
| Plasmid: SASS-6-mCherry | This paper | N/A |
| **Antibodies** | | |
| Mouse anti-HA | Covance | Cat# MMP-101P-200 |
| Mouse anti-thiophosphate ester | Abcam | Cat# ab92570 |
| Mouse anti-Flag | Sigma-Aldrich | Cat# F3165 |
| Mouse anti-MPM-2 | Sigma-Aldrich | Cat#05368 |
| Mouse anti-Maltose binding protein antibody | Abcam | Cat# ab119994 |
| Rabbit anti-Myc | Proteintech | Cat# 16286-1-AP |
| Chicken anti-GFP | Abcam | Cat# ab13970 |
| Rabbit anti-GFP | Abcam | Cat# ab290 |
| Mouse anti-His | TransGen | Cat# HT501 |
| Rabbit anti-Centrin 1 | Proteintech | Cat# 12794-1-AP |
| Mouse anti-SASS-6 | Santa Cruz | Cat# sc-81431 |
| Rabbit anti-CEP192 | Proteintech | Cat# 18832-1-AP |
| Mouse anti-γ-tubulin | Sigma-Aldrich | Cat# t6557 |
| Mouse anti-α-tubulin | Sigma-Aldrich | Cat# T9026 |
| Rabbit anti-SAS-6-pS408 | This study | N/A |
| Mouse anti-MAPK-YT | Sigma-Aldrich | Cat# M8159 |
| HRP-conjugated anti-mouse | Cell Signaling | Cat# 7076S |
| HRP-conjugated anti-chicken | Proteintech | Cat# SA00001-6 |
| Donkey anti-mouse Alexa Fluor 488 | Invitrogen | Cat# A21202 |

| Reagent/Resource | Reference or Source | Identifier or Catalog Number |
|---|---|---|
| Donkey anti-rabbit Alexa Fluor 488 | Invitrogen | Cat# A21206 |
| Goat anti-rabbit-STAR RED | Abberior | Cat# STRED-1002 |
| **Oligonucleotides and other sequence-based reagents** | | |
| PCR primers | This study | Table EV1 |
| **Chemicals, Enzymes and other reagents** | | |
| BamHI-HF Endonuclease | New England Biolabs | Cat#R3136S |
| NdeI | New England Biolabs | Cat#R0111S |
| DMSO | Sigma-Aldrich | Cat#472301 |
| RO3306 | Selleck | Cat#S7747 |
| Purvalanol A | MCE | Cat#HY-18299A |
| Paraformaldehyde | Sigma-Aldrich | Cat#P6148 |
| 1,6-Hexanediol | Macklin | Cat#H810887 |
| Sucrose | Sinopharm Chemical Reagent Co., Ltd | Cat#10021418 |
| Sarcosyl | Macklin | Cat#N812358 |
| Acrylamide | Macklin | Cat#A800656 |
| Sodium Acrylate | Macklin | Cat#S742463 |
| BIS | Macklin | Cat#N813086 |
| Ammonium Persulfate | Sigma-Aldrich | Cat#248614 |
| Tetramethylethylenediamine | Sigma-Aldrich | Cat#8920-OP |
| DAPI | Introgen | Cat#D1306 |
| Hoechst | Solarbio | Cat#C0031 |
| poly-D-lysine | Gibco | Cat#A3890404 |
| Vaseline | Solarbio | Cat#V8230 |
| VECTASHIELD Antifade Mounting Medium with DAPI | Vector Laboratories | Cat#H-1200 |
| **Software** | | |
| Photoshop | Adobe | https://www.adobe.com/products/photoshop.html |
| Graphpad Prism 8.0 | Graphpad Software | https://www.graphpad.com/scientificsoftware/prism/ |
| ImageJ | National Institutes of Health | https://imagej.nih.gov/ij/ |
| **Other** | | |
| Leica SP8 confocal microscope | Leica | N/A |
| HT-7800 transmission electron microscope | Hitachi | N/A |
| STED Super resolution microscopy | Leica | N/A |
| **Table EV1 PCR Primers** | | |
| Plasmid: GST-SAS-6 | | F: 5′-cccctgggatccccggaattcATGACTAGCAAAATTGCATTATTCG-3′<br>R: 5′-ctcgagtcgacccgggaattcTCGTTGAGCGGGTGGGGT-3′ |
| Plasmid: GST-SAS-6-ΔN | | F: 5′-ttccaggggcccctgggatccATGACTAGCAAAATTGCATTATTCG-3′<br>R: 5′-ctcgagtcgacccgggaattcTCGTTGAGCGGGTGG-3′ |
| Plasmid: GST-SAS-6-ΔM | | F: 5′-cgcagtccgcggagatccttctgggcttc-3′<br>R: 5′-gaagcccagaaggatctccgcggactgcg-3′ |
| Plasmid: GST-SAS-6-ΔC | | F: 5′-ttccaggggcccctgggatccATGACTAGCAAAATTGCATTATTCG-3′<br>R: 5′-acccgggaattccggggatccTTAAGCTGGACTAAACCGTTGGG-3′ |
| Plasmid: GST-SAS-6-N | | F: 5′-ttccaggggcccctgggatccATGACTAGCAAAATTGCATTATTCG-3′<br>R: 5′-acccgggaattccggggatccTTAATCTCCGCGGACTGCG-3′ |
| Plasmid: GST-SAS-6-M | | F: 5′-ttccaggggcccctgggatccCATTTAATTTCACATCTTCTGAAAATTTG-3′<br>R: 5′-acccgggaattccggggatccTTAAGCTGGACTAAACCGTTGGG-3′ |

| Table EV1 PCR Primers | |
|---|---|
| Plasmid: GST-SAS-6-C | F: 5'-cccctgggatccccggaattcCCTTCTGGGCTTCCCGGG-3'<br>R: 5'-ctcgagtcgacccgggaattcTCGTTGAGCGGGTGG-3' |
| Plasmid: SAS-6-mCherry-His | F: 5'-ctgttccaggggccccatatgATGACTAGCAAAATTGCATTATTCG-3'<br>R: 5'-gcccttgctcaccatggatccTCGTTGAGCGGGTGGGG-3' |
| Plasmid: SAS-6-C-mCherry-His | F: 5'-ctgttccaggggccccatatgCATTACCGAGCCCAACGGT-3'<br>R: 5'-gcccttgctcaccatggatccTCGTTGAGCGGGTGGGG-3' |
| Plasmid: SAS-6-S408A | F: 5'-cccattaccgagcccaacggtttgctccagctccttc-3'<br>R: 5'-gaaggagctggagcaaaccgttgggctcggtaatggg-3' |
| Plasmid: SAS-6-T426A | F: 5'-gcttgaatgatggtgtcaagcctatttgttagtccggtc-3'<br>R: 5'-gaccggactaacaaataggcttgcaccatcattcaagc-3' |
| Plasmid: SAS-6-T437A | F: 5'-attagctccatatggagcgtggggtccaagaacag-3'<br>R: 5'-ctgttcttggaccccacgctccatatggagctaat-3' |
| Plasmid: SAS-6-T464A | F: 5'-gaaacgaaaagcatgaggagctgcaattgtcgaattttgga-3'<br>R: 5'-tccaaaattcgacaattgcagctcctcatgctttcgtttc-3' |
| Plasmid: SAS-6-T487A | F: 5'-ttgagcgggtggggcgttcgtcacacttg-3'<br>R: 5'-caagtgtgacgaacgccccacccgctcaa-3' |
| Plasmid: SAS-6-S408D | F: 5'-cccattaccgagcccaacggtttgatccagctccttc-3'<br>R: 5'-gaaggagctggatcaaaccgttgggctcggtaatggg -3' |
| Plasmid: SAS-6-T426D | F: 5'-cagaccggactaacaaataggcttgatccatcattcaagcctgttctt-3'<br>R: 5'-aagaacaggcttgaatgatggatcaagcctatttgttagtccggtctg-3' |
| Plasmid: SAS-6-T437D | F: 5'-gcctgttcttggaccccacgatccatatggagctaatttg-3'<br>R: 5'-caaattagctccatatggatcgtggggtccaagaacaggc-3' |
| Plasmid: SAS-6-T464D | F: 5'-acaactcttaatttccaaaattcgacaattgcagatcctcatgcttttcgtt-3'<br>R: 5'-aacgaaaagcatgaggatctgcaattgtcgaattttggaaattaagagttgt-3' |
| Plasmid: SAS-6-T487D | F: 5'-gttcaagtgtgacgaacgacccacccgctcaacgag-3'<br>R: 5'-ctcgttgagcgggtgggtcgttcgtcacacttgaac-3' |
| Plasmid: SAS-6-mCherry-HA | F: 5'-ggtaccgcgggcccgggatccATGACTAGCAAAATTGCATTATTCG-3'<br>R: 5'-catgccgctgccgccggatccTCGTTGAGCGGGTGG-3' |
| Plasmid: HA-CDK1 | F: 5'-tggcggccgctcgagtctagaATGGAAGATTATACCAAAATAGAGAAAATT-3'<br>R: 5'-gacgtcgtatgggtatctagaCATCTTCTTAATCTGATTGTCCAAATC-3' |
| Plasmid: Flag-Cyclin B1 | F:5'-caagcttgcggccgcgaattcGATGGCGCTCCGAGTCACC-3'<br>R:5'-atcagatctatcgatgaattcTTACACCTTTGCCACAGCCTT-3' |
| Plasmid: SASS-6-GFP | F: 5'-ggtaccgcgggcccgggatccATGAGCCAAGTGCTGTTCCACC-3'<br>R: 5'-ttatctagatccggtggatccACTGTTTGGTAACTGCCCAGGG-3' |
| Plasmid: SASS-6-ΔC-GFP | F: 5'-ggtaccgcgggcccgggatccATGAGCCAAGTGCTGTTCCACC-3'<br>R: 5'-ttatctagatccggtggatccGCTGGAATGTGCAGGCGG-3' |
| Plasmid: SAS-4-GFP | F: 5'-ggtaccgcgggcccgggatccATGGCTTCCGATGAAAATATCG-3'<br>R: 5'-catgccgctgccgccggatccTTTTTTCCACTGGAACAAAGTTGT-3' |
| Plasmid: SAS-5-GFP | F: 5'-ggtaccgcgggcccgggatccATGAATAATTACGACGACTTACCCTG-3'<br>R: 5'-catgccgctgccgccggatccTTTCCTGCGAGCGTATTTTTC-3' |
| Plasmid: SAS-6-GFP | F: 5'-ggtaccgcgggcccgggatccATGACTAGCAAAATTGCATTATTCG-3'<br>R: 5'-catgccgctgccgccggatccTCGTTGAGCGGGTGGGGT-3' |
| Plasmid: MBP-SAS-6-GFP-His | F: 5'-gacgacgatgacaaggcggccgcAATGACTAGCAAAATTGCATTATTCG-3'<br>R: 5'-attgcgccggcgcctgcggccgcTCGTTGAGCGGGTGGGGT-3' |
| Plasmid: MBP-SAS-6-mCherry-His | F:5'-gacgacgatgacaaggcggccgcAATGACTAGCAAAATTGCATTATTCG-3'<br>R: 5'-tgatgatgatgatgtgcgatcgcCTTGTACAGCTCGTCCATGCC-3' |

## Worm strains and culture conditions

The wild-type *C. elegans* strain used in this study was Bristol N2, and strains used in this study is listed in Table EV1. All strains were maintained on nematode growth media (NGM) agar plates spread with *Escherichia coli* OP50 at 20 °C, unless stated otherwise. The following worm strains were generated by CRISPR/Cas9 genomic editing: PHX1706: *sas-6::degron::flag::ha::flag::ha;*

PHX3988: *sas-6(5 A)::degron::flag::ha::flag::ha;* PHX5687::*sas-6(5D)::degron::4xflag/nT1.* In brief, young adult N2 hermaphrodites were injected with a set of plasmids expressing Cas9 (*Peft-3Cas9-SV40 NLStbb-2 3'UTR*, 200 ng/μl), the targeting single guide RNA (100 ng/μl), and mCherry coinjection markers (pCFJ90, 5 ng/μl; pCFJ104, 5 ng/μl). Plasmid containing a donor template sequence were coinjected (100 ng/μl) to achieve precise genome editing. F1 progeny expressing mCherry coinjection

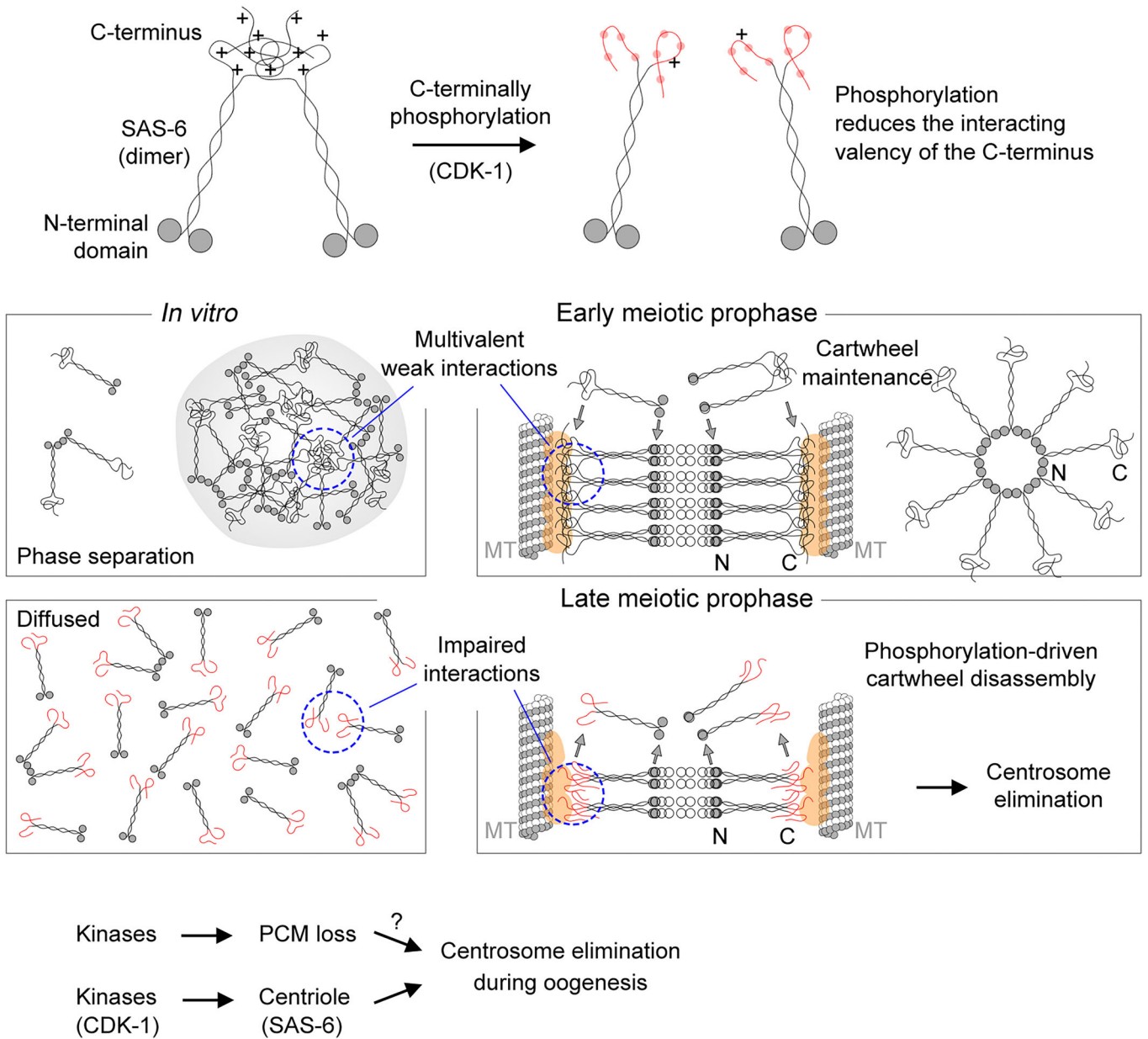

**Figure 10. Model of the regulation of SAS-6 properties and cartwheel dynamics by CDK-1-mediated phosphorylation.**

The C-terminus-mediated multivalent weak interactions promote SAS-6 phase separation in vitro. Such multivalent weak interactions may also support cartwheel assembly and maintenance in vivo, allowing the subunits of the highly ordered cartwheel to be dynamic. Phosphorylation of the SAS-6 C-terminus by CDK-1 may reduce the valency ('+') of these weak interactions, impairing intramolecular interactions and leading to cartwheel disassembly. This, in turn, disrupts centrosome assembly and facilitates timely centrosome elimination. Kinases-mediated PCM loss and cartwheel disassembly may work together to ensure efficient centrosome elimination during oogenesis.

marker was singled and screened for desired genome edits via PCR and sequencing.

## Antibodies used in this study

The following primary antibodies were used at the indicated dilutions: mouse anti-HA (1:2000 for western blot (WB); Covance, MMP-101P-200), mouse anti-thiophosphate ester (1:2000 for WB; Abcam, ab92570), mouse anti-Flag (1:5000 for WB; 1:2000 for IF;

Sigma-Aldrich, F3165), mouse anti-MPM-2 (1:2000 for WB and IF; Sigma-Aldrich, 05368), mouse anti-Maltose binding protein antibody (1:2000 for WB; Abcam, ab119994), rabbit anti-Myc (1:2000 for WB; Proteintech, 16286-1-AP), chicken anti-GFP (1:2000 for WB; Abcam, ab13970), mouse anti-His (1:2000 for WB, TransGen, HT501), rabbit anti-Centrin 1 (1:1000 for IF, Proteintech, 12794-1-AP), mouse anti-SASS-6 (1:1000 for IF, Santa Cruz, sc-81431), rabbit anti-CEP192 (1:1000 for IF; Proteintech, 18832-1-AP), mouse anti-MAPK-YT (1:200 for IF; Sigma-Aldrich, M8159),

mouse anti-α-tubulin (1:500 for IF; Sigma-Aldrich, T9026), and mouse anti-γ-tubulin (1:1000 for IF; Sigma-Aldrich, t6557). HRP-conjugated anti-mouse (1:5000; Cell Signaling, 7076S), and HRP-conjugated anti-chicken (1:5000; Proteintech, SA00001-6). Secondary antibodies for immunostaining were donkey anti-mouse Alexa Fluor 568 (Invitrogen, A21202), donkey anti-rabbit Alexa Fluor 488 (Invitrogen, A21206), and goat anti-rabbit-STAR RED (Abberior, STRED-1002).

Ser408 phospho-specific SAS-6 antibody was generated by GL Biochem (Shanghai) using phosphorylated antigen peptide CYRAQRF(pSer)PAPSGL conjugated with keyhole limpet hemocyanin (KLH). The control peptides without phosphorylation (CYRAQRFSPAPSGL) were synthesized and used during antibody purification. Rabbits injected with phosphopeptides were given boosters every 2 weeks. Bleeds collected from the rabbits immunized with the antigen peptides underwent two rounds of affinity purification. The serum was first purified by passing through a column to which the antigenic peptide was coupled in order to isolate phosphor-specific SAS-6 antibodies. The eluate was then passed through a second column to which the control peptide was coupled to remove any non-phosphorylated SAS-6 antibodies, and the flow-through was collected. An ELISA titer of ≥32000 against the modified peptide and a modified/unmodified titer ratio of ≥32 were used as validation criteria.

## Plasmid construction

To construct GFP- or mCherry-tagged centriolar proteins, the nematode cDNAs encoding SAS-4, SAS-5, and SAS-6 were inserted into the BamHI site of pEGFP-N1 or pmCherry-N1 vectors, respectively. To make prokaryotic expression plasmids for nematode or mammalian SAS-6, the PCR fragment encoding SAS-6 was subcloned into the BamHI site of pET.28a-mCherry vector. To create MBP-SAS-6-10xHis and GST-SAS-6 constructs, the PCR fragments encoding SAS-6 were subcloned into the pGEX4T1-MBP-10xHis or pGEX6P1-GST vectors, respectively. To construct HA-tagged CDK1 or Flag-tagged Cyclin B1, mammalian or nematode cDNAs encoding corresponding proteins were cloned into pCDNA3.1-3xHA-N or p3xFLAG-cmv-7.1 vector backbone. All point mutations and truncation mutations were introduced with the QuikChange II XL site-directed mutagenesis kit (Agilent Technologies). All plasmids were sequenced to verify the desired mutations and the absence of unintended mutations.

## Immunofluorescence microscopy

Gonad staining was carried out according to a previous study (Zhang et al, 2020). Gonads were dissected from young adult hermaphrodites (24 h post L4) in dissecting buffer (25 mM HEPES pH 7.4, 118 mM NaCl, 48 mM KCl, 2 mM EDTA, 5 mM EGTA, 0.1% Tween-20, and 10 mM NaN₃) on coverslips, and a freeze crack on dry ice was used to transfer the gonads onto polylysine glass slides. Gonads were fixed first in methanol (−20 °C) for 1 min and then in 4% paraformaldehyde in PBS at room temperature for 30 min. Slides were blocked with 0.5% BSA in PBST at room temperature for 1 h, followed by primary antibody incubation overnight at 4 °C. After three washes in PBST, slides were incubated with secondary antibodies at room temperature for 2 h. Chromatin was stained with DAPI (1 μg/ml) for 5 min and followed by five

washes in PBST. Finally, gonads were mounted with Vectashield mounting medium (Vector Laboratories) for image acquisition. For most experiments, a Leica SP8 confocal microscope equipped with a 63×/1.40 oil objective using LAS-X software was used for image acquisition. Fluorescence images are maximum-intensity projections through 3D data stacks of the whole cell. Wide-field images were generated through whole nuclei at 200-nm intervals on a DeltaVision OMX microscope system with 60×/1.42 objective lenses by SoftWoRx software in the conventional imaging mode. Images were deconvolved using a conservative algorithm with 10 iterations. Structure illumination microscopy images were captured as 125 nm-spaced Z stacks on the DeltaVision OMX microscope system with 60×/1.42 lens in 3D-SIM mode.

## Gonad spreading

Gonad spreading in C. elegans was conducted following the protocol described by Woglar et al (2022). Briefly, gonads from approximately 1000 adult worms were dissected in 30 μL of PBS-T (0.2× PBS, 0.1% Tween 20) on an ethanol-cleaned 22 × 22 mm coverslip. A 10 μL aliquot of the dissected gonads was transferred to a fresh ethanol-cleaned 22 × 22 mm coverslip, combined with 50 μL of spreading solution (32 μL Fixative [4% paraformaldehyde and 3.2% sucrose in water], 16 μL Tween-20 [1‰ Tween-20 in water], and 2 μL Sarcosyl solution [1% Sarcosyl in water]). The coverslips were air-dried at room temperature (RT) and subsequently incubated at 37 °C for 1 h. Processed coverslips were either immediately used for staining and expansion or stored at −80 °C for future use.

## Gonads expansion for U-Ex-STED

Ultrastructure expansion microscopy was conducted following the protocol described by Woglar et al (2022). Briefly, dried coverslips were incubated in methanol at −20 °C for 5 min, then washed three times in PBS-T for 5 min each, followed by two additional 5-min washes in PBS. The coverslips were then incubated overnight at room temperature with gentle agitation in Acrylamide/Formaldehyde solution (1% Acrylamide and 1% Formaldehyde in PBS) in a six-well plate. Subsequently, the coverslips were washed three times in PBS for 5 min each. For gelation, 50 μl of monomer solution (19% [wt/wt] Sodium Acrylate, 10% [wt/wt] Acrylamide, 0.05% [wt/wt] BIS in PBS) supplemented with 0.5% Tetramethylethylenediamine (TEMED) and 0.5% Ammonium Persulfate (APS) was added to each coverslip on a piece of Parafilm and incubated for 1 h at 37 °C in a humidified, dark environment. All subsequent steps were performed under mild agitation at room temperature unless otherwise noted. Gels were incubated in a 6 cm Petri dish for 15 min in denaturation buffer (200 mM SDS, 200 mM NaCl, and 50 mM Tris in distilled water, pH 9), followed by a 1-h incubation on a 95 °C hot plate in fresh denaturation buffer. The gels were then transferred to 15 cm Petri dishes, washed five times with distilled water for 20 min each, and incubated overnight in distilled water at 4 °C. The expansion factor was determined by measuring the gel dimensions with a ruler.

## Immunofluorescence of expanded gels

Following expansion, the gels were trimmed to fit into a six-well plate Prior to antibody staining, gels were incubated in blocking buffer (10 mM HEPES, pH 7.4, 3% BSA, 0.1% Tween 20, 0.05%

sodium azide) for 1 h at room temperature. Primary antibodies, diluted in blocking buffer, were then applied and incubated overnight at room temperature. After three 10-min washes in blocking buffer, gels were incubated with secondary antibodies (diluted in blocking buffer supplemented with 1 µg/mL Hoechst) at 37 °C in the dark for 3 h. Following another three 10-min washes in blocking buffer, gels were transferred to a 6 cm Petri dish for re-expansion by six sequential 20-min washes in distilled water. For imaging, gels were sectioned and mounted onto poly-D-lysine (Gibco, #A3890404) -coated coverslips (60 × 24 mm). To prevent desiccation, gel edges were sealed with Vaseline (Solarbio, #V8230).

## 1,6-Hexanediol treatment

To determine whether the SAS-6 condensates could be dissolved by 1,6-hexanediol, cells were cultured on 3.5-cm glass-bottom dishes and transfected with indicated plasmids for 24 h. Subsequently, cells were treated with 10% 1,6-hexanediol in DMEM medium during real-time image acquisition. Similarly, 10 µL of purified SAS-6 (FL or C-terminal domain) protein was mixed with 100 µL of 11% 1,6-hexanediol and then subjected to observe phase separation by real-time imaging. To observe the localization of centriolar proteins under the treatment of 10% 1,6-hexanediol, HEK293T cells transfected with indicated plasmids were fixed in methanol (−20 °C) for 5 min and then stained with DAPI. For 1,6-hexanediol treatment in vivo, gonads were dissected from young adults (24 h post L4) in 20 µL PBS buffer containing 0.05% Tween-20 (PBST) and then mixed with 200 µL of 11% 1,6-hexanediol for 5 min. Gonads were picked and transferred to a new coverslip and were fixed and stained with DAPI. For 1,6-hexanediol washout, gonads incubated in 10% 1,6-hexanediol for 5 min were picked and transferred to 1 mL PBST. After rapid centrifugation, the gonads were incubated in a new PBST solution for the indicated time, followed by fixed and stained with DAPI.

## Real-time imaging

Cells were plated on glass-bottom dishes (801001, NEST). 24 h after transfection of the indicated plasmids, cells were cultured in the indicated medium and imaged at 37 °C with 5% $CO_2$ using a 63× objective with immersion oil on a spinning disk confocal microscope. Cells were monitored over time in xyzt mode and images were taken every 3 min or 5 min for 24 h. For time-lapse imaging of meiosis, young adult worms (24 h post L4) were placed in embryonic culture medium (ECM; 84% Leibovitz L-15 medium [Invitrogen], 9.3% FBS [Hyclone], 4.7% sucrose, 0.01% levamisole, and 2 mM EGTA) on a coverslip and cut at the pharynx. The extruded gonads were transferred to a glass-bottom dish pretreated with poly-D-lysine [Invitrogen] and then subjected to live cell imaging according to the above method. Images were taken every 1 s or 5 s for 10 min to capture the dynamic processes during meiosis. For time-lapse tracking of phase separation, proteins were placed on the glass-bottom dish and monitored over time in xyt mode and images were taken every 3 s for 10 min.

## Fluorescence recovery after photobleaching

Cells cultured on glass-bottom dishes were transfected with the indicated plasmids for 24 h. FRAP experiments were performed on

a Leica Sp8 laser scanning confocal microscope at room temperature. Defined regions were photobleached at 488 nm or 563 nm and the fluorescence intensity was measured by ImageJ and further analyzed by GraphPad Prism8. Data are shown as the fractional recovery, which refers to the fraction of the difference between the intensity before and immediately after bleaching. For FRAP analysis in the germline, worms expressing GFP::SAS-5 or SAS-6::GFP were anesthetized in ECM on a coverslip, and their gonads were released by dissection. The dissected gonads were transferred to a glass-bottom dish preincubated with poly-D-lysine. Defined regions were photobleached at 488 nm and the fluorescence signal in these regions was collected every 1 s, and the measured intensity was normalized to the initial intensity before bleaching.

## In vitro phase separation assay

For phase separation capacity determination, purified proteins were diluted in ice-cold phase separation buffer (50 mM Tris-HCl pH 7.4, 150 mM NaCl) to the desired concentrations. The protein solutions were incubated at room temperature for 5 min to induce phase separation. After that, 10 µL protein solution was placed on the glass-bottom dish and then subjected to bright field and fluorescence microscopy with a 63x oil objective to examine the formation and fusion of liquid droplets. For phase separation analysis of phosphorylated proteins, immunoprecipitated CDK1 kinase from HEK293T cells was used to phosphorylate purified 2 ug SAS-6 protein in 50 µL kinase reaction buffer (150 mM NaCl, 10 mM $MgCl_2$, 50 mM Tris-HCl pH 8.0, 1 mM DTT, 10 mM NaF, 20 mM β-glycerophosphate, 2.5 mM $Na_4P_2O_7$, 500 µM ATP) at 30 °C for 2 h. 10 µL of supernatant solution containing phosphorylated proteins was used for phase separation examination and protein phosphorylation was validated by SDS-PAGE and western blot.

## Cell culture and transfection

HEK293T (from the American Type Culture Collection) cells were cultured in Dulbecco's Modified Eagle's Medium (DMEM) supplemented with 1% penicillin/ streptomycin and 10% fetal bovine serum (FBS), and maintained at 37 °C with 5% $CO_2$. Plasmid transfection was done with polyethyleneimine (PEI) reagent. Briefly, HEK293T cells were seeded in plates in DMEM with 10% FBS and without antibiotics. After 24 h of cell seeding, the transfection mixture containing plasmids and PEI was added to the culture medium of the adherent cells. After 24 h of transfection, cells were fixed with 4% paraformaldehyde (PFA) or −20 °C methanol for immunostaining or harvested for western blot.

## Immunoprecipitation and mass spectrometry

For the identification of phosphorylation sites in SAS-6, HEK293T cells were transfected with SAS-6-GFP expressing plasmid for 24 h, followed by treatment with 100 ng/ml nocodazole for 16-18 h. The harvested HEK293T cells were lysed with "P150 buffer" (50 mM Tris-HCl, pH 7.4, 150 mM NaCl, 10% glycerol, 0.5% NP-40, 1 mM dithiothreitol), and the lysate was incubated with GFP-Trap agarose beads (ChromoTek) for 4 h at 4 °C. Two methods were used to identify phosphorylation sites. One way is that immunoprecipitated proteins were eluted by boiling in 2% SDS

in 20 mM Tris (pH 7.4) for 5 min. Eluted proteins were precipitated with the ProteoExtract Protein Precipitation Kit (Calbiochem) and digested with trypsin for MS analysis. On the other way, the beads were resuspended in sample buffer and boiled at 95 °C for 10 min. Proteins eluted from the beads were analyzed by SDS-PAGE and immunoblotting. SAS-6-GFP was visualized by Coomassie Brilliant Blue (CBB) and processed for MS analysis to identify SAS-6 modification.

To identify the phosphorylation sites of SAS-6 in vivo, synchronized young adult worms were shredded by vortexing in 15-ml conical tube containing small sharp broken glass coverslips in ice-cold lysis buffer (150 mM NaCl, 20 mM KCl, 1.5 mM MgCl$_2$, 20 mM Tris-HCl pH 7.5, 1% NP-40) with protease inhibitors (PIC and 0.1 mM PMSF) and phosphatase inhibitors (10 mM NaF, 20 mM β-glycerophosphate, and phosphatase inhibitor cocktail II (MCE)). The embryos remained largely intact during the experimental procedure, allow an enrichment of centrosome protein extract from the germline. Embryos and worm debris was removed by centrifugation at 14000 rpm for 15 min, and cell extracts were incubated with Flag nanoantibody agarose beads for 4 °C for 6 h. Beads were washed four times with the lysis buffer, boiled in SDS sample buffer, and subjected to SDS-PAGE analysis and mass spec analysis with gel sections.

## Transmission electron microscopy

HEK293T cells transfected with SAS-6-GFP were suspended and fixed with 0.25% glutaraldehyde for 24 h at 4 °C. The cells were then post-fixed in 1% osmium tetroxide for 1 h, followed by staining with 2% uranyl acetate. After dehydration by a series of ethanol solutions, samples were embedded in Spurr low viscosity resin and cured for 72 h at 65 °C. 70 nm sections were cut and stained with uranyl acetate and lead citrate, and examined using a Hitachi HT-7800 transmission electron microscope at 80 kV.

## Protein purification and GST pulldown

The prokaryotic expression plasmids were transformed into BL21 (DE3)-competent cells (Stratagene). Cells were grown in LB broth under antibiotic selection at 37 °C until OD$_{600}$ at 0.5–0.7 and protein expression was induced by 0.5 mM IPTG for 16 h at 16 °C. Cells were lysed by sonication in buffer A (20 mM Tris-HCl, pH 8.0, 500 mM NaCl, 1 mM EDTA, 1% Triton X-100; for GST fusion protein), or buffer B (20 mM Tris-HCl, pH 8.0, 500 mM NaCl, 1% Triton X-100, 20 mM imidazole; for His tagged proteins). The lysate was clarified by centrifugation at 12,000 rpm for 1 h. The supernatant was incubated with Glutathione Sepharose 4B (GE Healthcare) or Ni-NTA resin (Qiagen) in respective lysis buffer. The resin was washed 5 times with wash buffer (20 mM Tris-HCl, pH 7.4, 150 mM NaCl), or eluted with 250 mM imidazole.

For GST pulldown of proteins from cell lysates, HEK293T cells expressing target proteins were lysed in the "P150 buffer" containing protease inhibitors and phosphatase inhibitors. The lysates were centrifuged at 15,000 rpm for 30 min, and the resulting supernatants were first precleared with glutathione Sepharose 4B beads, then incubated with GST fusion proteins immobilized to glutathione Sepharose 4B beads at 4 °C for 4 h. Subsequently, the beads were washed 5 times with the same buffer and subjected to immunoblotting or CBB staining.

## In vitro kinase assay

HEK293T cells were co-transfected with HA-tagged CDK1 plasmid and Flag-tagged Cyclin B1 plasmid. After 24 h, the cells were lysed in P150 buffer. CDK1/Cyclin B1 kinase complex was then purified using anti-HA magnetic beads. The beads were washed three times with P150 buffer (50 mM Tris-HCl, pH 8.0, 150 mM NaCl, 20 mM imidazole) and once with kinase buffer (50 mM Tris-HCl, pH 8.0, 10 mM MgCl$_2$ and 150 mM NaCl). For phosphorylation detection using anti-MPM-2 antibody, the purified kinase and SAS-6 proteins were mixed with ATP (500 μM) in kinase assay buffer. After 1 h incubation at 30 °C, the reaction was terminated by adding EDTA to a final concentration of 20 mM (pH 8.0) and incubating at 37 °C for 5 min. For phosphorylation detection using the anti-thiophosphate ester antibody, ATP-γ-S (500 μM) was used instead of ATP. After terminating the kinase reaction, p-nitro benzyl-mesylate (PNBM) (Abcam, ab138910) (2.5 mM) was added and incubated at 25 °C for 1 h to allow the formation of thiophosphate ester side chains in the reaction with ATP-γ-S. Western blot analysis was then performed using the respective antibodies to determine kinase activity.

## Auxin treatment

NGM plates containing 1 mM auxin were prepared according to Zhang et al (2015), seeded with OP50 E. coli bacteria and left at room temperature for 1–2 days to allow bacterial lawn growth. The natural auxin indole-3-acetic acid (IAA) was purchased from Sigma-Aldrich (#45533). Control plates contained the same dilution of ethanol, in which the auxin stock solution was prepared, as auxin plates. L4 animals were first plated on standard NGM plates containing OP50 and incubated at 20 °C for 12 h, followed by transfer to auxin or control plates and 12 h of treatment. Subsequently, the gonads were dissected for immunofluorescence.

## Feeding RNAi and CDK-1 inhibitor exposure

Feeding RNAi was performed to knock down *cdk-1*. To construct the RNAi clone, the full-length *cdk-1* cDNA was PCR-amplified with a pair of primers (forward primer: 5′-caggaattcgatatcaagct-tatggatcctattcgcgaagga-3′; reverse primer: 5′-gtcgacggtatcgataagct-taaagagttcgagttctcctcggt-3′) and inserted into the pL4440 vector at the HindIII site through Gibson assembly. The constructed plasmids were transferred into HT115 bacteria, and bacteria carrying the empty pL4440 vector were used as control. Preparation of RNAi plates was performed as previously described (Gao et al, 2015). L4 worms were transferred to the RNAi plates for 24 h, followed by immunofluorescence microscopy analysis. For CDK-1 inhibitor exposure, inhibitors were added into the medium at indicated concentrations while preparing the NGM plates. The concentrated OP50 bacteria resuspended with 50 μL LB medium containing 1 μL RO3306 stock (27 mM) or 1 μL PA stock (100 mM), was seeded on the NGM plate. DMSO was added for the control treatments. L4-stage animals were placed on the plates for 24 h and then subjected to analysis by immunostaining.

## Ethanol fixation for centrosome observation in intestinal cells

To examine centrosome elimination in intestine cells, synchronized L2 worms were washed off the plate into a 1.5-mL tube using PBS-T and

subsequently washed once in PBS-T and kept for 60 min in PBS-T to allow emptying of intestines. PBS-T was removed, and 1 mL of 100% ethanol was added and the tube was incubated for 3 min at room temperature. Worms were precipitated by gravity, and ethanol then removed completely before resuspension of worms in 1 mL PBST containing DAPI (1 μg/mL) for 10 min. After washing three times with PBST, worms were mounted on a slide with a coverslip.

## Protein structure analysis

The protein sequence analysis and secondary structure prediction were performed with Porter 5 (distilldeep.ucd.ie/porter) as described previously (Zhang et al, 2020).

## Quantification and statistical analysis

To quantify the fluorescence signal of centriolar proteins, z-stacks of whole germ cells were captured with a 63x/1.40 oil objective at 0.5 μm intervals using fixed parameters for all genotypes and treatments. A maximum-intensity projection of the 3D data was generated, and measurements were performed with ImageJ (National Institutes of Health, Bethesda, MD, USA) and Rstudio. The projection image of each cell was first converted to "Text image" with ImageJ. Further calculations were performed in batch with in-house R scripts to identify and measure the brightest focus (5 pixels × 5 pixels) signal intensity for each cell. The corresponding background intensity was defined as the mean of the lower half of the pixels in the "Text image" when sorted by intensity values. The fluorescence intensity ratio between the brightest focus and the background was defined as the relative centrosome protein fluorescent intensity. For cells exhibiting two centrosome foci, the brighter one was automatically selected and measured. Measurements from regions without a centrosome typically yielded values between 1.5 and 2.0, depending on background signal noise.

MPM-2 intensity was measured using ImageJ. For each cell, the total (cumulative) intensity and mean background intensity were quantified. The relative MPM-2 intensity was then calculated by normalizing the cumulative MPM-2 intensity to the mean background intensity. The average intensity of cells at a specific stage in a gonad was used for further statistical comparisons.

Statistical analysis was carried out with the GraphPad Prism 8.0 software, and all data were presented as mean ± SEM or mean ± SD as indicated in the figure legends. Statistical significance was determined by the Student's t-test for comparison between two groups and by the ANOVA test for multiple comparisons. P-values less than 0.05 were considered statistically significant. Details of statistical significance can be found in the figures and figure legends.

## Data availability

This study includes no data deposited in external repositories. All data supporting the findings of this study are available within the paper.

The source data of this paper are collected in the following database record: biostudies:S-SCDT-10_1038-S44319-025-00485-7.

## Peer review information

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

## Acknowledgements

This work was supported by grants from the National Natural Science Foundation of China (32370780 to JG, 32230025 to JZ, 32000481 to FQ, and 32022018 to JG), the Natural Science Foundation of Shandong Province (ZR2020QC072 to FQ), and the China Postdoctoral Science Foundation (2020M672115 to FQ). We thank Professor Guangshuo Ou (Tsinghua University, Beijing, China) for kindly providing GOU899 worms and SunyBiotech (Fuzhou, China) for assisting in creating some mutant strains. We also thank Alexander Woglar and Pierre Gönczy (Swiss Federal Institute of Technology Lausanne, Switzerland) for their valuable advice on the expansion microscopy analysis. Some strains were provided by the Caenorhabditis

Genetics Center, which is funded by the National Institutes of Health Office of Research Infrastructure Programs (P40 OD010440).

## Author contributions

**Feifei Qi**: Conceptualization; Formal analysis; Funding acquisition; Investigation; Methodology; Writing—original draft; Writing—review and editing. **Shanshan Yin**: Investigation. **Xiangrui Yang**: Investigation. **Ning Ju**: Investigation. **Bohan Liu**: Investigation. **Xing Zhang**: Investigation. **Zixuan Zhu**: Investigation. **Li Ji**: Investigation. **Fuxin Zhang**: Investigation. **Li Zhao**: Investigation. **Ruoxi Wang**: Investigation. **Min Liu**: Investigation. **Liangran Zhang**: Resources. **Huijie Zhao**: Resources. **Jun Zhou**: Conceptualization; Resources; Supervision; Funding acquisition; Writing—original draft; Writing—review and editing. **Jinmin Gao**: Conceptualization; Resources; Funding acquisition; Writing—original draft; Project administration; Writing—review and editing.

Source data underlying figure panels in this paper may have individual authorship assigned. Where available, figure panel/source data authorship is listed in the following database record: biostudies:S-SCDT-10_1038-S44319-025-00485-7.

## Disclosure and competing interests statement

The authors declare no competing interests.

# Expanded View Figures

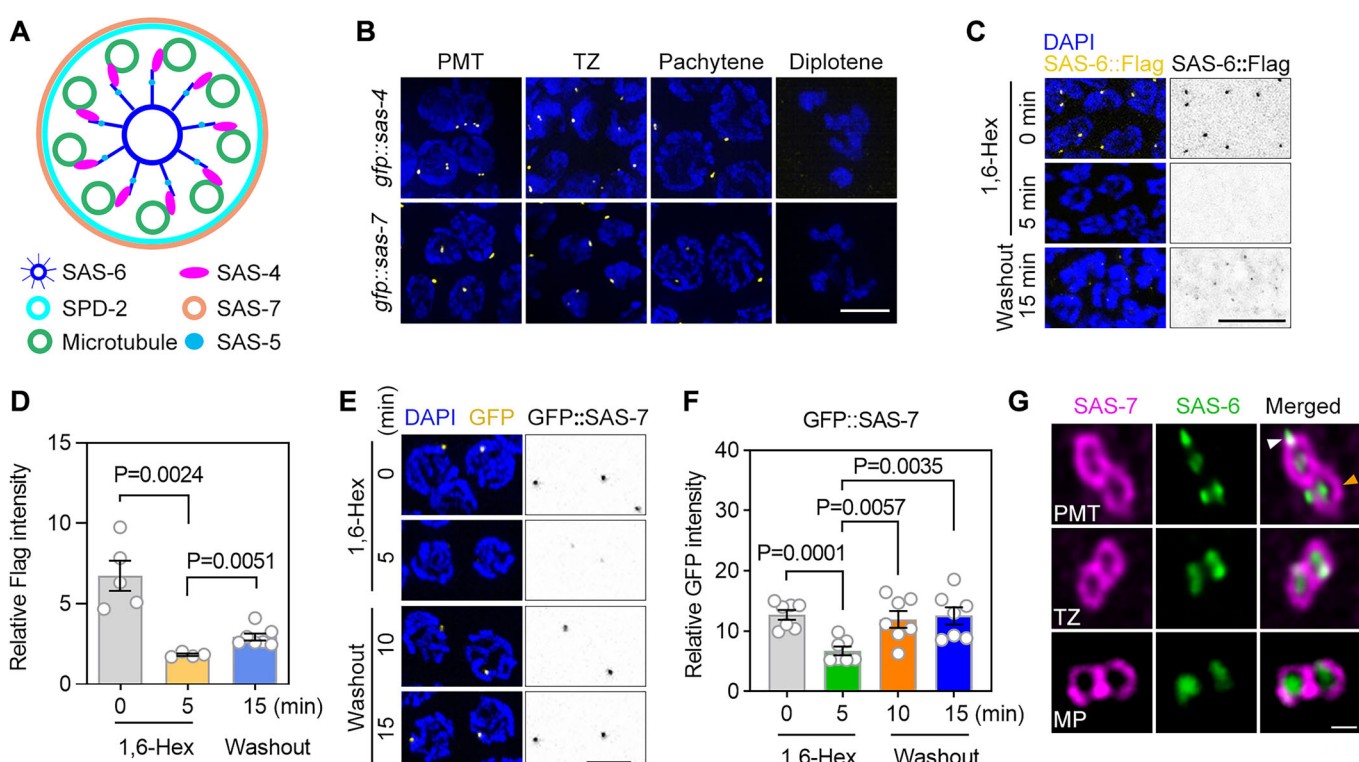

**Figure EV1. Centrosome elimination and dynamic properties of centriole proteins during meiotic prophase in *C. elegans*.**

(A) Schematic representation of the top view of the centriole, with centriole proteins depicted in various colors. (B) Immunofluorescence images of germ cells expressing GFP::SAS-4 or GFP::SAS-7 at the indicated meiotic stages. Chromatin was stained with DAPI (blue). In all four gonads imaged per genotype, GFP signals began to decline at diplotene, consistent with representative images shown. Scale bar, 5 μm. (C, D) Representative images (C) and quantification (D) of SAS-6::Flag fluorescence intensity before or after treatment of *sas-6::flag* worms with 10% 1,6-hexanediol for 5 min, followed by washout for 15 min. Quantification data are shown as mean ± SEM. *P* values were determined using two-tailed unpaired t-tests. At least four gonads were measured. DNA was stained with DAPI (blue). Scale bar, 10 μm. (E, F) Representative images (E) and quantification (F) of GFP::SAS-7 fluorescence intensity before and after treatment of *gfp::sas-7* worms with 10% 1,6-hexanediol for 5 min, followed by washout for 10 min or 15 min. Quantification data are shown as mean ± SEM. *P* values were determined using two-tailed unpaired t-tests. Each dot represents data from a single gonad (*n* = 7 per group). DNA was stained with DAPI (blue). Scale bar, 5 μm. (G) Expansion microscopy analysis of GFP::SAS-7 and SAS-6::FLAG during premeiotic tip (PMT) and early meiotic prophase in germ cells of *gfp::sas-7;sas-6::flag* worms. White arrowheads point to procentrioles, and orange arrowheads to centrioles. SAS-6 and SAS-7 localization patterns were consistent across all gonads examined (*n* = 3). TZ, transition zone; MP, mid pachytene. Scale bar, 500 nm.

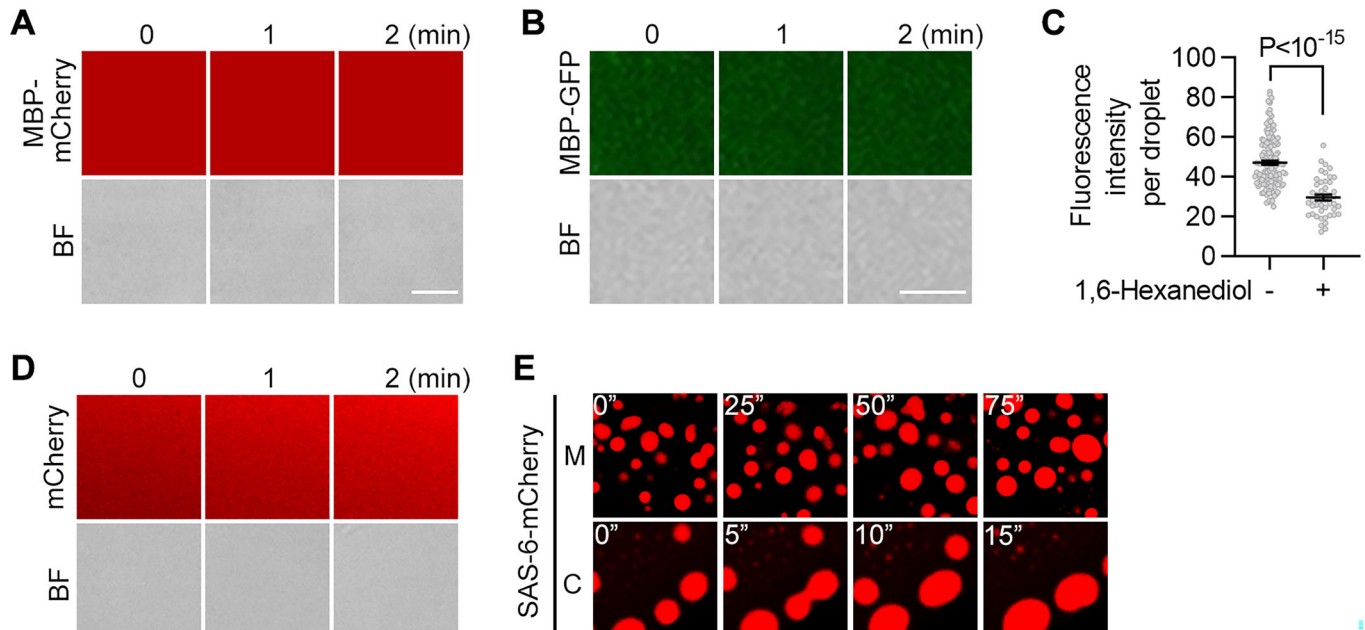

**Figure EV2. Purified SAS-6 protein undergoes phase separation in vitro.**

(A) Fluorescence and bright field (BF) examination of in vitro phase separation of purified MBP-mCherry. Results were consistent across two independent replicates. Scale bar, 5 μm. (B) Fluorescence and BF examination of in vitro phase separation of purified MBP-GFP. Results were consistent across two independent replicates. Scale bar, 5 μm. (C) Quantification of the fluorescence intensity of SAS6-mCherry droplets with or without 10% 1,6-hexanediol treatment as performed in Fig. 2G. Data are shown as mean ± SEM. P values were determined using two-tailed unpaired t-tests. Droplets analyzed: −, n = 150; +, n = 50. (D) Fluorescence and BF examination of in vitro phase separation of purified mCherry. Results were consistent across four independent replicates. Scale bar, 5 μm. (E) Fluorescence images showing that SAS-6-M-mCherry and SAS-6-C-mCherry droplets grow over time. Phase separation capacity of these proteins was consistent across three independent replicates. Scale bar, 5 μm.

**A** DAPI SAS-6-mCherry

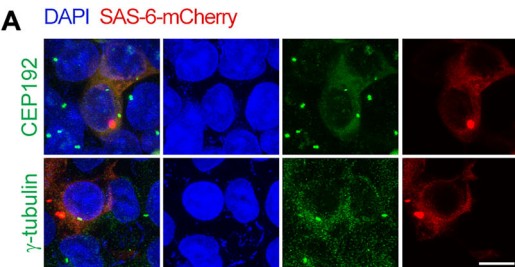

**B** SAS-6-GFP over-expressed cell

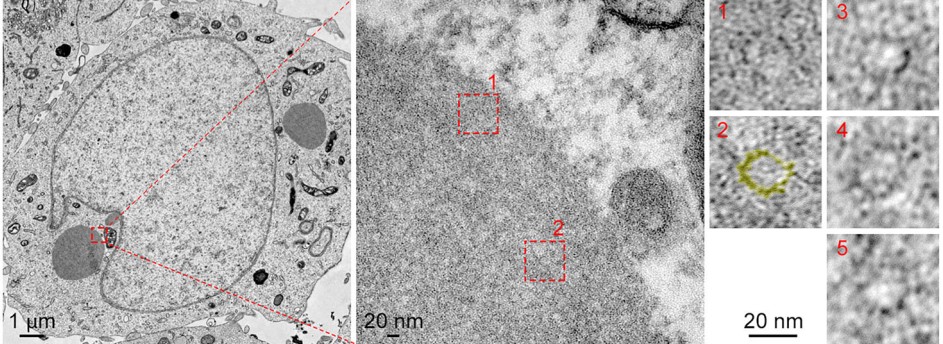

**Figure EV3. SAS-6 droplet formation in HEK293T cells.**

(A) Representative images showing immunostaining of endogenous centriolar proteins (green) and exogenously overexpressed SAS-6-mCherry (red) in HEK293T cells. The relative localization of SAS-6-mCherry droplets and centriolar proteins was consistent across three biological replicates (≥ 10 transfected cells analyzed). Scale bar, 10 μm. (B) Representative negative-stain electron micrograph of SAS-6-GFP condensates formed in HEK293T cells. Examples of cartwheel ring structures are shown on the right. The first two examples (1 and 2) are from the field on the left, and the last three (3, 4, and 5) are from other fields. A total of ten spherical droplets, each 0.8-2.0 μm in diameter, were imaged, with an average of approximately 8 cartwheel-like structures observed per droplet. Scale bars are labeled in the figure.

## A   syb1706: sas-6::degron::flag::ha::flag::ha

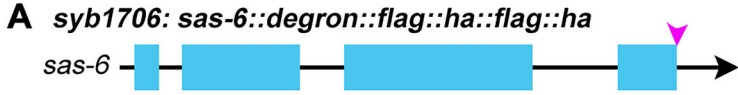

sas-6

Inserted sequence (flanking sequence):
GCTCAACGA**CCTAAAGATCCAGCCAAACCTCCGGCCAAGGCACAAGTTGTGGGATGGCCACCGGTGAGATCATACCGGAA
GAACGTGATGGTTTCCTGCCAAAAATCAAGCGGTGGCCCGGAGGCGGCGGCGTTCGTGAAGGATTACAAGGACGATGACA
AGTACCCATACGATGTTCCAGATTACGCTGATTACAAGGACGATGACAAGTACCCATACGATGTTCCAGATTACGCT**TAA
aatctt

## syb3988: sas-6(5A)::degron::flag::ha::flag::ha

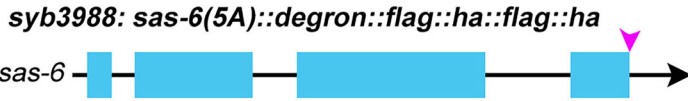

sas-6

Repair template sequence:
GCGCCAGCTCCTAGTGGGCTTCCCGGGCTTCAGACCGGACTAACAAATAGGCTTGCACCATCATTCAAGCCTGTTCTTGG
ACCGCATGCTCCATATGGAGCTAATTTGAACTCGgttagtagaagttctacgaaatgagctacattatcgattttttccgg
caatttccaattttcacacaattttttgagcattttttccgaactgtttcaactcaaaaaaatatttttaatttgttcaaatt
tcagCGAACTCCATTCCGTGACAATACAACTCTTAATTTCCAAAATTCGACAATTGCAGCTCCTCATGCTTTTCGTTTCA
ACAGTCAACTAATCGCCGACGAAACTACTGGTTCAAGTGTGACGAACGCCCCACCCGCTCAACGACCTAAAGATCCAGCC
AAACCTCCGGCCAAGGCACAAGTTGTGGGATGGCCACCGGTGAGATCATACCGGAAGAACGTGATGGTTTCCTGCCAAAA
ATCAAGCGGTGGCCCGGAGGCGGCGGCGTTCGTGAAGGATTACAAGGACGATGACAAGTACCCATACGATGTTCCAGATT
ACGCTGATTACAAGGACGATGACAAGTACCCATACGATGTTCCAGATTACGCT

## syb5687: sas-6(5D)::degron::4xflag

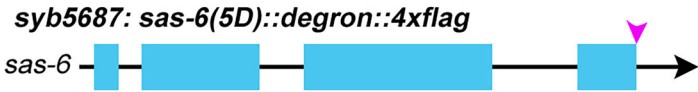

sas-6

Repair template sequence:
GATCCAGCTCCTAGTGGGCTTCCCGGGCTTCAGACCGGACTAACAAATAGGCTGGATCCATCATTCAAGCCTGTTCTTGG
ACCCCACGATCCATATGGAGCTAATTTGAACTCGgttagtagaagttctacgaaatgagctacattatcgattttttccgg
caatttccaattttcacacaattttttgagcattttttccgaactgtttcaactcaaaaaaatatttttaatttgttcaaatt
tcagCGCACGCCATTCCGTGACAATACAACTCTTAATTTCCAAAATTCGACAATTGCAGATCCTCATGCTTTTCGTTTCA
ACAGTCAACTAATCGCCGACGAAACTACTGGTTCAAGTGTGACGAACGACCCACCCGCTCAACGACCTAAAGATCCAGCC
AAACCTCCGGCCAAGGCACAAGTTGTGGGATGGCCACCGGTGAGATCATACCGGAAGAACGTGATGGTTTCCTGCCAAAA
ATCAAGCGGTGGCCCGGAGGCGGCGGCGTTCGTGAAGGATTACAAGGACGATGACAAGTACCCATACGATGTTCCAGATT
ACGCTGATTACAAGGACGATGACAAGTACCCATACGATGTTCCAGATTACGCT

## B

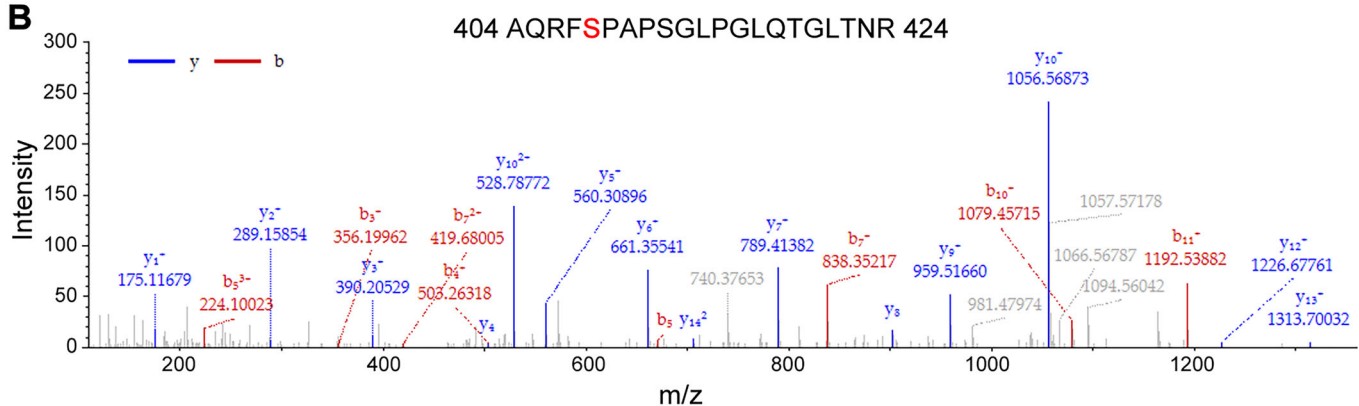

404 AQRF**S**PAPSGLPGLQTGLTNR 424

◄ **Figure EV4.  Creation of *sas-6::degron* and mutant strains by the CRISPR/Cas9 method and detection of SAS-6 phosphorylation in *C. elegans*.**

(A) Top: schematic diagram of genome editing to create the endogenously tagged *sas-6::degron::flag* strain. The sequence encoding the degron and FLAG tag is shown in pink. Depending on the context, the genotypes of this strain are labeled as *sas-6::degron::flag*, *sas-6::degron*, or *sas-6::flag* in this study. Middle and Bottom: schematic diagrams of genome editing to create the *sas-6(5A)* and *sas-6(5D)* mutants. Mutation sites are indicated in red, and inserted tagging sequences are shown in pink. The genotypes of these strains are labeled as *sas-6(5 A)* or *sas-6(5 A)::flag*, and *sas-6(5D)* or *sas-6(5D)::flag*, respectively, in this study. (B) Mass spectrometry identification of the SAS-6 peptide with Ser408 phosphorylation. Mass spectrometry analysis was performed with anti-FLAG immunoprecipitation of worm lysates from synchronized *sas-6::flag* young adults.

    

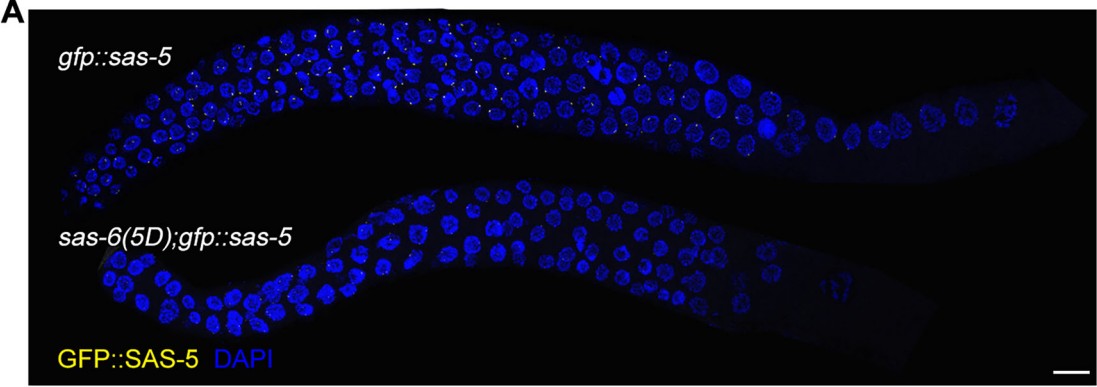

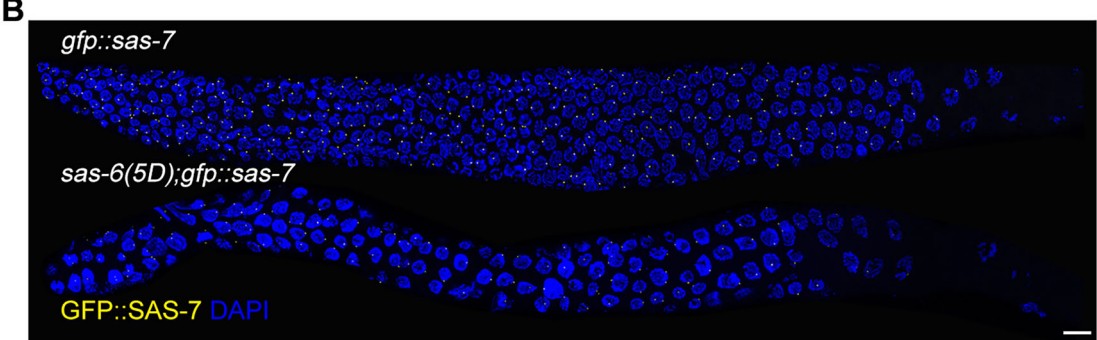

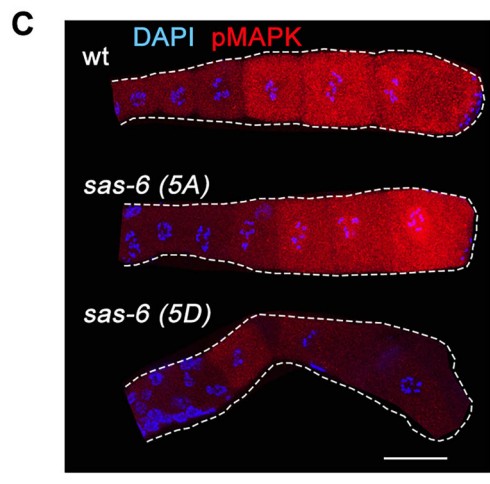

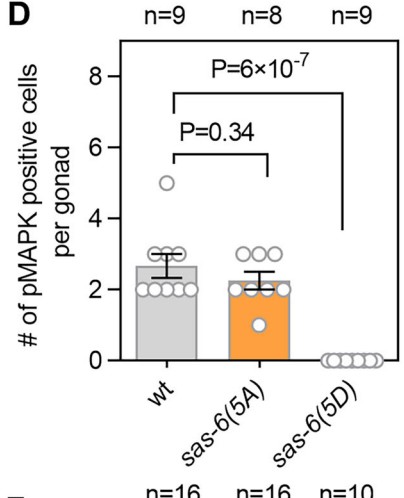

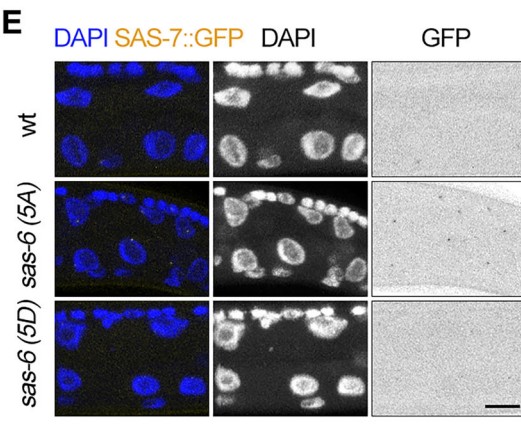

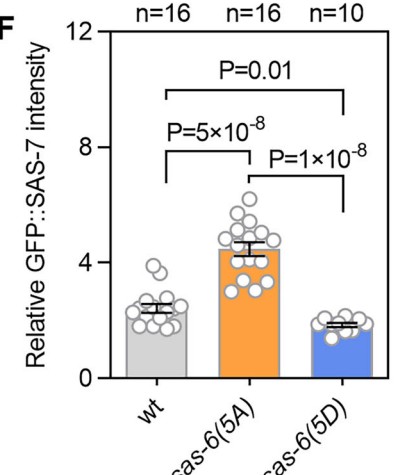

◀ **Figure EV5.** **Centrosome duplication and elimination in germline and intestine of *sas-6* mutant worms.**

(A) Projection images showing GFP::SAS-5 (yellow) focus formation in the germlines of the indicated genotypes. DNA was stained with DAPI (blue). Scale bar, 10 μm. (B) Projection images showing GFP::SAS-7 (yellow) focus formation in the germlines of the indicated genotypes. DNA was stained with DAPI (blue). Scale bar, 10 μm. (C, D) Immunofluorescence images (C) and quantification (D) of phosphorylated MAPK (pMAPK) positive oocytes per gonad arm. Quantification data are shown as mean ± SEM. *P* values were determined using two-tailed unpaired t-tests. The numbers of worms measured are indicated. (E) Maintenance of GFP::SAS-7 foci in intestinal cells of L2 worms with the indicated genotypes. Intestinal cells were distinguished from the other cells based on their location and nuclear morphology. (F) Quantification of the SAS-7 focus intensity in the intestine of the indicated genotype. Data are shown as mean ± SEM. Each dot represents the average fluorescence intensity of GFP::SAS-7 across all intestinal cells of a worm. *P* values were determined using two-tailed unpaired t-tests. The numbers of worms measured are indicated.

