## [Peer Review File · EMBO Reports]

Dynamic SAS-6 phosphorylation aids centrosome duplication and elimination in *C. elegans* oogenesis

Feifei Qi, Shanshan Yin, Xiangrui Yang, Ning Ju, Bohan Liu, Xing Zhang, Zixuan Zhu, Li Ji, Fuxin Zhang, Li Zhao, Ruoxi Wang, Min Liu, Liangran Zhang, Huijie Zhao, Jun Zhou, and Jinmin Gao

Corresponding author(s): Jinmin Gao (jinmingao@sdu.edu.cn) , Jun Zhou (junzhou@sdu.edu.cn)

Review Timeline:

Submission Date:	9th Aug 24
Editorial Decision:	8th Oct 24
Revision Received:	3rd Feb 25
Editorial Decision:	21st Mar 25
Revision Received:	11th Apr 25
Accepted:	9th May 25

Editor: Deniz Senyilmaz Tiebe

Transaction Report:

Dear Prof. Gao,

Thank you for submitting your research manuscript to our journal, which was now seen by three referees, whose reports are copied below.

I apologize for this unusual delay in getting back to you, it took longer than anticipated to receive the full set of referee reports.

Referees express interest in the proposed roles of CDK1 mediated SAS6 phosphorylation by CDK1 in centrosome elimination. However, they also raise significant concerns that need to be addressed for publication here. In particular, all referees raised concerns regarding the proposed direct causal link between SAS6 phosphorylation and centriole elimination. All referees make constructive suggestions to address this concern experimentally and/or textually. I am not going to detail their recommendations here as they seem straightforward. However, please do contact me if you would like to discuss any of the referee points further, also by video chat.

Should you be able to address the referee concerns satisfactorily, we would like to invite you to submit a revised manuscript. Please revise your manuscript with the understanding that the referee concerns (as in their reports) must be fully addressed and their suggestions taken on board. Please address all referee concerns in a complete point-by-point response. Acceptance of the manuscript will depend on a positive outcome of a second round of review. It is EMBO reports policy to allow a single round of major experimental revision only and acceptance or rejection of the manuscript will therefore depend on the completeness of your responses included in the next, final version of the manuscript.

We realize that it is difficult to revise to a specific deadline. In the interest of protecting the conceptual advance provided by the work, we recommend a revision within 3 months. Please discuss the revision progress ahead of this time with me if you require more time to complete the revisions, or if you have questions or comments regarding the revision (also by video chat).

1. A data availability section providing access to data deposited in public databases is missing (where applicable).
2. Your manuscript contains statistics and error bars based on $n=2$. Please use scatter plots in these cases.

You can submit the revision either as a Scientific Report or as a Research Article. For Scientific Reports, the revised manuscript can contain up to 5 main figures and 5 Expanded View figures, and it should not exceed 27000 characters. If the revision leads to a manuscript with more than 5 main figures it will be published as a Research Article. In this case the Results and Discussion section should be separate. If a Scientific Report is submitted, these sections have to be combined. This will help to shorten the manuscript text by eliminating some redundancy that is inevitable when discussing the same experiments twice. In either case, all materials and methods should be included in the main manuscript file.

3) We replaced Supplementary Information with Expanded View (EV) Figures and Tables that are collapsible/expandable online. A maximum of 5 EV Figures can be typeset. EV Figures should be cited as 'Figure EV1, Figure EV2' etc... in the text and their respective legends should be included in the main text after the legends of regular figures.

4) a .docx formatted letter INCLUDING the reviewers' reports and your detailed point-by-point responses to their comments. As

part of the EMBO publication's Transparent Editorial Process, EMBO reports publishes online a Review Process File (RPF) to accompany accepted manuscripts. This File will be published in conjunction with your paper and will include the referee reports, your point-by-point response and all pertinent correspondence relating to the manuscript.

<https://www.embopress.org/page/journal/14693178/authorguide#transparentprocess>

5) a complete author checklist, which you can download from our author guidelines

<https://www.embopress.org/page/journal/14693178/authorguide>. Please insert information in the checklist that is also reflected in the manuscript. The completed author checklist will also be part of the RPF.

6) Please note that all corresponding authors are required to supply an ORCID ID for their name upon submission of a revised manuscript (<<https://orcid.org/>>). Please find instructions on how to link your ORCID ID to your account in our manuscript tracking system in our Author guidelines

<<https://www.embopress.org/page/journal/14693178/authorguide#authorshipguidelines>>

7) Before submitting your revision, primary datasets produced in this study need to be deposited in an appropriate public database (see <https://www.embopress.org/page/journal/14693178/authorguide#datadeposition>). Please remember to provide a reviewer password if the datasets are not yet public. The accession numbers and database should be listed in a formal "Data Availability" section placed after Materials & Method (see also

<https://www.embopress.org/page/journal/14693178/authorguide#datadeposition>). Please note that the Data Availability Section is restricted to new primary data that are part of this study. * Note - All links should resolve to a page where the data can be accessed. *

Additional information on source data and instruction on how to label the files are available:

<https://www.embopress.org/page/journal/14693178/authorguide#sourcedata>

9) Our journal encourages inclusion of *data citations in the reference list* to directly cite datasets that were re-used and obtained from public databases. Data citations in the article text are distinct from normal bibliographical citations and should directly link to the database records from which the data can be accessed. In the main text, data citations are formatted as follows: "Data ref: Smith et al, 2001" or "Data ref: NCBI Sequence Read Archive PRJNA342805, 2017". In the Reference list, data citations must be labeled with "[DATASET]". A data reference must provide the database name, accession number/identifiers and a resolvable link to the landing page from which the data can be accessed at the end of the reference. Further instructions are available at <http://www.embopress.org/page/journal/14693178/authorguide#referencesformat>

10) Regarding data quantification (see Figure Legends:

<https://www.embopress.org/page/journal/14693178/authorguide#figureformat>)

11) The journal requires a statement specifying whether or not authors have competing interests (defined as all potential or

actual interests that could be perceived to influence the presentation or interpretation of an article). In case of competing interests, this must be specified in your disclosure statement. Further information: <https://www.embopress.org/competing-interests>

12) Please also note our reference format:

13) All Materials and Methods need to be described in the main text using our 'Structured Methods' format, which is required for all research articles. According to this format, the Methods section includes a Reagents and Tools Table (listing key reagents, experimental models, software and relevant equipment and including their sources and relevant identifiers) followed by a Methods and Protocols section describing the methods using a step-by-step protocol format. The aim is to facilitate adoption of the methodologies across labs. More information on how to adhere to this format as well as a downloadable template (.docx) for the Reagents and Tools Table can be found in our author guidelines:

I look forward to seeing a revised version of your manuscript when it is ready. Please let me know if you have questions or comments regarding the revision.

Kind regards,

Deniz Senyilmaz Tiebe

Deniz Senyilmaz Tiebe, PhD
Senior Scientific Editor
EMBO Reports

Referee #1:

This paper examines the role of phosphorylation of the centriole scaffold SAS-6 by the cell cycle kinase CDK-1 using a combination of in vitro, tissue culture and *C. elegans* experiments. The authors argue that SAS-6 phosphorylation prevents its assembly in centrioles and contributes to centrosome elimination during oogenesis. An alternative hypothesis, also consistent with their data however, is that SAS-6 phosphorylation prevents overduplication of centrosomes during the mitotic cell cycle and only indirectly extends the period during which centrosomes can be detected in meiotic oocytes before complete elimination.

The authors should acknowledge this alternative interpretation and revise their title or provide additional data to justify favoring one model over the other. The organization of the paper is also quite confusing going back and forth between biochemical experiments in vitro, cell biological experiments in cell culture and genetic/cell biological experiments in *C. elegans*. It may be preferable to keep each data type together and present them one after the other for better flow.

Specific comments:

1. How do the authors interpret the hexanediol wash out experiments? Is there a SAS-6 independent structure that can reassemble centrioles/recruit SAS-6 back during the washout or are centrioles reassembled de novo?
2. FRAP experiments are interpreted as SAS-6 being dynamic within the centriolar scaffold. Can the authors exclude the possibility that the dynamic pool of SAS-6 is actually not in centrioles but in some other centrosome-associated compartment such as the PCM???
3. Are all the phospho sites identified in human SAS6 conserved as S/T-P motifs in *C. elegans*?? Can the phosphor-specific SAS-6 antibody be used in vivo to visualize where/when SAS6 is phosphorylated in *C. elegans* gonads or tissue culture cells??
4. sas-6(5D) appears to cause lower levels of gfp:sas-5 and 7 throughout the germline. This mutation clearly also affects SAS-6 levels before the period where centrosomes are eliminated. This should be stated clearly.
5. Do SAS6(5A) droplets in tissue culture cells resist solubilization during M Phase?

Do the SAS-6 droplets localize with the centrosome? Are they affected by CDK1 inhibition (RO3306)? The relevance of SAS-6 condensation to its role in centriole assembly is not clear.

6. Fig. 5: Why does SAS-7 resist SAS-6 depletion more than SAS-5 ??? Is this consistent with a known hierarchy of centriole assembly?

7. Fig. 6J is confusing as it appears to show average centriole scoring for only a subset of oocytes??? I suggest showing the actual data for ALL oocytes.

8. Page 11 Line 35 -38: The claim that sas-6(5A) does not affect assembly seems incorrect since SAS-6(5A) appears MORE abundant from the transition zone onward (Fig. 7C). This should be mentioned in text and could explain the apparent delay in elimination: SAS-6(5A) causes an increase in centriolar material which then takes longer to eliminate in oocytes. Opposite is true for SAS-6(5D) which starts out lower and then disappears faster. The defects observed in the two phospho mutant arise BEFORE centrosome elimination and likely only indirectly affects centrosome elimination.

General question: Do the stats for the intensity analyses based on the mean of each experiment or on each individual point?

Referee #2:

This interesting study endeavors to provide new insights into the regulation of centriole stability and centrosome elimination. Centrosome elimination is an evolutionarily conserved process during meiosis and guarantees that the zygote inherits an appropriate number of centrioles, matching the DNA content. Yet, it is one of the processes for which the molecular mechanisms remain poorly understood, and *C. elegans* is an ideal model system to address this question. SAS-6 forms the centriole cartwheel and is essential for centriole biogenesis in every studied model.

In their experiments, Qi et al show that SAS-6 might undergo LLPS and that this behavior is driven by its unstructured C-terminus. This weak interaction is speculated to be needed for cartwheel stacking. Further, the authors show that CDK-1-mediated phosphorylation inhibits weak interactions and propose that this is needed for centrosome elimination.

Overall, this is a very good study on an interesting topic with some high-quality data presented. In my view the data, as presented, supports the idea that the C-terminus of SAS-6 undergoes LLPS, however, it does not sufficiently support the proposed model of cartwheel stacking and centrosome elimination. I have a few comments that I believe would make the study stronger.

Major points:

1. The impact on centrosome elimination (PA/RO3306/CDK-1RNAi/SAS-6_5A/degron) is assessed by the strength of the SAS-5 and SAS-7 signals. A low signal does not necessarily translate into a partially disassembled centriole. To make a convincing claim regarding disassembly/elimination authors should consider using ultrastructure-based methods. The best would be CLEM. Alternatively, TEM or expansion microscopy would reveal the true impact on disassembly.
2. For germlines that are treated with 1,6-Hexanendiol, SAS-5, and SAS-7 levels drop to a very low level after 5 min and recover within 10-15 min. Does this mean that centrioles can be disassembled or rebuilt so quickly? Most likely not. Again, here some ultrastructural evidence might be helpful to correlate signal intensities with structural changes. This kind of experiment might also reveal the timeline of disassembly once SAS-6 is destabilized.
3. Figures 6 and 7. I'm a bit surprised that the authors do not detect the SAS-7 signal further distal in the gonad (oocyte -3, -2), as it was published in Pierreon et al 2023. In general, the stage of the meiotic germ cells is assessed solely by the DNA condensation pattern and the position in the germline. This might be misleading, especially for the distal part in some mutant backgrounds, where sometimes mature oocytes pile up or DNA condenses differently. Authors should consider using an oocyte maturation marker such as RME and score the presence/absence of the centrosomes in relationship to this marker.
4. Centrioles are also eliminated in later embryos and the intestine. How is centriole elimination affected by SAS-6(5A) or CDK-1 downregulation in these settings? Perhaps authors can use one of these lineages to reveal the impact of SAS-6(5D) on centriole elimination.
5. Where in the germline is CDK-1 expressed? The authors use MPM2 staining as a proxy. Has this antibody been validated in worms? One could verify the antibody in the CDK-1-depleted embryos. Also, GFP-tagged CDK-1 strains have been published and could be used for CDK-1 expression (DOI: 10.1126/science.aao2840).

6. Where in the germline does CDK-1 phosphorylate SAS-6? The authors might want to make use of their phosphor-specific antibody to answer this question.

7. Figure 8I, the authors might want to comment on the M fragment of SAS-6 on the beads recruiting SAS-4. Was this tested in the pull-downs with and without ATP? How does this fit into the model? It would also be helpful to show the SAS-4 bead recruitment assay with and without ATP for all conditions.

Minor points:

- Page 6/line 11 the authors propose that SAS-6 foci 'disperse' over time. Instead, these foci could also form de novo, rather than 'dispersing'. Please refer to similar SAS-6 foci observed in Pierron 2023.
- Figures 1D/C, it is not clear how the quantifications were performed, the authors might consider including a small diagram of how the measurements were done or explain it better in the text.
- In Figures 1G/I and others, a faint white dot on the black background is hardly detectable. The authors could reverse the colors and use a white background with a black signal.
- Figure EV1D, why are the actin levels so different?
- Figure 1F/EV1E, 'Clustering and separation' of the centrioles in the germline is not very convincing. Can this be quantified? Is this put into relationship nuclei? Overall, this experiment and the assessment of the number of centrioles in the germline do not add much to the building of the model, they can be moved to the supplemental data or omitted.
- Page 7/line 8, Figure EV1A is a typo
- Figure EV2D, could authors show the endogenous centrioles as a control? Does the SAS-6 blob localize close to the human centrioles?
- Can SAS-6 in HEK cells recruit SAS-5 or SAS-4?
- Page 9/line 6,39 and others, the term 'in vivo' is used for worms, cells are referred to as 'in vitro' system, this is not entirely correct...
- Page 9/12 'Figure EV4B' is incorrect
- Figure 5 C-F, centriole disassembly should be quantified also for diplotene, where this process takes place.
- Figure 6 A would benefit from insets and a better explanation/schematic of how the intensities are measured
- Figure 7 H, could authors show a zoomed-in picture of the centrioles imaged by SIM? Is there any structural difference detected by SIM in the sas-6(5D) centrioles?
- Figure 8, the authors use many different combinations of fragments for their interaction studies. To make it more reader-friendly authors might want to show schematics for the deletion fragments and combinations used in different settings.
- Figure 9, model oligolymer should be oligomer
- Figure 9, in the model the authors might want to incorporate the change in charge upon CDK-1 phosphorylation

Referee #3:

The manuscript under consideration by Qi et.al. aims to shed light on the mechanisms driving centrosome elimination during oogenesis, using in vivo experiments in *C. elegans* in combination with in vitro experiments and studies in human cells. The authors provide evidence that the centriole cartwheel protein SAS-6 forms droplets when overexpressed in human cells and can phase separate in vitro. They then went on to find sites in SAS-6 that are phosphorylated in vivo, and demonstrated that mutation of these sites to alanine delayed centriole elimination while phospho-mimetic mutations reduced the fluorescence intensity of centriole components, suggesting centriole defects. Moreover, the authors found that inhibition of CDK1 delayed centriole disassembly and that CDK1 can phosphorylate SAS-6 in vitro. Putting these findings together, the authors propose that CDK1 phosphorylates SAS-6 to promote centriole disassembly.

This paper contains interesting findings, and most of the experiments are well-performed and rigorous (a few exceptions, where further quantification is needed, are listed in the specific points below). However, the weakest part of the manuscript relates to the claim that these findings shed light on the mechanisms driving centrosome elimination during oogenesis. The effects on centrosome elimination that were examined in vivo (in the SAS-6 mutants or following CDK-1 inhibition), were very mild - there was a slight delay in elimination with CDK1 inhibitors/RNAi and in the SAS-6 5A mutant, but centrioles were still eliminated. Moreover, since there are likely other effects on the germline due to inhibiting CDK1 (a major cell cycle kinase), I didn't think that those effects could be definitively linked to SAS-6 phosphorylation. In addition, the SAS-6 5D mutant experiments were hard to interpret, since the germlines were small and abnormal, and the intensity of SAS-5 and SAS-7 foci was reduced throughout the germline (implying that the centrioles did not form properly - thus, this mutant could not be used to assay the timing of centriole elimination).

Putting all of the data from the paper together, I think that it is fine for the authors to PROPOSE the intriguing model that CDK1 promotes centriole elimination (in the discussion). However, stating this as a conclusion of the paper (in the title and abstract) is too strong based on the data presented. Thus, while this paper contains interesting findings that are likely to be of interest to researchers in the cell division field, the manuscript needs to be substantially revised so that the conclusions drawn are better supported by the experiments presented (either adding more data to support the conclusions, or revising the conclusions to better match the presented data).

Specific points:

- Figure EV1D: it is stated in the text that total protein levels of centriolar components don't change following the temperature shift (page 6 line 10), but the western blot samples were from whole worms (not only from the late prophase cells shown in Figure EV1C that are being described). Protein levels could drop in oocytes, but not in the rest of the worm (so a decrease in the germline could be masked).
- Figure EV1F: it is stated that centrioles exhibit dynamic movements but only two examples are shown and the centrioles in these examples have only minimal changes over time, so this is not very convincing. The authors would need more examples and quantification to make this claim.
- The results in Figure EV2A/B are somewhat variable, depending on the protein and the tag. For example, SPD-2 looks like it forms clear aggregates with mCherry but not GFP. The opposite is true for SAS-4 and SAS-5, calling the results in these figure panels into question. Since no quantification is provided, it is not clear if these images were the rare cases where such aggregates are found. The variability makes me wonder if the observed protein aggregation is really physiological, versus something that can occasionally happen when proteins are overexpressed in cells.
- Figure 2I: the authors claim that this figure shows that droplets diffuse during mitosis, but this is not convincing from the data presented. Only one example is shown - how frequently does this occur? In what proportion of cells? This result should be quantified somehow if the authors wish to make this claim. Related to this point, Page 8, Line 30 uses the phrasing "the observation that SAS-6 droplets diffused...". If the authors don't provide more convincing evidence to support this claim, this statement should be modified.
- The fact that the authors generated a phospho-specific antibody against Ser408 and confirmed that this residue is phosphorylated *in vivo* via western blotting (Figure 4B) is a nice result, but it doesn't show that this modification is present in the germline. Page 9, lines 3-4 states that since the lysate preparation protocol used to generate samples for the western blot does not disrupt embryos, the identified modification (phospho-Ser408) likely takes place in the germline. However, aren't centrosomes present in somatic cells? If so, this rationale does not make sense, as the modification detected on the western blot could have come from non-germline cells. If the authors wish to state that the modification is present in the germline more evidence would be needed, beyond just the western blot. Does this antibody recognize centrioles in the germline (via immunofluorescence)? Unless this is demonstrated (or other additional evidence is provided), the conclusion that SAS-6 is phosphorylated in the germline is not convincing.
- Page 9, section beginning on line 36: This section argues that "CDK-1 mediated phosphorylation inhibits phase separation of SAS-6", but the data do not show this. The phospho-mutant analysis suggests that phosphorylation affects phase separation (but does not shed light on which kinase regulates the ability of SAS-6 to phase separate). The demonstration that CDK-1 can phosphorylate SAS-6 *in vitro* suggests that this might be the case, but does not prove it. The language in this section and in the section header should therefore be softened ("our data are consistent with the hypothesis that...etc." rather than stating "our observations reveal that", which is too strong).
- I was confused by the interpretation of Figure 6A, where the authors are using MPM2 staining as a readout of the phosphorylation levels of CDK1 substrates (to determine where CDK1 activity peaks in the germline). It is clear that the MPM2 signal decreases with the PA inhibitor, but it is not gone. So can MPM2 really be used as a readout of CDK-1 activity, since other kinases also contribute to the MPM2 signal? This weakens the argument made by the authors that CDK1 activity peaks at the end of pachytene (where, in their model, it acts to trigger centrosome disassembly).
- Page 11 line 38-39 and page 12 lines 1-2: it is stated that foci formed by the SAS-6 5D mutant are weaker at the transition zone and become almost undetectable in pachytene, leading the authors to suggest that "sustained phosphorylation of SAS-6 could lead to premature elimination of centrosomes". However, the quantification in Figure 7C makes it look like the intensity of foci is similar between the TZ and pachytene, suggesting that these centrioles are not being prematurely eliminated, but that the centrioles are simply weaker/defective throughout the germline (i.e. the centrioles were never properly formed and therefore the signal is weak in the entire germline). To claim that centrioles are being prematurely eliminated, the authors would need to provide stronger evidence.
- For the bead recruitment assays in Figure 8, I was not sure how to interpret the negative results (e.g. the lack of recruitment of the M and C proteins). Couldn't this be because the truncated mutants do not properly fold? Some evidence to support that these proteins are properly folded would help support the results.

Minor/typos:

- Page 6, Line 22: should this be "duplicated centrioles" (not centrosomes)?
- The labeling on Figures 8E and 8G is confusing. The columns are labeled + or - ATP, but if I understand correctly these conditions actually denote whether the proteins were preincubated with CDK1 (according to the figure legend). Editing the labels would help clarity.

EMBOR-2024-60173V2

Dynamic SAS-6 phosphorylation aids centrosome duplication and elimination in *C. elegans* oogenesis

Feifei Qi, Xiangrui Yang, Shanshan Yin, Ning Ju, Bohan Liu, Xing Zhang, Li Ji, Fuxin Zhang, Li Zhao, Ruoxi Wang, Min Liu, Liangran Zhang, Huijie Zhao, Jun Zhou, and Jinmin Gao

Response to Reviewers' Comments:

We appreciate the reviewers' valuable comments. In response, we have reorganized the article framework, revised the title and abstract, made extensive modifications to the text and figures, and ensured that our conclusions more accurately reflect the data. The title has been updated to "Dynamic SAS-6 phosphorylation aids centrosome duplication and elimination in *C. elegans* oogenesis." In the Results section, we have restructured the data as follows:

1. Centrioles are disassembled during late meiotic prophase
2. Cartwheel proteins have dynamic properties *in vivo*
3. SAS-6 undergoes phase separation *in vitro*
4. SAS-6 forms droplets in cells
5. SAS-6 is phosphorylated in cells and in *C. elegans*
6. CDK-1 directly phosphorylates SAS-6 *in vitro*
7. Phosphorylation inhibits the phase separation of SAS-6
8. SAS-6 phosphorylation attenuates interactions between centriolar proteins
9. Human SAS-6 C-terminus exhibits phase-separate properties regulated by phosphorylation
10. SAS-6 promotes centrosome stability in *C. elegans* germline
11. CDK-1 promotes timely centriole disassembly during oogenesis
12. Dynamic regulation of SAS-6 phosphorylation is crucial for centrosome assembly and subsequent elimination during oogenesis

In line with the improved manuscript organization and the reviewers' suggestions, we have removed several results that do not significantly contribute to our major findings. These include the analysis of SAS-6 expression under 27°C heat shock conditions, live-cell imaging of centrosome movements during early prophase, *in vitro* kinase assays comparing the kinase activity of human CDK1 and *C. elegans* CDK-1, and the examination of aggregate formation of different centrosome proteins in mammalian cells.

The revised manuscript now contains 10 main figures, 5 Expansion View Figures, and 5 Appendix Figures. Below please find the comments from the reviewers in black and our responses in blue.

Referee #1:

This paper examines the role of phosphorylation of the centriole scaffold SAS-6 by the cell cycle kinase CDK-1 using a combination of in vitro, tissue culture and *C. elegans* experiments. The authors argue that SAS-6 phosphorylation prevents its assembly in centrioles and contributes to centrosome elimination during oogenesis. An alternative hypothesis, also consistent with their data however, is that SAS-6 phosphorylation prevents overduplication of centrosomes during the mitotic cell cycle and only indirectly extends the period during which centrosomes can be detected in meiotic oocytes before complete elimination.

The authors should acknowledge this alternative interpretation and revise their title or provide additional data to justify favoring one model over the other.

We agree with the hypothesis proposed. Based on our data, we believe that the reviewer's suggestion—that phosphorylation of SAS-6 prevents excessive centrosome duplication in pre-meiotic oocytes, thereby influencing centrosome elimination during oogenesis—is reasonable. We currently lack evidence to support the notion that SAS-6 phosphorylation directly promotes centrosome elimination.

To demonstrate that SAS-6 phosphorylation is a causal factor in centrosome elimination, we would need to create a worm strain that induces endogenous SAS-6 knockout/degradation during early meiotic prophase, while simultaneously expressing the phosphor-dead SAS-6 (5D) protein. However, constructing such a worm strain presents significant technical challenges.

In light of the reviewer's valuable suggestions, we revised our title as "Dynamic SAS-6 phosphorylation aids centrosome duplication and elimination in *C. elegans* oogenesis" and hypothesis accordingly.

The organization of the paper is also quite confusing going back and forth between biochemical experiments in vitro, cell biological experiments in cell culture and genetic/cell biological experiments in *C. elegans*. It may be preferable to keep each data type together and present them one after the other for better flow.

To improve the flow, we have reorganized the order of the data as presented earlier. Thank you for the suggestion.

Specific comments:

1. How do the authors interpret the hexanediol wash out experiments? Is there a SAS-6 independent structure that can reassemble centrioles/recruit SAS-6 back during the washout or are centrioles reassembled de novo?

The mature centrosome is a membrane-less organelle composed of a pair of centrioles and PCM. The components of these structures exhibit different dynamic properties. For instance, Cep57 and Cep63 display dynamic behavior, while others, such as the acetylated and glutamylated microtubule walls of the centrioles, are likely stable and serve as structural elements that provide stability. Hexanediol treatment likely disrupt only the dynamic components away from the centrosome, leaving the scaffold components largely intact, including centriole microtubule wall and PMC components. This preservation allows for the recruitment of dynamic components following hexanediol washout.

Consistently, treatment of U2OS cells with 1,6-hexanediol results in diminished signals for Cep63 and Cep152, whereas glutamylated tubulin in the centriolar microtubules remains unaffected (Ahn et al., 2020, PMID: 33208041).

Additionally, using ultrastructure expansion and stimulated emission depletion (U-Ex-STED) microscopy, we confirmed the retention of centriole microtubule structures following hexanediol treatment. These data are presented in Fig. 2G and H and cited in the last paragraph on page 5:

“This suggests that different centrosome components may have different dynamic properties. The ability of cartwheel proteins to reassemble implies that certain substructures of the centriole may remain stable and intact during 1,6-hexanediol exposure, providing the structural framework for the reassembly of other dynamic components after hexanediol washout. Consistent with this, we observed that the microtubule core structure of the centriole remained unchanged following 1,6-hexanediol treatment, as visualized by ultrastructure expansion and stimulated emission depletion (U-Ex-STED) microscopy (Fig 1G and H).”

2. FRAP experiments are interpreted as SAS-6 being dynamic within the centriolar scaffold. Can the authors exclude the possibility that the dynamic pool of SAS-6 is actually not in centrioles but in some other centrosome-associated compartment such as the PCM???

Previous studies using STED, U-ExM, and U-Ex-STED microscopy have shown that SAS-6 is exclusively detected in the centrioles of human cells and *C. elegans* germline (Yoshida et al., 2019, PMID: 31164447; Le Guennec et al., 2020, PMID: 32110738; Woglar et al., 2022, PMID: 36107993). To our knowledge, no evidence has been reported indicating a significant pool of SAS-6 in the PCM. Therefore, the FRAP signals are unlikely to result from SAS-6 localized within the PCM.

To further investigate this, we performed expansion microscopy to examine the localization of SAS-6 and the PCM protein SAS-7. Consistent with previous findings, we did not detect SAS-6 within the PCM. These data are now presented in Fig. EV1G and cited in the text on page 6, lines 5-9:

“In line with previous reports (Le Guennec, Klena et al., 2020, Woglar et al., 2022, Yoshida, Tsuchiya et al., 2019), our expansion microscopy analysis suggests that SAS-6 does not have a major pool within the PCM (Fig EV1G). Thus, we propose that the observed dynamics of these proteins reflect the behavior of cartwheel proteins within the centriole.”

3. Are all the phospho sites identified in human SAS6 conserved as S/T-P motifs in *C. elegans*??

This is discussed in the text on page 10, lines 37-39:

“Within the SASS6 C-terminus, two putative phosphorylation sites, Thr495 and Ser510, are present, both located within S/T-P motifs, resembling the phosphorylation sites found in *C. elegans* SAS-6.”

-Can the phosphor-specific SAS-6 antibody be used in vivo to visualize where/when SAS6 is phosphorylated in *C. elegans* gonads or tissue culture cells?

Although we attempted to generate multiple phospho-specific SAS-6 antibodies from two different companies, we were unable to obtain one suitable for immunostaining in *C. elegans* gonads. As a

result, we currently lack the necessary reagent to confirm SAS-6 phosphorylation by immunofluorescence in the germline.

However, the Ser408 phosphorylation-specific antibody works for immunostaining in cultured cells with overexpressed SAS-6. The antibody specifically stained the diffuse SAS-6 in mitotic cells, showed weak staining in SAS-6 condensates formed in interphase cells, and did not stain SAS-6(5A) droplets. These data are presented in Appendix Fig. S1A and C and cited in the first paragraph on page 9:

“Consistently, immunofluorescence microscopy revealed that the pSer408 antibody robustly detected overexpressed SAS-6 in mitotic cells but not in cells rest in interphase cells or in cells expressing SAS-6(5A) mutant (Appendix Fig S1A and B). Furthermore, treatment of mitotic cells with RO3306 led to the formation of SAS-6 droplets and a near-complete loss of the pSer408 immunostaining signal (Appendix Fig S1C).”

4. sas-6(5D) appears to causes lower levels of gfp:sas-5 and 7 throughout the germline. This mutation clearly also affects SAS-6 levels before the period where centrosomes are eliminated. This should be stated clearly.

We have modified the text as suggested on page 12, lines 12-16:

“Cytological analysis revealed that SAS-6 foci in sas-6(5A) mutants exhibited stronger fluorescence intensity than wild-type, indicating excessive centriole assembly (Fig 9B and C). In contrast, SAS-6(5D) foci displayed reduced intensity throughout the germline and were nearly undetectable by late pachytene (Fig 9B and C), suggesting that sustained phosphorylation may impair centriole assembly and duplication.”

5. Do SAS6(5A) droplets in tissue culture cells resist solubilization during M Phase? Do the SAS-6 droplets localize with the centrosome? Are they affected by CDK1 inhibition (RO3306)?

This is an interesting question, and we have examined the droplet formation of the SAS-6(5A) mutant in both interphase and mitotic cells. Indeed, these droplets resist solubilization during M phase. Furthermore, adding a CDK1 inhibitor induced the formation of SAS-6 droplets in M phase. These data are now presented in Appendix Fig. S1A-C and cited in the first paragraph on page 9:

“Consistently, immunofluorescence microscopy revealed that the pSer408 antibody robustly detected overexpressed SAS-6 in mitotic cells but not in cells rest in interphase cells or in cells expressing SAS-6(5A) mutant (Appendix Fig S1A and B). Furthermore, treatment of mitotic cells with RO3306 led to the formation of SAS-6 droplets and a near-complete loss of the pSer408 immunostaining signal (Appendix Fig S1C).”

The relevance of SAS-6 condensation to its role in centriole assembly is not clear.

We now provide an explanation at the beginning of the Discussion on page 13:

“Our findings reveal the properties of SAS-6 across *in vitro*, cellular, and *C. elegans* germline contexts (Fig 10). Purified SAS-6 undergoes phase separation *in vitro* and forms droplets in cells when overexpressed. These droplets are large and disorganized, with the unstructured C-terminal region

playing a key role in mediating weak interactions that drive phase separation. In contrast, within the centriole *in vivo*, SAS-6 forms a highly ordered structure, which we propose also relies on multivalent weak interactions mediated by the C-terminal domain. In both contexts, phosphorylation at the C-terminal domain alters the behavior of SAS-6, affecting its phase separation *in vitro* and cartwheel assembly *in vivo*. While these processes exhibit distinct forms of organization, they share similar amino-acid-based interaction mechanisms. The functional role of the C-terminal domain provides insights into cartwheel stacking, centrosome elimination during oogenesis, and potentially other biological processes involving centrosome elimination.”

6. Fig. 5: Why does SAS-7 resist SAS-6 depletion more than SAS-5 ??? Is this consistent with a known hierarchy of centriole assembly?

This is explained on page 11, lines 20-23:

“The depletion of SAS-6 had a more pronounced effect on SAS-5, likely due to the direct interaction between SAS-5 and the coiled-coil domain of SAS-6 (Qiao, Cabral et al., 2012), whereas SAS-7 is primarily localized to the PCM (Woglar et al., 2022) (Fig EV1G).”

7. Fig. 6J is confusing as it appears to show average centriole scoring for only a subset of oocytes???. I suggest showing the actual data for ALL oocytes.

Our labeling may not have been entirely straightforward. We previously presented data for oocytes at all positions but did not label all of them due to space constraints. In the updated figures, we now label all oocyte positions in Figures 7 and 8. However, in Figure 9, space constraints still prevent us from labeling all positions, though all oocyte data are shown.

8. Page 11 Line 35 -38: The claim that sas-6(5A) does not affect assembly seems incorrect since SAS-6(5A) appears MORE abundant from the transition zone onward (Fig. 7C). This should be mentioned in text and could explain the apparent delay in elimination: SAS-6(5A) causes an increase in centriolar material which then takes longer to eliminate in oocytes. Opposite is true for SAS-6(5D) which starts out lower and then disappears faster. The defects observed in the two phospho mutant arise BEFORE centrosome elimination and likely only indirectly affects centrosome elimination.

We appreciate the reviewer’s comments and have revised the text accordingly on page 12, lines 12-16, 18-20, and 31-34:

“Cytological analysis revealed that SAS-6 foci in *sas-6(5A)* mutants exhibited stronger fluorescence intensity than wild-type, indicating excessive centriole assembly (Fig 9B and C). In contrast, SAS-6(5D) foci displayed reduced intensity throughout the germline and were nearly undetectable by late pachytene (Fig 9B and C), suggesting that sustained phosphorylation may impair centriole assembly and duplication.”

“In *sas-6(5A)* mutants, GFP::*SAS-5* and GFP::*SAS-7* foci persisted into diakinesis, indicating delayed centrosome elimination, likely due to centriole overduplication in pre-meiotic cells (Fig 9D-G, EV5A and B).”

“The intensity of GFP::SAS-5 and GFP::SAS-7 foci was significantly reduced in *sas-6(5D)* mutants, with GFP::SAS-5 foci nearly absent by late pachytene (Fig 9I-L). Premature centrosome elimination may result from reduced centriolar material assembly in pre-meiotic cells.”

General question: Do the stats for the intensity analyses based on the mean of each experiment or on each individual point?

Statistical analysis of intensity was performed using data from individual cells, and the quantification method has been updated to present a single data point for each gonad or independent replicate.

Referee #2:

This interesting study endeavors to provide new insights into the regulation of centriole stability and centrosome elimination. Centrosome elimination is an evolutionarily conserved process during meiosis and guarantees that the zygote inherits an appropriate number of centrioles, matching the DNA content. Yet, it is one of the processes for which the molecular mechanisms remain poorly understood, and *C. elegans* is an ideal model system to address this question. SAS-6 forms the centriole cartwheel and is essential for centriole biogenesis in every studied model.

In their experiments, Qi et al show that SAS-6 might undergo LLPS and that this behavior is driven by its unstructured C-terminus. This weak interaction is speculated to be needed for cartwheel stacking. Further, the authors show that CDK-1-mediated phosphorylation inhibits weak interactions and propose that this is needed for centrosome elimination.

Overall, this is a very good study on an interesting topic with some high-quality data presented. In my view the data, as presented, supports the idea that the C-terminus of SAS-6 undergoes LLPS, however, it does not sufficiently support the proposed model of cartwheel stacking and centrosome elimination. I have a few comments that I believe would make the study stronger.

We appreciate the reviewer’s positive overall assessment of our work and the constructive comments.

Major points:

1. The impact on centrosome elimination (PA/RO3306/CDK-1RNAi/SAS-6_5A/degron) is assessed by the strength of the SAS-5 and SAS-7 signals. A low signal does not necessarily translate into a partially disassembled centriole. To make a convincing claim regarding disassembly/elimination authors should consider using ultrastructure-based methods. The best would be CLEM. Alternatively, TEM or expansion microscopy would reveal the true impact on disassembly.

We currently lack access to CLEM, and our electron microscope is not capable of performing 50 nm serial sectioning. Therefore, we attempted expansion microscopy coupled with STED, which we have worked hard to set up in the lab over the past few months. We were fortunate to detect centriole microtubules using this technology with an anti-tubulin antibody (Fig. 1G). However, we still need to optimize conditions for co-immunostaining of different centrosome proteins to correlate reduced signal with centriole disassembly.

We would also like to note that in Professor Pierre Gönczy's recent publication on centrosome elimination in *C. elegans* oocytes, the fluorescence intensity of centriolar proteins begins to decrease in diplotene, a stage when the central tube becomes undetectable (Pierron et al., 2023, PMID: 37987153). This observation suggests that the reduction in fluorescence intensity is linked to centriole disassembly.

Due to time constraints, we would like to explain our experimental limitations in the Results section on page 5, lines 7-11:

“The progressive loss of foci enriched with centriolar and PCM components during late meiotic prophase aligns with a gradual process of centriole elimination (Pierron et al., 2023). Therefore, we use changes in the intensity of centrosome protein foci as indicators of the centrosome elimination process, although the actual disassembly of centrioles and centrosomes may occur only after reaching a specific threshold.”

2. For germlines that are treated with 1,6-Hexanediol, SAS-5, and SAS-7 levels drop to a very low level after 5 min and recover within 10-15 min. Does this mean that centrioles can be disassembled or rebuilt so quickly? Most likely not. Again, here some ultrastructural evidence might be helpful to correlate signal intensities with structural changes. This kind of experiment might also reveal the timeline of disassembly once SAS-6 is destabilized.

This comment closely resembles comment #1 from Referee 1. Please refer to our earlier response.

We have performed ultrastructure expansion and stimulated emission depletion (U-Ex-STED) microscopy, which revealed the retention of centriole microtubule structures under hexanediol treatment. These data are presented in Fig. 2G. Additionally, we have included a cartoon to help illustrate the mechanisms underlying the removal of dynamic components by 1,6-hex and their reinstallation after 1,6-hex washout. This result is cited in the text on page 5, the last paragraph:

“This suggests that different centrosome components may have different dynamic properties. The ability of cartwheel proteins to reassemble implies that certain substructures of the centriole may remain stable and intact during 1,6-hexanediol exposure, providing the structural framework for the reassembly of other dynamic components after hexanediol washout. Consistent with this, we observed that the microtubule core structure of the centriole remained unchanged following 1,6-hexanediol treatment, as visualized by ultrastructure expansion and stimulated emission depletion (U-Ex-STED) microscopy (Fig 1G and H).”

3. Figures 6 and 7. I'm a bit surprised that the authors do not detect the SAS-7 signal further distal in the gonad (oocyte -3, -2), as it was published in Pierron et al 2023. In general, the stage of the meiotic germ cells is assessed solely by the DNA condensation pattern and the position in the germline. This might be misleading, especially for the distal part in some mutant backgrounds, where sometimes mature oocytes pile up or DNA condenses differently. Authors should consider using an oocyte maturation marker such as RME and score the presence/absence of the centrosomes in relationship to this marker.

We appreciate the suggestion. The differences in the duration of the SAS-7 signal detected during late meiotic prophase in our study and the work from Professor Gönczy's lab may be related to variations in the microscopy detection limits or thresholds, which can differ between microscopy

settings. Since we do not have *rme-2::gfp* worms, we used MAPK-YT (Achache et al., 2021, PMID: 34061407) as a marker to observe oocyte maturation in wild-type, *sas-6(5A)*, and *sas-6(5D)* mutant worms. Overall, oocyte activation was not affected in the *sas-6(5A)* mutant background. However, no oocyte activation was detected in the *sas-6(5D)* mutants. Given that centrosome elimination corresponds to the stage of oocyte activation, the premature elimination of the centrosome in *sas-6(5D)* mutants is likely not due to premature oocyte activation. This result is now presented in Fig. EV5F and G, and cited in the text on page 12, lines 22-25 and 34-36:

“Furthermore, this delay in elimination was not due to changes in the timing of oocyte maturation, as indicated by the unchanged number of oocytes positively stained for phosphorylated MAPK, a marker of matured oocytes (Lee, Ohmachi et al., 2007) (Fig EV5C and D).”

“Since oocytes in *sas-6(5D)* mutants were not activated (Fig EV5C and D), this premature centrosome elimination cannot be attributed to early oocyte activation.”

4. Centrioles are also eliminated in later embryos and the intestine. How is centriole elimination affected by SAS-6(5A) or CDK-1 downregulation in these settings? Perhaps authors can use one of these lineages to reveal the impact of SAS-6(5D) on centriole elimination.

We agree that investigating centriole elimination in later embryos and the intestine would provide valuable insights into the impact of SAS-6/CDK-1 regulation on centriole elimination across different biological processes. However, due to the extensive work required and the added complexity of analyzing centriole elimination in later embryos—such as the need for long-term imaging and quantification of hundreds of nuclei per embryo—we have not yet reached a conclusion from our preliminary data in embryos.

Nevertheless, we were able to examine the centrosome elimination process in the intestine of wild-type, *sas-6(5A)*, and *sas-6(5D)* mutants by analyzing GFP::SAS-7, as performed in a previous study from Professor Gönczy’s lab (Sci Adv. 2023 May 31;9(22):eadg8682). Similar to our observations in oocytes, we found an enhanced SAS-7 signal in the *sas-6(5A)* intestine and premature elimination in the *sas-6(5D)* intestine. This suggests that SAS-6 regulation impacts centrosome elimination in various biological processes. We now present this data in Fig. EV5D and E, and cite it in the text on page 13, lines 5-7:

“Furthermore, this regulatory mechanism may extend beyond oogenesis, as evidenced by the altered elimination timing observed in SAS-6 phospho-mutants in the intestine (Fig EV5E and F).”

5. Where in the germline is CDK-1 expressed? The authors use MPM2 staining as a proxy. Has this antibody been validated in worms? One could verify the antibody in the CDK-1-depleted embryos. Also, GFP-tagged CDK-1 strains have been published and could be used for CDK-1 expression (DOI: 10.1126/science.aao2840).

As suggested, we have examined CDK-1 expression in the germline and validated MPM2 in a *cdk-1* mutant. Using two available transgenic worm strains expressing GFP-tagged CDK-1 and MYC-tagged CDK-1, we observed nuclear enrichment of CDK-1 starting in late pachytene, with increasing abundance towards the end of meiotic prophase. The analysis of CDK-1 expression is presented in Appendix Fig. S5A and B, and is cited in the text on page 11, lines 28-31:

“To investigate whether CDK-1 activity is required for centrosome elimination, we analyzed the expression and activity dynamics of CDK-1. Using strains expressing tagged CDK-1, we observed CDK-1 accumulation in the nuclei of late prophase oocytes, a pattern coinciding with centrosome disappearance (Appendix Fig S5A and B).”

MPM2 has previously been used as a readout of CDK-1 activity in wild-type worms (Belew et al., 2023, PMID: 37478128). In this study, we further validate the MPM2 antibody in a *cdk-1* mutant and observe a reduction in MPM2 signal, particularly during late meiotic prophase, which is consistent with our analysis using a CDK-1 inhibitor. The new data is presented in Appendix Fig. S5C and cited in the text on page 11, lines 35-39:

“In *cdk-1(ne2257)* mutants (with partially impaired kinase activity) and worms treated with the CDK-1 inhibitor purvalanol A (PA), MPM-2 signal was reduced but not abolished, indicating that CDK-1 contributes to the MPM-2 signal (Figs 8A, B and Appendix Fig S5C). These findings suggest that CDK-1 activity is activated during oogenesis.”

6. Where in the germline does CDK-1 phosphorylate SAS-6? The authors might want to make use of their phospho-specific antibody to answer this question.

As mentioned in our response to Reviewer #1, our phospho-specific antibody staining did not work for immunofluorescence in the germline, and we are currently unable to assess this. We acknowledge the limitation in the results section on page 8, at the end of the first paragraph:

“These results confirmed that SAS-6 is phosphorylated *in vivo* at Ser408. Although the five putative phosphorylation sites could not be directly confirmed in the germline, analysis of CDK-1 expression patterns and additional evidence suggest that they are likely phosphorylated in the germline, where they may together perform a regulatory function.”

7. Figure 8I, the authors might want to comment on the M fragment of SAS-6 on the beads recruiting SAS-4. Was this tested in the pull-downs with and without ATP? How does this fit into the model? It would also be helpful to show the SAS-4 bead recruitment assay with and without ATP for all conditions.

In Fig. 5J, we have replaced the bead recruitment assay data with a new dataset, including results on the effect of SAS-6 phosphorylation on SAS-4 recruitment. This is cited in the text on page 10, lines 11-15, with an explanation added for the M fragment:

“Moreover, GFP-SAS-4 recruitment was impaired when FL SAS-6 and SAS-6(Δ N) were pre-phosphorylated (Fig 5I and J). A weak interaction between SAS-4 and the SAS-6 M domain was detected, which remained unaffected by pre-phosphorylation (Fig 5I). We speculate that this interaction might not occur in FL SAS-6 due to potential steric hindrance from the terminal domains.”

Although we tested all SAS-6 fragments, due to space constraints, we only present pre-phosphorylated FL, M, and SAS-6(Δ N) fragments that show detectable interaction without phosphorylation. The other fragments did not show changes in their SAS-4 recruitment after pre-phosphorylation.

Minor points:

- Page 6/line 11 the authors propose that SAS-6 foci 'disperse' over time. Instead, these foci could also form de novo, rather than 'dispersing'. Please refer to similar SAS-6 foci observed in Pierron 2023.

The term 'disperse' was originally used to describe phenotypes associated with SAS-6 foci formed following heat shock. To make the manuscript clearer and more straightforward, we have removed this data.

- Figures 1D/C, it is not clear how the quantifications were performed, the authors might consider including a small diagram of how the measurements were done or explain it better in the text.

A diagram illustrating how the measurements were performed is provided in Fig. 1C, with additional details on quantification and statistical analysis described in the methods section.

- In Figures 1G/I and others, a faint white dot on the black background is hardly detectable. The authors could reverse the colors and use a white background with a black signal.

Thanks for the suggestion. Reversing the colors indeed makes the signal more visible. In addition to Fig. 1E, we have also reversed the display of the separated channels for centriolar protein signals in other relevant figures.

- Figure EV1D, why are the actin levels so different?

The original data presented used whole worm lysates prepared with an equal amount of worms. The differences in actin levels are likely due to variations in the western blot technique. Since this non-germline-specific analysis does not significantly contribute to the main findings of the manuscript, we have removed this data from the revised version for clarity and simplicity, given the extensive data already presented.

- Figure 1F/EV1E, 'Clustering and separation' of the centrioles in the germline is not very convincing. Can this be quantified? Is this put into relationship nuclei? Overall, this experiment and the assessment of the number of centrioles in the germline do not add much to the building of the model, they can be moved to the supplemental data or omitted.

We agree that this observation does not significantly contribute to the main findings of the manuscript. As a result, we have omitted this experiment from the revised version.

- Page 7/line 8, Figure EV1A is a typo

Corrected as EV1E.

- Figure EV2D, could authors show the endogenous centrioles as a control? Does the SAS-6 blob localize close to the human centrioles?

We have examined the localization of endogenous centrioles in cells with SAS-6 droplets. Our findings show that the SAS-6 droplets do not colocalize with the endogenous centrioles. This result is presented in Fig EV3A and cited in the text on page 7, lines 2-3:

“However, SAS-6 droplets did not colocalize with the endogenous centrosome (Fig EV3A).”

- Can SAS-6 in HEK cells recruit SAS-5 or SAS-4?

We have examined the ability of SAS-6 in HEK293T cells to recruit SAS-5 and SAS-4. The data are presented in Appendix Fig S4A and B, and cited in the text on page 10, lines 19-23:

“Furthermore, colocalization analysis in cells also suggested the impact of phosphorylation in regulating protein interactions. While SAS-4-GFP is co-recruited with wild-type SAS-6 to form droplets, it became diffused in the cytoplasm when co-expressed with the SAS-6(5D) mutant (Appendix Fig S4A). Similarly, SAS-5 also formed bright droplets with wild-type SAS-6, but not in cells co-expressed with SAS-6(5D) mutant (Appendix Fig S4B).”

- Page 9/line 6,39 and others, the term 'in vivo' is used for worms, cells are referred to as 'in vitro' system, this is not entirely correct...

We have updated the description throughout the manuscript and now use 'in cells' when referring to findings in cells.

- Page 9/12 'Figure EV4B' is incorrect

On page 8, line 4, it is now cited as Fig EV4A, which shows the design of the *sas-5(5A)* mutant.

- Figure 5 C-F, centriole disassembly should be quantified also for diplotene, where this process takes place.

As suggested, we have provided representative images for late prophase oocytes and included quantifications, now presented in Figure 7C-J.

- Figure 6 A would benefit from insets and a better explanation/schematic of how the intensities are measured

We have provided explanations in both the figure legend and the methods section for the measurement of MPM2 immunostaining signal intensity.

Fig 8A legend: “Each point represents the average relative MPM-2 intensity of cells at a specific stage within a gonad.”

Method section on page 23, lines 11-15: “MPM-2 intensity was measured using ImageJ. For each cell, the total (cumulative) intensity and mean background intensity were quantified. The relative MPM-2 intensity was then calculated by normalizing the cumulative MPM-2 intensity to the mean background intensity. The average intensity of cells at a specific stage in a gonad was used for further statistical comparisons.”

- Figure 7 H, could authors show a zoomed-in picture of the centrioles imaged by SIM? Is there any structural difference detected by SIM in the *sas-6(5D)* centrioles?

Due to the resolution limitations of SIM (~100 nm), we were unable to detect structural differences in the centrioles. However, SIM was able to detect a reduction in centriole number, consistent with defects in centriole duplication.

We also attempted U-Ex-STED microscopy with anti-FLAG immunostaining but were not able to confidently assess structural differences at the current resolution. We have attached examples of SAS-6 staining in the indicated genotypes for reference.

- Figure 8, the authors use many different combinations of fragments for their interaction studies. To make it more reader-friendly authors might want to show schematics for the deletion fragments and combinations used in different settings.

Schematics of the deletion fragments are provided in Figure 4A, and the interaction results are summarized in Figures 4F and 4J.

- Figure 9, model oligolymer should be oligomer

Corrected.

- Figure 9, in the model the authors might want to incorporate the change in charge upon CDK-1 phosphorylation

In Figure 5L, we have used the '+' symbol to represent interaction valency, and this has been incorporated into the model in Figure 10. The meaning is explained in the Figure 5L legend:

"The '+' symbols represent interacting valences, which increase when SAS-6 multimers form and decrease when the C-terminus is phosphorylated (not illustrated)."

Referee #3:

The manuscript under consideration by Qi et.al. aims to shed light on the mechanisms driving centrosome elimination during oogenesis, using in vivo experiments in *C. elegans* in combination with in vitro experiments and studies in human cells. The authors provide evidence that the centriole cartwheel protein SAS-6 forms droplets when overexpressed in human cells and can phase separate

in vitro. They then went on to find sites in SAS-6 that are phosphorylated in vivo, and demonstrated that mutation of these sites to alanine delayed centriole elimination while phospho-mimetic mutations reduced the fluorescence intensity of centriole components, suggesting centriole defects. Moreover, the authors found that inhibition of CDK1 delayed centriole disassembly and that CDK1 can phosphorylate SAS-6 in vitro. Putting these findings together, the authors propose that CDK1 phosphorylates SAS-6 to promote centriole disassembly.

This paper contains interesting findings, and most of the experiments are well-performed and rigorous (a few exceptions, where further quantification is needed, are listed in the specific points below). However, the weakest part of the manuscript relates to the claim that these findings shed light on the mechanisms driving centrosome elimination during oogenesis. The effects on centrosome elimination that were examined in vivo (in the SAS-6 mutants or following CDK-1 inhibition), were very mild - there was a slight delay in elimination with CDK1 inhibitors/RNAi and in the SAS-6 5A mutant, but centrioles were still eliminated. Moreover, since there are likely other effects on the germline due to inhibiting CDK1 (a major cell cycle kinase), I didn't think that those effects could be definitively linked to SAS-6 phosphorylation. In addition, the SAS-6 5D mutant experiments were hard to interpret, since the germlines were small and abnormal, and the intensity of SAS-5 and SAS-7 foci was reduced throughout the germline (implying that the centrioles did not form properly - thus, this mutant could not be used to assay the timing of centriole elimination).

Putting all of the data from the paper together, I think that it is fine for the authors to PROPOSE the intriguing model that CDK1 promotes centriole elimination (in the discussion). However, stating this as a conclusion of the paper (in the title and abstract) is too strong based on the data presented. Thus, while this paper contains interesting findings that are likely to be of interest to researchers in the cell division field, the manuscript needs to be substantially revised so that the conclusions drawn are better supported by the experiments presented (either adding more data to support the conclusions, or revising the conclusions to better match the presented data).

We acknowledge the limitations of our current analysis as pointed out by the reviewer and have revised our conclusions accordingly. These revisions better align with both the presented data and the newly generated data during the revision process. Please refer to the revised title, abstract, and text for a comprehensive update.

Specific points:

- Figure EV1D: it is stated in the text that total protein levels of centriolar components don't change following the temperature shift (page 6 line 10), but the western blot samples were from whole worms (not only from the late prophase cells shown in Figure EV1C that are being described). Protein levels could drop in oocytes, but not in the rest of the worm (so a decrease in the germline could be masked).

We attempted to perform western blot analysis with dissected gonads or DIA quantitative mass spectrometry to assess protein abundances. However, due to the limited number of gonads collected (100 per sample), we were unable to detect a signal by western blot, and the mass spectrometry data did not qualify for quantitative analysis. Since the western blot analysis did not significantly contribute to the main findings of the manuscript, we have removed this data from the revised version.

- Figure EV1F: it is stated that centrioles exhibit dynamic movements but only two examples are shown and the centrioles in these examples have only minimal changes over time, so this is not very convincing. The authors would need more examples and quantification to make this claim.

As suggested by Reviewer #2, we agree that this observation does not significantly contribute to the main findings of the manuscript, and it has been removed in the revised version.

- The results in Figure EV2A/B are somewhat variable, depending on the protein and the tag. For example, SPD-2 looks like it forms clear aggregates with mCherry but not GFP. The opposite is true for SAS-4 and SAS-5, calling the results in these figure panels into question. Since no quantification is provided, it is not clear if these images were the rare cases where such aggregates are found. The variability makes me wonder if the observed protein aggregation is really physiological, versus something that can occasionally happen when proteins are overexpressed in cells.

As noted in our response to Reviewer #1, we have reorganized the order of the data to improve the manuscript's flow. This particular data set is not central to the main findings and has been removed in the revised version.

However, we would like to clarify that the images presented are representative, and quantitative data support the protein aggregation properties. It is also worth mentioning that GFP tags are known to have weak dimerizing properties, which could explain some of the differences in aggregation, especially when the tagged protein itself does not strongly aggregate. Importantly, neither GFP nor mCherry tags alone form aggregates.

- Figure 2I: the authors claim that this figure shows that droplets diffuse during mitosis, but this is not convincing from the data presented. Only one example is shown - how frequently does this occur? In what proportion of cells? This result should be quantified somehow if the authors wish to make this claim. Related to this point, Page 8, Line 30 uses the phrasing "the observation that SAS-6 droplets diffused...". If the authors don't provide more convincing evidence to support this claim, this statement should be modified.

Four cell division events were examined, all of which showed a diffused pattern of SAS-6 droplets in mitotic cells. We have now quantified SAS-6 droplet formation in both interphase and mitotic cells using fixed sample slides. The data is presented in Appendix Fig S1A and B, and is cited in the text on page 7, lines 20-21:

"This cell cycle-dependent behavior was further confirmed in fixed cells (Appendix Fig S1A and B)."

- The fact that the authors generated a phospho-specific antibody against Ser408 and confirmed that this residue is phosphorylated in vivo via western blotting (Figure 4B) is a nice result, but it doesn't show that this modification is present in the germline. Page 9, lines 3-4 states that since the lysate preparation protocol used to generate samples for the western blot does not disrupt embryos, the identified modification (phospho-Ser408) likely takes place in the germline. However, aren't centrosomes present in somatic cells? If so, this rationale does not make sense, as the modification detected on the western blot could have come from non-germline cells. If the authors wish to state that the modification is present in the germline more evidence would be needed, beyond just the western

blot. Does this antibody recognize centrioles in the germline (via immunofluorescence)? Unless this is demonstrated (or other additional evidence is provided), the conclusion that SAS-6 is phosphorylated in the germline is not convincing.

We attempted to conduct Western blot analysis using dissected gonads and performed DIA quantitative mass spectrometry to investigate protein phosphorylation. However, due to the limited number of gonads that can be manually dissected and collected for analysis, we were unable to detect a Western blot signal. Additionally, the DIA mass spectrometry analysis did not provide sufficient coverage to allow for quantitative analysis or phosphorylation identification.

We acknowledge the limitation pointed out by the reviewer and have modified the relevant section on page 8, first paragraph, accordingly:

“To further validate these findings, we generated a phosphorylation-specific antibody targeting SAS-6 phosphorylated at Ser408 (pSer408) by immunizing rabbits. This antibody detected phosphorylation of SAS-6::FLAG immunoprecipitated from worm lysates but did not recognize SAS-6(5A)::FLAG immunoprecipitated from lysates of mutant worms carrying phospho-dead mutations, including Ser408Ala (Figs 4B and EV4A). These results confirmed that SAS-6 is phosphorylated *in vivo* at Ser408. Although the five putative phosphorylation sites could not be directly confirmed in the germline, analysis of CDK-1 expression patterns and additional evidence suggest that they are likely phosphorylated in the germline, where they may together perform a regulatory function.”

- Page 9, section beginning on line 36: This section argues that "CDK-1 mediated phosphorylation inhibits phase separation of SAS-6", but the data do not show this. The phospho-mutant analysis suggests that phosphorylation affects phase separation (but does not shed light on which kinase regulates the ability of SAS-6 to phase separate). The demonstration that CDK-1 can phosphorylate SAS-6 *in vitro* suggests that this might be the case, but does not prove it. The language in this section and in the section header should therefore be softened ("our data are consistent with the hypothesis that...etc." rather than stating "our observations reveal that", which is too strong).

We thank the reviewer for the suggestion and have updated the section subtitle to “Phosphorylation Inhibits Phase Separation of SAS-6”. We have removed the statement “CDK-1 mediated phosphorylation inhibits phase separation of SAS-6” from the text within this section. Additionally, we have revised the last paragraph (page 9, lines 14-16) as follows:

“Additionally, *in vitro* phosphorylation of full-length SAS-6 by CDK-1 abolished its phase separation ability (Fig 4N), consistent with the hypothesis that CDK-1-mediated phosphorylation disrupts SAS-6 phase separation.”

- I was confused by the interpretation of Figure 6A, where the authors are using MPM2 staining as a readout of the phosphorylation levels of CDK1 substrates (to determine where CDK1 activity peaks in the germline). It is clear that the MPM2 signal decreases with the PA inhibitor, but it is not gone. So can MPM2 really be used as a readout of CDK-1 activity, since other kinases also contribute to the MPM2 signal? This weakens the argument made by the authors that CDK1 activity peaks at the end of pachytene (where, in their model, it acts to trigger centrosome disassembly).

Several studies have used MPM-2 staining as a marker for CDK-1 activity in both human cells (Gabielli et al., 2007, PMID: 17182611; Rajgopal et al., 2007, PMID: 17171635) and *C. elegans*

(Belew et al., 2023, PMID: 37478128). As noted in our response to Reviewer #2, we further validate the use of the MPM-2 antibody in *cdk-1* mutants. We observed a reduction in the MPM-2 signal, particularly during late meiotic prophase, which aligns with our findings using a CDK-1 inhibitor. This new data is presented in Appendix Fig S5C and is cited in the text on page 11, lines 35-39:

“In *cdk-1(ne2257)* mutants (with partially impaired kinase activity) and worms treated with the CDK-1 inhibitor purvalanol A (PA), MPM-2 signal was reduced but not abolished, indicating that CDK-1 may contribute to part of the MPM-2 signal (Figs 8A, B and Appendix Fig S5C). These findings suggest that CDK-1 activity is activated during oogenesis.”

- Page 11 line 38-39 and page 12 lines 1-2: it is stated that foci formed by the SAS-6 5D mutant are weaker at the transition zone and become almost undetectable in pachytene, leading the authors to suggest that "sustained phosphorylation of SAS-6 could lead to premature elimination of centrosomes". However, the quantification in Figure 7C makes it look like the intensity of foci is similar between the TZ and pachytene, suggesting that these centrioles are not being prematurely eliminated, but that the centrioles are simply weaker/defective throughout the germline (i.e. the centrioles were never properly formed and therefore the signal is weak in the entire germline). To claim that centrioles are being prematurely eliminated, the authors would need to provide stronger evidence.

In response to the reviewers' suggestions and our experimental results, we have revised the section. We also fully agree with Reviewer 1's comment that SAS-6 phosphorylation may inhibit centriole overduplication in pre-meiotic cells, and that this regulation could influence the centriole elimination process. Accordingly, we have revised the text on page 12, lines 12-16, 18-20, and 31-34:

“Cytological analysis revealed that SAS-6 foci in *sas-6(5A)* mutants exhibited stronger fluorescence intensity than wild-type, indicating excessive centriole assembly (Fig 9B and C). In contrast, SAS-6(5D) foci displayed reduced intensity throughout the germline and were nearly undetectable by late pachytene (Fig 9B and C), suggesting that sustained phosphorylation may impair centriole assembly and duplication.”

“In *sas-6(5A)* mutants, GFP::SAS-5 and GFP::SAS-7 foci persisted into diakinesis, indicating delayed centrosome elimination, likely due to centriole overduplication in pre-meiotic cells (Fig 9D-G, EV5A and B).”

“The intensity of GFP::SAS-5 and GFP::SAS-7 foci was significantly reduced in *sas-6(5D)* mutants, with GFP::SAS-5 foci nearly absent by late pachytene (Fig 9I-L). Premature centrosome elimination may result from reduced centriolar material assembly in pre-meiotic cells.”

- For the bead recruitment assays in Figure 8, I was not sure how to interpret the negative results (e.g. the lack of recruitment of the \square M and \square C proteins). Couldn't this be because the truncated mutants do not properly fold? Some evidence to support that these proteins are properly folded would help support the results.

The structures of the remaining domains in the truncated proteins are predicted to remain unaffected. We have provided protein structure predictions in Appendix Fig S3, as cited in the text on page 9, lines 24-26:

“Protein structure predictions indicated that these deletions do not disrupt the folding of the remaining domains (Appendix Fig S3).”

Additionally, an explanation has been added on page 9, lines 29-33:

“Previous models suggest that the N-terminal domain of SAS-6 mediates ring structure formation, while the coiled-coil domain facilitates self-dimerization. The requirement of the central domain for detectable interactions with the C-terminus may reflect the need for multimerization of the C-terminal domain to enable measurable interactions.”

Minor/typos:

- Page 6, Line 22: should this be "duplicated centrioles" (not centrosomes)?

The section describing the movement of the centrosomes has been removed in the revised manuscript.

- The labeling on Figures 8E and 8G is confusing. The columns are labeled + or - ATP, but if I understand correctly these conditions actually denote whether the proteins were preincubated with CDK1 (according to the figure legend). Editing the labels would help clarity.

We have relabeled the columns as 'Pre-P' (now in Fig 5E) and clarified in the figure legend as 'pre-phosphorylation by CDK1' to improve clarity.

Dear Prof. Gao,

Thank you for submitting your revised manuscript. It has now been seen by all of the original referees.

As you can see, referees find that the study is significantly improved during revision and recommend publication. However, I need you to address the points below before I can accept the manuscript.

- Please address the remaining minor concerns of all referees.
- Please reduce the number of keywords to 5, which is the maximum number we can accommodate.
- Please move the Disclosure and competing interests statement after the Acknowledgements section.
- Please remove the 'Author Contributions' section from the manuscript text.
- As per our format requirements, in the reference list, citations should be listed in alphabetical order and then chronologically, with the authors' surnames and initials inverted; where there are more than 10 authors on a paper, 10 will be listed, followed by 'et al.'. Please see <https://www.embopress.org/page/journal/14693178/authorguide#referencesformat>
- Please remove the referral to a piece of data that is not actually shown (pls see p35 - "...are from another field not shown") or include the data.
- We note that the funding information on the National Institutes of Health Office of Research Infrastructure Programs (P40 OD010440) has not been entered into the manuscript tracking system.
- We note that there is one Table EV1 provided in the manuscript; EV tables need to be uploaded separately but if you wish to keep the table in the manuscript, then it should be updated as Table 1.
- During our routine figure checks, we note a potential partial imaging field reuse between Figure 2C and F. Please clarify.
- Our production/data editors have asked you to clarify several points in the figure legends - Figure Legends (main + EV):
 - o Please note that the figure 1 I-K is mislabeled as figure 1 K-M in the manuscript. This needs to be rectified.
 - o Please note that the exact p values are not provided in the legends of figures 1F, 2D, 4J, 6B, 7D, H; 9C, J, L; EV1 F, EV2 C.
 - o Please indicate the statistical test used for data analysis in the legends of figures 1D, F; 2D, 4J, 6B, 7D, H; 8B, 9A, C, J, L; EV1 D, F; EV2 C.
 - o Please note that information related to n is missing in the legends of figures 1F, 7D, F, H, J; 8B, 9A, J, L; EV1 D, F; EV2 C, EV5 D.
 - o Although 'n' is provided, please describe the nature of entity for 'n' in the legend of figure EV5 F.
 - o Please note that the error bars are not defined in the legends of figures 1D, F; 2D, 4J, 6B, 7D, F, H, J; 8B, 8A, C, E, G, J, L; EV1 D, F; EV2 C; EV5 D, F.
- Papers published in EMBO Reports include a 'synopsis' and 'bullet points' to further enhance discoverability. Both are displayed on the html version of the paper and are freely accessible to all readers. The synopsis includes a short standfirst summarizing the study in 1 or 2 sentences (max 35 words) that summarize the paper and are provided by the authors and streamlined by the handling editor. I would therefore ask you to include your synopsis blurb and 3-5 bullet points listing the key experimental findings.
- In addition, please provide an image for the synopsis. This image should provide a rapid overview of the question addressed in the study but still needs to be kept fairly modest since the image size cannot exceed 550 (width) x 300-600 (height) pixels.

Thank you again for giving us to consider your manuscript for EMBO Reports, I look forward to your minor revision.

Kind regards,

Deniz Senyilmaz Tiebe

--

Deniz Senyilmaz Tiebe, PhD
Senior Scientific Editor
EMBO Reports

Referee #1:

The authors have addressed the reviewers' comments, and their revisions have improved the clarity of the manuscript. The question investigated in this study is of interest to the centriole field. Below are a few suggestions to improve clarity. It is not always immediately apparent, from the figures alone, what specific questions the authors are addressing with certain experiments or which system (in vivo, in vitro, or cell lines) is being used, which can make some experiments appear repetitive. It would be useful for the authors to add this information to the figures. Perhaps use the words "in vitro" and "in vivo" (or HEK293/C. elegans). Also, some experiments could have a bit more information in the figure, such as "Fusion events" in figures 2E and 2F.

o Figures specific changes:

Figure 1B: scale bar missing;

Figure 6I: it would be great if the authors could add the respective quantifications to this data set;

Figure 7B: Add flag on Sas-6 name;

Figure 8/9: It seems like the relative GFP intensities quantifications do not fully match the representative images;

Figure 9J/L: name GFP.

Other comments:

Ev1E/F: The authors claim the following: "In addition, the fluorescence intensity of GFP::SAS-7, a PCM component, was reduced by 1,6-hexanediol exposure, but to a lesser extent compared to GFP::SAS-5 and SAS-6::GFP (Fig EV1E and F)." These differences (between Sas-6/Sas-5 and Sas-7) do not appear very strong. Moreover, the results in Figure 1G are not fully convincing on their own. How does the Sas-7 washout look like? The washout experiment may answer this question more precisely. If this conclusion is true, the authors would observe a slower Sas-7 recovery upon 1.6-Hex washout.

Figure 2H/I/J: the authors claim the following "In SAS-6-mCherry droplets, fluorescence intensity fully recovered within 20 seconds after bleaching (Fig 2H and I). Similarly, SAS-6-GFP fluorescence intensity completely recovered within 60 seconds after bleaching (Fig 2H and J)." According to their results, SAS-6-GFP seems to recover slower than SAS-6-mCherry. The authors should acknowledge/discuss these differences.

EV3A: The authors do not observe the SAS-6-mCherry condensates at the centrosome. But do they observe SAS-6 centriolar foci in cells where Sas-6 expression is expected (S-phase and Mitotic cells)?

Figure 4B: "Although the five putative phosphorylation sites could not be directly confirmed in the germline, analysis of CDK-1 expression patterns and additional evidence suggest that they are likely phosphorylated in the germline, where they may together perform a regulatory function." CDK1 expression patterns haven't been analyzed at this point in the manuscript. Also, the authors should specify what the "additional evidence" is.

Referee #2:

Here, Qi et al. submit a revised version of their manuscript. Their original manuscript had compelling data, and I find the revised version is clear and well-structured. Experiments are well-conducted and properly controlled. The authors have addressed all of my comments, and the manuscript is ready for publication.

One minor comment: please indicate n's for Figure 1G

Referee #3:

The revised manuscript by Qi, et.al. is substantially improved. The authors have made a major effort to address the concerns raised by the reviewers. In particular, the reorganization of the manuscript is a major improvement that makes the presentation of the data much clearer. Moreover, the authors have done a good job editing the text so that the conclusions they draw better match their data - I appreciate the thoroughness of this revision. Finally, as before, I think that this manuscript reports interesting findings that will advance the field.

I am largely satisfied with the way the authors addressed my specific concerns. There are only a few remaining concerns, which I think can be addressed without further experiments.

Specific points:

- Quantification is still lacking in a few places. I am confident that the authors have these data - they just need to add more details to the text. For example, one of my original comments related to the fact that only one movie was shown in the original Figure 2I (now Figure 3I). In the response document, the authors clarified that they took movies of four cells that all showed the same behavior, but as far as I can tell this information was not added to the text (though sorry if I missed it somewhere!). I am confident that the data presented is sound (especially since the authors added quantification of fixed cells in the Appendix), but I still think that the number of cells imaged should be stated in cases where only one representative movie is shown. (They could simply add "Phenotype consistent in 4/4 cells imaged" to the legend for the relevant panel.) The same should be done for any other cases where only one representative movie or image is shown and other quantification is not provided (e.g. Figure 2E, 2F, 3C, 3H, 3I, 4L, 4M, 4N, etc.).

- Page 7 line 14: it is stated that ring-like structures "were only occasionally observed". Could the authors provide some numbers in the legend for Figure EV3 to define occasional? (e.g. state how many spherical assemblies were imaged, and how often cartwheel structures were observed).

- Figure EV4B is very small and hard to read - please make the graph larger.

- Page 8 lines 8-11: It is stated that "analysis of CDK-1 expression patterns and additional evidence suggests that (the putative sites) are likely phosphorylated in the germline...". However, no expression patterns or additional evidence is provided in this section. Are the authors referring to the CDK-1 data they show much later in the paper (Figure S5)? Also, what is the "additional evidence" they are referring to? This evidence should be provided or not mentioned. Overall, I found this sentence confusing

and out of place because a proper explanation was not provided (maybe this point could be moved later in the paper, after the "additional evidence" is presented and explained?).

Typos:

- Page 4 line 17: "has been proposed" should be "have been proposed"

EMBOR-2024-60173V3

Dynamic SAS-6 phosphorylation aids centrosome duplication and elimination in *C. elegans* oogenesis

Feifei Qi, Shanshan Yin, Xiangrui Yang, Ning Ju, Bohan Liu, Xing Zhang, Zixuan Zhu, Li Ji, Fuxin Zhang, Li Zhao, Ruoxi Wang, Min Liu, Liangran Zhang, Huijie Zhao, Jun Zhou, and Jinmin Gao

Response to Reviewers' Comments:

We appreciate the reviewers' comments. In response, we have further revised the text and provided new experimental data. Please see our detailed responses below in blue.

Referee #1:

The authors have addressed the reviewers' comments, and their revisions have improved the clarity of the manuscript. The question investigated in this study is of interest to the centriole field. Below are a few suggestions to improve clarity.

It is not always immediately apparent, from the figures alone, what specific questions the authors are addressing with certain experiments or which system (in vivo, in vitro, or cell lines) is being used, which can make some experiments appear repetitive. It would be useful for the authors to add this information to the figures. Perhaps use the words "in vitro" and "in vivo" (or HEK293/C. elegans). Also, some experiments could have a bit more information in the figure, such as "Fusion events" in figures 2E and 2F.

We thank the reviewer for their comments. As suggested, we have added relevant experimental details (e.g., cell types, in vitro assays, meiotic stages) to most panels in Figures 2, 3, 4, and 6, as well as to select panels in Figures 1 and 5. We have also included key experimental information in the titles of the figure legends for clarity.

Figures specific changes:

Figure 1B: scale bar missing;

Added.

Figure 6I: it would be great if the authors could add the respective quantifications to this data set;

Added as suggested (Fig 6J).

Figure 7B: Add flag on Sas-6 name;

Added as suggested.

Figure 8/9: It seems like the relative GFP intensities quantifications do not fully match the representative images;

With our quantification method (described on page 23, first paragraph), measurements from regions lacking centrosomes typically yielded values between 1.5 and 2.0, depending on background signal noise. Because centrosome signals are often very weak during late meiotic prophase, their relative intensities may not consistently exceed those of background regions between gonad arms. As a result, fluorescence intensity measurements are more precise and reliable during earlier meiotic stages. To address the limitations of this method

in later stages, we used the additional quantification approach that assesses the proportion of cells with distinguishable centrosome foci.

We recognize that the original methods section may not have provided sufficient detail. To enhance clarity, we have revised the description of background intensity measurements accordingly on page 23, lines 6-12:

“The corresponding background intensity was defined as the mean of the lower half of the pixels in the "Text image" when sorted by intensity values. ... Measurements from regions without a centrosome typically yielded values between 1.5-2.0, depending on background signal noise.”

Figure 9J/L: name GFP.

GFP fusion protein information added.

Other comments:

EV 1E/F: The authors claim the following: "In addition, the fluorescence intensity of GFP::SAS-7, a PCM component, was reduced by 1,6-hexanediol exposure, but to a lesser extent compared to GFP::SAS-5 and SAS-6::GFP (Fig EV1E and F)." These differences (between Sas-6/Sas-5 and Sas-7) do not appear very strong. Moreover, the results in Figure 1G are not fully convincing on their own. How does the Sas-7 washout look like? The washout experiment may answer this question more precisely. If this conclusion is true, the authors would observe a slower Sas-7 recovery upon 1.6-Hex washout.

As suggested, we performed a washout experiment in the *C. elegans* germline expressing GFP::SAS-7. The fluorescence intensity of SAS-7 foci decreased after 5 minutes of 1,6-hexanediol treatment and significantly recovered within 10 minutes post-washout, similar to SAS-5 and SAS-6. This data has been incorporated into Fig EV1E and F, and the corresponding description has been revised in the text on page 5, lines 27–30:

“In addition, the fluorescence intensity of GFP::SAS-7, a PCM component, decreased upon 1,6-hexanediol treatment and was notably restored within 10 minutes after washout (Fig EV1E and F). These observations suggest that multiple centrosome components may have dynamic properties.”

Figure 2H/I/J: the authors claim the following "In SAS-6-mCherry droplets, fluorescence intensity fully recovered within 20 seconds after bleaching (Fig 2H and I). Similarly, SAS-6-GFP fluorescence intensity completely recovered within 60 seconds after bleaching (Fig 2H and J)." According to their results, SAS-6-GFP seems to recover slower than SAS-6-mCherry. The authors should acknowledge/discuss these differences.

Thanks for pointing out the difference. We have revised the text and added an explanation on page 6, lines 22-27:

“In SAS-6-mCherry droplets, fluorescence intensity fully recovered within 20 seconds after bleaching (Fig 2H and I), while SAS-6-GFP droplets showed complete fluorescence recovery within 60 seconds (Fig 2H and J). These observations demonstrate that SAS-6 undergoes phase separation *in vitro* and that the formed droplets exhibit liquid-like properties. A slightly slower recovery of GFP-tagged SAS-6 may be due to the weak dimerization property of GFP, which increases interaction valency and has been previously shown to modestly affect droplet properties (Wang et al, 2024).”

EV3A: The authors do not observe the SAS-6-mCherry condensates at the centrosome. But do they observe SAS-6 centriolar foci in cells where Sas-6 expression is expected (S-phase and Mitotic cells)?

This is an interesting question. To address it, we performed immunostaining using an anti-SASS6 antibody in cells expressing SAS-6-mCherry. Notably, the antibody detected signals in all major SAS-6-mCherry droplets (see representative IF image below). There are two possible explanations for this observation: first, the anti-SASS6 antibody may exhibit cross-reactivity with the *C. elegans* SAS-6 protein, despite the limited sequence conservation; second, the SAS-6-mCherry droplets may recruit a portion of endogenous SASS6. Given the current limitations of the available antibody, we are unable to draw a definitive conclusion. Therefore, we have opted not to include this data in the manuscript.

Figure 4B: "Although the five putative phosphorylation sites could not be directly confirmed in the germline, analysis of CDK-1 expression patterns and additional evidence suggest that they are likely phosphorylated in the germline, where they may together perform a regulatory function." CDK1 expression patterns haven't been analyzed at this point in the manuscript. Also, the authors should specify what the "additional evidence" is.

We agree that the original placement of this statement was inappropriate. We have revised the text accordingly and relocated it to page 12, lines 4-7:

"Although direct confirmation of the five putative phosphorylation sites at the SAS-6 C-terminus in the germline was not achieved, the *in vivo* expression pattern of CDK-1, along with its direct interaction with and phosphorylation of SAS-6 *in vitro*, suggests that these sites are likely phosphorylated in the germline, where they may together perform a regulatory function."

Referee #2:

Here, Qi et al. submit a revised version of their manuscript. Their original manuscript had compelling data, and I find the revised version is clear and well-structured. Experiments are well-conducted and properly controlled. The authors have addressed all of my comments, and the manuscript is ready for publication.

One minor comment: please indicate n's for Figure 1G

This have now been indicated in the figure legends: "Centrioles from six cells, each likely originating from a different gonad, were reconstructed per condition, with consistent phenotypes observed."

Referee #3:

The revised manuscript by Qi, et.al. is substantially improved. The authors have made a major effort to address the concerns raised by the reviewers. In particular, the reorganization of the manuscript is a major improvement that makes the presentation of the data much clearer. Moreover, the authors have done a good job editing the text so that the conclusions

they draw better match their data - I appreciate the thoroughness of this revision. Finally, as before, I think that this manuscript reports interesting findings that will advance the field.

We thank the reviewer for the comments.

I am largely satisfied with the way the authors addressed my specific concerns. There are only a few remaining concerns, which I think can be addressed without further experiments.

Specific points:

- Quantification is still lacking in a few places. I am confident that the authors have these data - they just need to add more details to the text. For example, one of my original comments related to the fact that only one movie was shown in the original Figure 2I (now Figure 3I). In the response document, the authors clarified that they took movies of four cells that all showed the same behavior, but as far as I can tell this information was not added to the text (though sorry if I missed it somewhere!). I am confident that the data presented is sound (especially since the authors added quantification of fixed cells in the Appendix), but I still think that the number of cells imaged should be stated in cases where only one representative movie is shown. (They could simply add "Phenotype consistent in 4/4 cells imaged" to the legend for the relevant panel.) The same should be done for any other cases where only one representative movie or image is shown and other quantification is not provided (e.g. Figure 2E, 2F, 3C, 3H, 3I, 4L, 4M, 4N, etc.).

We appreciate the suggestion and have added information on the numbers of experiments performed or objects analyzed for panels presenting representative movies or images throughout the figure legends. Please refer to the revised manuscript for details.

- Page 7 line 14: it is stated that ring-like structures "were only occasionally observed". Could the authors provide some numbers in the legend for Figure EV3 to define occasional? (e.g. state how many spherical assemblies were imaged, and how often cartwheel structures were observed).

We have added the following statements in the legend for Figure EV3:

"A total of ten spherical droplets, each 0.8-2.0 μm in diameter, were imaged, with an average of approximately 8 cartwheel-like structures observed per droplet."

- Figure EV4B is very small and hard to read - please make the graph larger.

As suggested, we have enlarged the mass spectrum image shown in the EV4B.

- Page 8 lines 8-11: It is stated that "analysis of CDK-1 expression patterns and additional evidence suggests that (the putative sites) are likely phosphorylated in the germline...". However, no expression patterns or additional evidence is provided in this section. Are the authors referring to the CDK-1 data they show much later in the paper (Figure S5)? Also, what is the "additional evidence" they are referring to? This evidence should be provided or not mentioned. Overall, I found this sentence confusing and out of place because a proper explanation was not provided (maybe this point could be moved later in the paper, after the "additional evidence" is presented and explained?).

Similar suggestion was also made by Reviewer 1. We agree that the original placement of this statement was inappropriate. We have revised the text accordingly and relocated it to page 12, lines 4-7:

"Although direct confirmation of the five putative phosphorylation sites at the SAS-6 C-terminus in the germline was not achieved, the *in vivo* expression pattern of CDK-1, along with its direct interaction with and phosphorylation of SAS-6 *in vitro*, suggests that these

sites are likely phosphorylated in the germline, where they may together perform a regulatory function.”

Typos:

- Page 4 line 17: "has been proposed" should be "have been proposed"

Corrected.

Prof. Jinmin Gao
Shandong Normal University
College of Life Sciences
No. 1 University Road
Changqing District
Jinan, Shandong 250358
China

Dear Prof. Gao,

Thank you for submitting your revised manuscript. I have now looked at everything and all is fine. Therefore, I am very pleased to accept your manuscript for publication in EMBO Reports.

Congratulations on a nice work!

Kind regards,

Deniz Senyilmaz Tiebe

--

Deniz Senyilmaz Tiebe, PhD
Senior Scientific Editor
EMBO Reports

--
